# VTA monosynaptic connections by local glutamate and GABA neurons and their distinct roles in behavior

M. Flavia Barbano [1,4], Huiling Wang[1,4], Shiliang Zhang [2,4], Alexey V. Shevelkin[2], Kevin J. Yu[2], Christopher T. Richie [3], Bing Liu[1], Suyun Hahn [1], Rong Ye[2] & Marisela Morales [1] ✉

The ventral tegmental area (VTA) dopamine neurons have been implicated in diverse behaviors. These VTA$^{dopamine}$ neurons are intermixed with neurons that co-transmit glutamate and GABA (VTA$^{glutamate-GABA}$), transmit glutamate (VTA$^{glutamate-only}$) or GABA (VTA$^{GABA-only}$). In dual recombinase *vglut2-Cre/vgat-Flp* transgenic mice, we combined quantitative ultrastructural analysis with 3D correlative light and electron microscopy and found that VTA$^{glutamate-only}$ neurons frequently established synapses on VTA$^{dopamine}$ and VTA$^{glutamate-only}$ neurons, and that VTA$^{GABA-only}$ neurons mostly synapsed on VTA$^{dopamine}$ neurons. By selective targeting of VTA subpopulations of neurons, we demonstrated that activation of VTA$^{glutamate-only}$ neurons is rewarding and decreases feeding behavior, while activation of VTA$^{GABA-only}$ neurons is aversive. We found that activation of VTA$^{glutamate-only}$ or VTA$^{GABA-only}$ neurons negatively affected learning to obtain food reward, and impaired cue-induced reinstatement of food-seeking behavior. Collectively, we demonstrated the monosynaptic properties of an unexpected VTA microcircuitry in which distinct neuronal components integrate information related to reward, aversion, and feeding.

Ventral tegmental area (VTA) dopamine (VTA$^{dopamine}$) neurons have been implicated in regulating adaptive responses to a variety of positive and negative reinforcers through their specific connectivity with different brain structures[1–3]. However, in addition to dopamine neurons, the VTA has GABA neurons (VTA$^{GABA}$ neurons, expressing the vesicular GABA transporter, VGaT), glutamate neurons (VTA$^{glutamate}$ neurons, expressing the vesicular glutamate transporter type 2, VGluT2), and combinatorial neurons that use more than one type of neurotransmitter as signaling molecules, including neurons that co-transmit glutamate and GABA (VTA$^{glutamate-GABA}$ neurons, co-expressing VGluT2 and VGaT)[3]. The different VTA neurons have unique and shared properties; for instance, we have recently demonstrated that in common with VTA$^{dopamine}$ neurons, the activity of VTA$^{glutamate}$ neurons

increases in response to rewards, aversive stimuli or cues predicting them[4–7]. VTA$^{GABA}$ neurons have been also shown to encode aversive stimuli[5,8–10], to signal unconditioned reward value[11], and to mediate mouse post-stress blunted reward-seeking[12]. Regarding combinatorial neurons, we discovered dual VTA$^{glutamate-GABA}$ neurons that signal biologically relevant appetitive and aversive stimuli, but they do not signal cues that predict these stimuli[5]. Furthermore, some electrophysiological properties of these combinatorial neurons, such as the resting membrane potential, the activation threshold (i.e., rheobase) and the spontaneous firing frequency, resemble those of VTA$^{glutamate-only}$ neurons[13].

The high level of heterogeneity observed in behavioral responses mediated by VTA neurons appears to depend not just on different

[1]Integrative Neuroscience Research Branch, National Institute on Drug Abuse, National Institutes of Health, Baltimore, MD, USA. [2]Confocal and Electron Microscopy Core, National Institute on Drug Abuse, National Institutes of Health, Baltimore, MD, USA. [3]Genetic Engineering and Viral Vector Core, National Institute on Drug Abuse, National Institutes of Health, Baltimore, MD, USA. [4]These authors contributed equally: M. Flavia Barbano, Huiling Wang, Shiliang Zhang. ✉e-mail: maria.barbano@nih.gov

classes of neurons but also on their distinct brain connectivity that includes long range and local connections. While several studies have demonstrated that VTA[GABA] [5,8,11,14,15], VTA[glutamate] [5,15–18] and dual VTA[glutamate-GABA] neurons[5,16] send long-range projections to different brain areas, converging evidence indicates that VTA[non-dopamine] neurons also establish local synaptic connections. For instance, results from ex vivo electrophysiological recordings and electron microscopy analysis have shown that VTA[GABA] neurons establish synapses with VTA[dopamine] and VTA[GABA] neurons[8,19–23]. Similarly, by applying electrophysiological and ultrastructural analysis, we have shown that VTA[glutamate] neurons[22,24] establish monosynaptic connections with dopaminergic and non-dopaminergic neurons. In contrast, it is unclear the extent to which dual VTA[glutamate-GABA] neurons establish local synapses, and the extent to which these neurons contribute to the synaptic organization and behaviors previously ascribed to VTA[GABA] or VTA[glutamate] neurons obtained by using single recombinase transgenic mice.

To selectively target VTA[glutamate-only], VTA[GABA-only] and VTA[glutamate-GABA] neuronal subpopulations, we used dual recombinase *vglut2-Cre/vgat-Flp* transgenic mice in combination with INTRSECT (intronic recombinase sites enabling combinatorial targeting) viral vectors[25,26]. This approach, combined with quantitative ultrastructure analyses, the implementation of 3D correlative light and electron microscopy, optogenetics, and behavioral studies, allowed us to address the VTA synaptic connectivity of VTA[GABA-only], VTA[glutamate-only], or dual VTA[glutamate-GABA] neurons and to determine the extent to which local activation of these VTA neurons plays a role in reward, aversion, or innate motivated behaviors, such as feeding.

## Results

### Selective targeting of VTA[glutamate-only], VTA[GABA-only], and VTA[glutamate-GABA] neurons

To target select subpopulations of VTA neurons, we generated a *vglut2-Cre/vgat-Flp* mouse line (by crossing *vglut2-Cre* mice with *vgat-Flp* mice; Fig. 1A) and injected INTRSECT adeno-associated (AAV) viral vectors into the VTA of these mice (Fig. 1B–D)[5,25]. We used AAV-Con/Fon-ChR2-eYFP vectors requiring the presence of both Cre (Con) and Flp (Fon) recombinases for expression of the enhanced yellow fluorescent protein (eYFP) in VTA[glutamate-GABA] neurons; AAV-Con/Foff-ChR2-eYFP vectors requiring the presence of Cre and the absence of Flp (Foff) recombinases for the expression of eYFP in VTA[glutamate-only] neurons, and AAV-Coff/Fon-ChR2-eYFP vectors requiring the absence of Cre (Coff) and the presence of Flp recombinases for the expression of eYFP in VTA[GABA-only] neurons (Fig. 1B–D).

After confirming VTA neuronal expression of eYFP, we examined the expression of VGluT2 or VGaT mRNAs within each type of VTA eYFP neurons (Fig. 1E). Within the total population of targeted VTA[glutamate-GABA] neurons expressing eYFP (1514 neurons, 3 mice; Fig. 1E, F), we found that more than 87% expressed both VGluT2 and VGaT mRNAs (87.5% ± 2.7%; 1321/1514), about 10% expressed only VGluT2 mRNA (9.6% ± 2.5%; 151/1514 neurons), about 1% expressed only VGaT mRNA (1.0% ± 0.2%; 16/1514 neurons), and few lacked VGluT2 and VGaT mRNAs (1.9% ± 0.5%; 26/1514). Within the total population of targeted VTA[glutamate-only] neurons expressing eYFP (837 neurons, 3 mice; Fig. 1E, G), we found that more than 88% expressed only VGluT2 mRNA (88.7% ± 0.5%; 742/837 neurons), rarely expressed VGaT mRNA alone (0.1% ± 0.1%; 1/837 neurons) or together with VGluT2 mRNA (4.0% ± 0.6%; 34/837 neurons), and a small number lacked VGluT2 and VGaT mRNAs (7.2% ± 0.5%; 60/837 neurons). Within the total population of targeted VTA[GABA-only] neurons expressing eYFP (1149 neurons, 3 mice; Fig. 1E, H), close to 85% expressed only VGaT mRNA (84.4% ± 2.5%; 971/1149 neurons), around 10% lacked VGluT2 and VGaT mRNAs (10.7% ± 2.6%; 120/1149 neurons), close to 3% expressed VGluT2 mRNA alone (2.8% ± 0.5%; 32/1149 neurons), and infrequently had both VGluT2 and VGaT mRNAs (2.2% ± 0.8%; 26/1149 neurons). Collectively, these findings indicate the

utility of using dual *vglut2-Cre/vgat-Flp* mice in combination with INTRSECT viral vectors for the selective targeting of VTA neuronal phenotypes.

We used another cohort of *vglut2-Cre/vgat-Flp* mice for intra-VTA injection of a cocktail of INTRSECT2.0 viral vectors to detect VTA distribution of transfected VTA[glutamate-GABA] (expressing mCherry), VTA[glutamate-only] (expressing eYFP) and VTA[GABA-only] (expressing blue fluorescent protein, BFP) neurons (Sup. Fig. 1A). By confocal microscopy analysis of the rostrocaudal distribution of VTA neurons, we observed VTA[glutamate-GABA] neurons in the medial (bregmas −3.08 mm, −3.40 mm and −3.64 mm), and the mediolateral VTA (bregmas −3.40 mm and −3.64 mm; Sup. Fig. 1B–D'). VTA[glutamate-only] neurons were found throughout the rostrocaudal VTA (bregmas −3.08 mm, −3.40 mm, and −3.64 mm) within its medial and mediolateral aspects, with the highest concentration at the rostral VTA (bregma −3.08 mm; Sup. Fig. 1E–G). In contrast, we found a mediolateral and lateral distribution of VTA[GABA-only] neurons with increasing concentration from the rostral to the caudal VTA (Sup. Fig. 1H–J). Thus, the distribution of the different classes of VTA transfected neurons was consistent with our prior description of distribution of VTA[glutamate-GABA], VTA[glutamate-only] and VTA[GABA-only] neurons[5].

Next, we drove the expression of eYFP in the different classes of VTA neurons by intra-VTA injections of INTRSECT viral vectors in different cohorts of *vglut2-Cre/vgat-Flp* mice (Sup. Fig. 2A). We observed 3 different classes of eYFP-positive neurons in the VTA: VTA[glutamate-GABA], VTA[glutamate-only], and VTA[GABA-only] neurons (Sup. Fig. 2B–D). We then characterized VTA axon terminals (seen as puncta) from local neurons by immunodetection of eYFP, synaptophysin, VGluT2 and VGaT (Sup. Fig. 2E–G). We rarely observed VGluT2-VGaT axon terminals within the VTA (Sup. Fig. 2H–H", Sup. Table 1), indicating that dual VTA[glutamate-GABA] neurons do not establish local synapses. We detected VGluT2-puncta from VTA[glutamate-only] and VGaT-puncta from VTA[GABA-only] neurons within the medial, mediolateral, and lateral aspects of the VTA, and while axon terminals from VTA[glutamate-only] neurons showed similar frequency throughout the rostrocaudal VTA (Sup. Figure 2I–I", Sup. Table 1), the axon terminals from VTA[GABA-only] neurons were more concentrated in the caudal VTA (Sup. Figure 2J-J", Sup. Table 1). Furthermore, we determined that the total number of axon terminals from VTA[glutamate-only] neurons (9107 axon terminals) was higher than those from VTA[GABA-only] neurons (4969 axon terminals; Sup. Table 1).

### VTA[glutamate-only] and VTA[GABA-only] neurons establish monosynaptic synapses with different classes of postsynaptic VTA neurons

Given the infrequent presence of glutamate-GABA terminals in the VTA, we next focused on determining the ultrastructural synaptic connections between axon terminals from VTA[glutamate-only] or VTA[GABA-only] neurons with different classes of postsynaptic VTA neurons (Fig. 2). By triple immuno-electron microscopy and quantitative ultrastructural synaptic analysis of VTA from mice expressing eYFP in VTA[glutamate-only] neurons (Fig. 2A), we determined that axon terminals from these neurons synapsed with similar frequency on VTA[dopamine-only] neurons (25.1 ± 3.6%; 201/794 total synapses; Sup. Table 2; Fig. 2B, C) and on VTA[glutamate-only] neurons (26.0 ± 5.2%; 205/794 total synapses; Sup. Table 2; Fig. 2B, D), but infrequently synapsed on VTA[dopamine-glutamate] neurons (3.6 ± 0.7%; 28/794 total synapses; Sup. Table 2; Fig. 2B, E). In addition, we detected a high frequency of VTA[glutamate-only] neurons synapsing on postsynaptic VTA neurons that lacked eYFP (for the detection of VTA[glutamate-only] neurons) or tyrosine hydroxylase (TH), potentially postsynaptic VTA[glutamate-GABA] or VTA[GABA-only] neurons (45.3 ± 3.1%; 360/794 total synapses; Sup. Table 2; Fig. 2B, F). In a complementary study, by quantitative ultrastructural synaptic analysis of VTA from mice expressing eYFP in VTA[GABA-only] neurons (Fig. 2G), we found that axon terminals from these neurons mainly synapsed on VTA[dopamine-only] neurons (65.9 ± 1.1%; 465/705 total synapses; Sup. Table 3; Fig. 2H, I), and less frequently on VTA[GABA-only] neurons

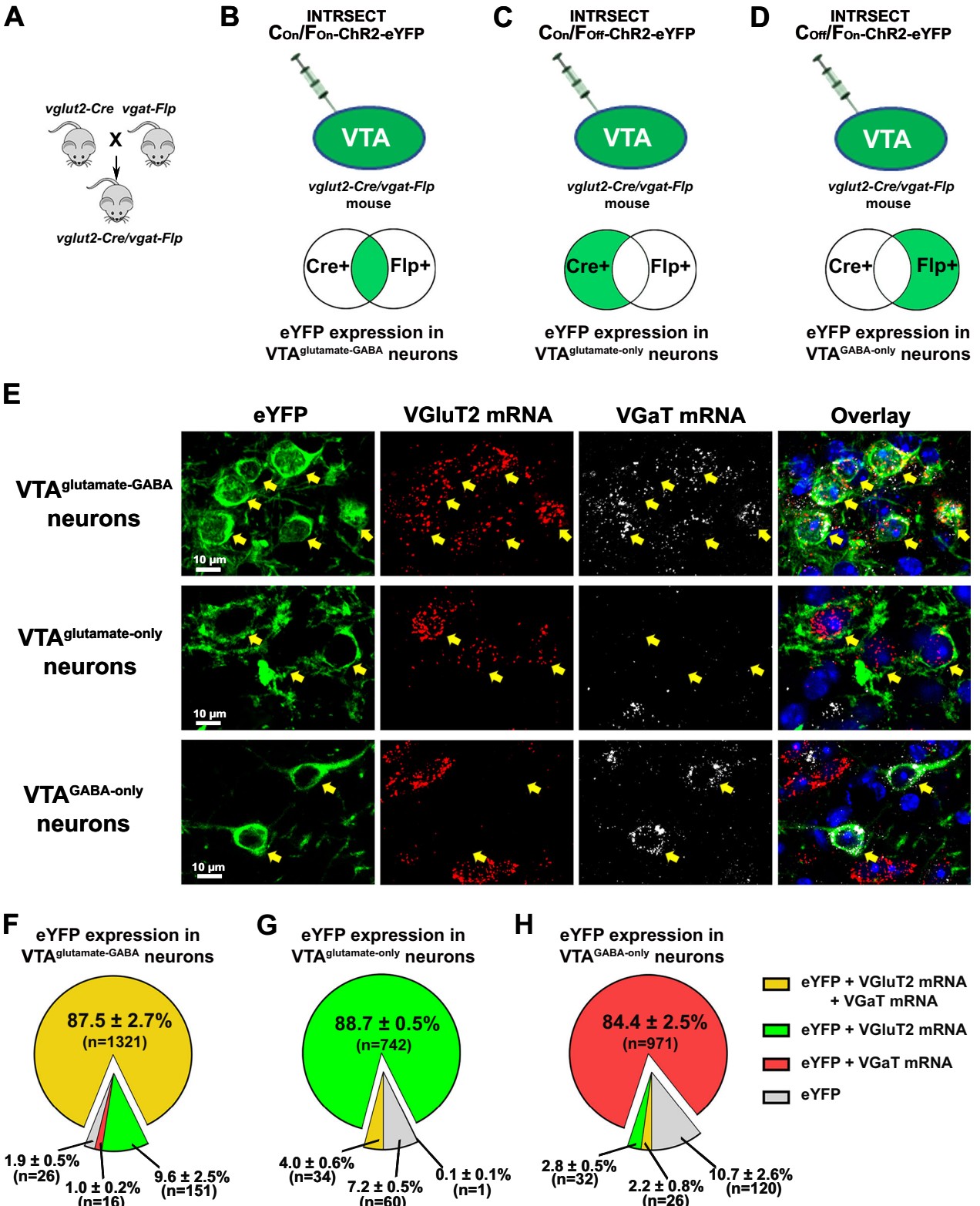

**Fig. 1 | Selective targeting of VTA$^{glutamate-GABA}$, VTA$^{glutamate-only}$ and VTA$^{GABA-only}$ neurons.** **A** Crossing of *vglut2-Cre* mice with *vgat-Flp* mice to generate *vglut2-Cre/vgat-Flp* mice (reproduced from ref. 5). **B**–**D** Intra-VTA injections of INTRSECT AAV-Con/Fon-ChR2-eYFP to target VTA$^{glutamate-GABA}$ neurons (**B**), AAV-Con/Foff-ChR2-eYFP to target VTA$^{glutamate-only}$ neurons (**C**), or AAV-Coff/Fon-ChR2-eYFP vectors to target VTA$^{GABA-only}$ neurons (**D**). **E** VTA$^{glutamate-GABA}$ neurons co-expressing eYFP, VGluT2 and VGaT mRNAs, VTA$^{glutamate-only}$ neurons expressing eYFP and VGluT2 mRNA without VGaT mRNA and VTA$^{GABA-only}$ neurons expressing eYFP and VGaT mRNA without VGluT2 mRNA. **F** Detection of VGluT2 and VGaT mRNAs within the subpopulation of VTA neurons co-expressing eYFP. **G** Detection of VGluT2 mRNAs within the subpopulation of VTA neurons co-expressing eYFP. **H** Detection of VGaT mRNAs within the subpopulation of VTA neurons co-expressing eYFP. Data are shown as mean ± SEM. The number of total counted neurons ("*n*") is shown in each pie graph (three mice/group). Source data are provided as a Source Data file.

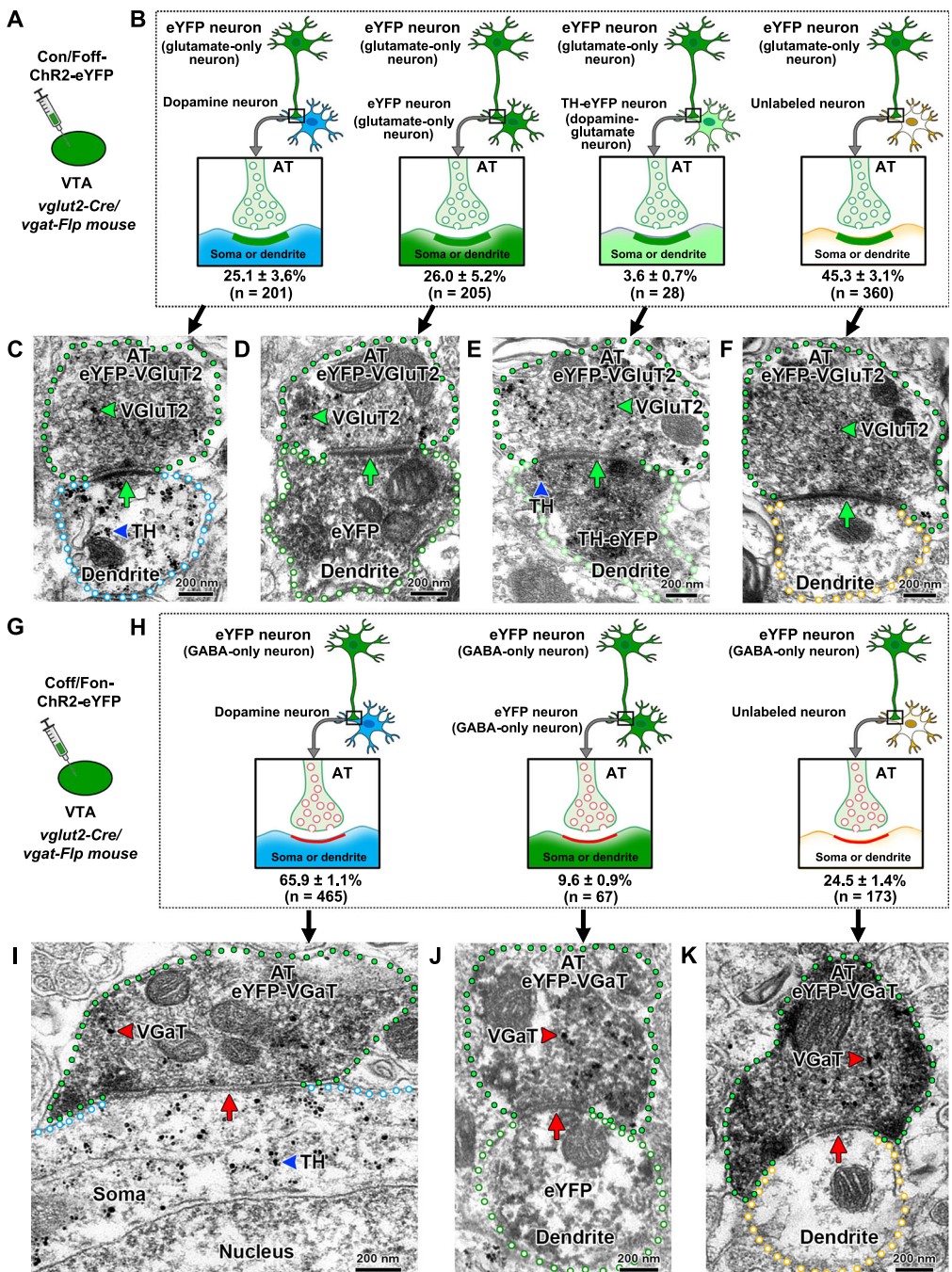

**Fig. 2 | VTA synaptic connectivity by axon terminals from VTA$^{glutamate-only}$ or VTA$^{GABA-only}$ neurons. A** VTA injection of Con/Foff-ChR2-eYFP viral vector in *vglut2-Cre/vgat-Flp* mice to target VTA$^{glutamate-only}$ neurons. **B–F** Asymmetric synapses between axon terminals (ATs) from VTA$^{glutamate-only}$ neurons (eYFP) and postsynaptic VTA$^{dopamine-only}$ (TH), VTA$^{glutamate-only}$ (eYFP), VTA$^{dopamine-glutamate}$ (TH-eYFP) or unlabeled neurons. **B** Diagram indicating the frequency of synapses between ATs from VTA$^{glutamate-only}$ neurons and different classes of postsynaptic VTA neurons (created with Motifolio and BioRender [Zhang, S. (2025) https://BioRender.com/hksfm4r]). **C–F** Electron micrographs showing ATs (green outlines) from VTA$^{glutamate-only}$ neurons co-expressing eYFP (scattered dark material) and VGluT2 (gold particles, green arrowheads) making asymmetric synapses (green arrows) with different classes of dendrites. **C** AT from a VTA$^{glutamate-only}$ neuron synapsing on a VTA$^{dopamine-only}$ dendrite (TH signal, gold particles indicated by blue arrowhead). **D** AT from a VTA$^{glutamate-only}$ neuron synapsing on a VTA$^{glutamate-only}$ dendrite (eYFP signal, scattered dark material). **E** AT from a VTA$^{glutamate-only}$ neuron synapsing on a VTA$^{dopamine-glutamate}$ dendrite (eYFP signal, scattered dark material, and TH signal, gold particles

indicated by blue arrowhead). **F** AT from a VTA$^{glutamate-only}$ neuron synapsing on a dendrite lacking TH and eYFP signals. **G** VTA injection of Coff/Fon-ChR2-eYFP viral vector in *vglut2-Cre/vgat-Flp* mice to target VTA$^{GABA-only}$ neurons. **H–K** Symmetric synapses between ATs from VTA$^{GABA-only}$ neurons (eYFP) and postsynaptic VTA$^{dopamine-only}$ (TH), VTA$^{GABA-only}$ (eYFP) or unlabeled neurons. **H** Diagram indicating the frequency of synapses between ATs from VTA$^{GABA-only}$ neurons and different classes of postsynaptic VTA neurons (created with Motifolio and BioRender [Zhang, S. (2025) https://BioRender.com/hksfm4r]). **I–K** Electron micrographs showing ATs (green outlines) from VTA$^{GABA-only}$ neurons co-expressing eYFP (scattered dark material) and VGaT (gold particles, red arrowheads) making symmetric synapses (red arrows) with different classes of dendrites. **I** AT from a VTA$^{GABA-only}$ neuron synapsing on a VTA$^{dopamine-only}$ soma (TH signal, gold particles indicated by blue arrowhead). **J** AT from a VTA$^{GABA-only}$ neuron synapsing on a VTA$^{GABA-only}$ dendrite (eYFP signal, scattered dark material). **K** AT from a VTA$^{GABA-only}$ neuron synapsing on a dendrite lacking TH and eYFP signals. n = number of axon terminals from 3 *vglut2-Cre/vgat-Flp* mice in panels B and H.

(9.6 ± 0.9%; 67/705 total synapses; Sup. Table 3; Fig. 2H, J). In addition, we determined that ≈ 25% of axon terminals from VTA$^{GABA-only}$ neurons synapsed on VTA neurons that lacked eYFP (for the detection of VTA$^{GABA-only}$ neurons) or TH, potentially postsynaptic VTA$^{glutamate-GABA}$ or VTA$^{glutamate-only}$ neurons (24.5 ± 1.4%; 173/705 total synapses; Sup. Table 3; Fig. 2H, K).

To gain a better understanding of the ultrastructural features of the VTA microcircuitry, we implemented a 3D ultrastructural analysis of synapses between axon terminals from VTA$^{glutamate-only}$ or VTA$^{GABA-only}$ neurons and postsynaptic VTA$^{dopamine-only}$ or VTA$^{glutamate-GABA}$ neurons. To achieve this type of analysis, we developed a correlative light and electron microscopy (CLEM) protocol in combination with injections of a cocktail of INTRSECT viral vectors in the VTA of *vglut2-Cre/vgat-Flp* mice (Fig. 3A, B). In a first set of studies, we drove simultaneous expression of eYFP in VTA$^{glutamate-only}$ neurons and mCherry in VTA$^{GABA-only}$ neurons, which required the developing and validation of an AAV-Coff/Fon-mCherry viral vector (Sup. Fig. 3). By confocal microscopy, we detected expression of eYFP (from VTA$^{glutamate-only}$ neurons) in VGluT2-puncta (eYFP-VGluT2 puncta) and expression of mCherry (from VTA$^{GABA-only}$ neurons) in VGaT-puncta (mCherry-VGaT puncta). Both eYFP-VGluT2 and mCherry-VGaT puncta were adjacent to VTA$^{dopamine-only}$ neurons (Fig. 3B, $C_1$, $E_1$) and to VTA neurons that lacked eYFP, mCherry and TH, potentially VTA$^{glutamate-GABA}$ neurons (Fig. 3B, $D_1$, $F_1$). Next, we applied a CLEM procedure to correlate VTA serial confocal images (providing the molecular nature of the presynaptic and postsynaptic compartments) with the corresponding VTA scanning electron microscopic images (providing the ultrastructural synaptic connectivity by characterizing the type of synapses between presynaptic (axon terminal) and postsynaptic neuronal compartments (soma and dendrites)).

By CLEM analysis, we identified axon terminals from VTA$^{glutamate-only}$ neurons establishing asymmetric synapses on the soma of VTA$^{dopamine-only}$ (Fig. 3$C_2$, $C_3$) or the soma of putative VTA$^{glutamate-GABA}$ neurons (Fig. 3$D_2$, $D_3$). The ultrastructural visualization of asymmetric synapses was achieved by segmentation and 3D reconstruction from CLEM images (Fig. 3$C_4$, $C_5$, $D_4$, $D_5$, Sup. Movie 1, 2). The detection of synapses between terminals from VTA$^{glutamate-only}$ neurons and soma from VTA$^{dopamine-only}$ or putative VTA$^{glutamate-GABA}$ neurons indicates that both VTA$^{dopamine}$ and putative dual VTA$^{glutamate-GABA}$ neurons may potentially be regulated by local excitatory inputs from VTA$^{glutamate-only}$ neurons. By CLEM analysis of the same material, we simultaneously identified axon terminals from VTA$^{GABA-only}$ neurons, establishing symmetric synapses on the soma of VTA$^{dopamine-only}$ (Fig. 3$E_2$, $E_3$) or the soma of putative VTA$^{glutamate-GABA}$ neurons (Fig. 3$F_2$, $F_3$). The ultrastructural visualization of symmetric synapses was achieved by segmentation and 3D reconstruction from CLEM images (Fig. 3$E_4$, $E_5$, $F_4$, $F_5$, Sup. Movies 3, 4). The detection of synapses between terminals from VTA$^{GABA-only}$ neurons and the soma from VTA$^{dopamine-only}$ or putative VTA$^{glutamate-GABA}$ neurons indicates that both VTA$^{dopamine}$ and putative dual VTA$^{glutamate-GABA}$ neurons may potentially be regulated by local inhibitory inputs from VTA$^{GABA-only}$ neurons. To determine the extent to which VTA$^{glutamate-only}$ neurons establish synapses on dual VTA$^{glutamate-GABA}$ neurons, we performed CLEM analysis in the VTA of *vglut2-Cre/vgat-Flp* mice expressing eYFP in VTA$^{glutamate-only}$ neurons and mCherry in VTA$^{glutamate-GABA}$ neurons, driven by intra-VTA injection of a cocktail of INTRSECT viral vectors (Sup. Fig. 4A, B). By confocal microscopy, we detected expression of eYFP (from VTA$^{glutamate-only}$ neurons) in VGluT2-puncta (eYFP-VGluT2 puncta), which were adjacent to VTA$^{glutamate-GABA}$ neurons expressing mCherry (Sup. Fig. 4B, C). By CLEM analysis, we identified axon terminals from VTA$^{glutamate-only}$ neurons establishing asymmetric synapses on VTA$^{glutamate-GABA}$ dendrites (Sup. Fig. 4D). The ultrastructural visualization of asymmetric synapses was achieved by segmentation and 3D reconstruction from CLEM images (Sup. Fig. 4E–H, Sup. Movie 5). Collectively, results obtained by immuno-transmission electron microscopy (Fig. 2) and CLEM (Fig. 3, Sup.

Fig. 4) indicate that VTA$^{glutamate-only}$ neurons synapse with the same frequency on VTA$^{dopamine}$ and VTA$^{glutamate-only}$ neurons, and while it seems that VTA$^{glutamate-only}$ neurons do not establish synapses on dual VTA$^{dopamine-glutamate}$ neurons, they appear to provide a major input to dual VTA$^{glutamate-GABA}$ neurons. In contrast, VTA$^{GABA-only}$ neurons synapse mostly on VTA$^{dopamine}$ neurons and with lower frequency on some VTA$^{glutamate-only}$ and VTA$^{glutamate-GABA}$ neurons.

As indicated above, VTA$^{dopamine}$ neurons (expressing TH) were a major target of both VTA$^{glutamate-only}$ and VTA$^{GABA-only}$ neurons, thus we applied ex-vivo recordings and post hoc immunolabeling to evaluate electrophysiological responses of VTA$^{dopamine}$ neurons in response to release of glutamate from VTA$^{glutamate-only}$ or GABA release from VTA$^{GABA-only}$ neurons. We drove expression of channelrhodopsin2 (ChR2)-mCherry in VTA$^{glutamate-only}$ neurons (Sup. Fig. 5A, D) and found that their photostimulation evoked excitatory postsynaptic currents (EPSCs) in both TH-positive (VTA$^{dopamine}$) and TH-negative (VTA$^{non-dopamine}$) neurons. In all recorded neurons, EPSCs were blocked by tetrodotoxin (TTX) and reinstated by 4-aminopyridine (4-AP, Sup. Fig. 5B, C, E, F), indicating monosynaptic connections. The evoked EPSCs were abolished by the AMPA-receptor antagonist 6-cyano-7-nitroquinoxaline-2,3-dione (CNQX), suggesting an AMPA receptor-mediated response (Sup. Fig. 5B, C, E, F). In addition, VTA photostimulation of axons from VTA$^{glutamate-only}$ neurons induced membrane depolarization and increased the frequency of action potentials in response to higher photostimulation frequencies and durations in TH-positive and TH-negative neurons (Sup. Fig. 5G–J). Next, we drove expression of ChR2-mCherry in VTA$^{GABA-only}$ neurons (Sup. Fig. 6A, D), and found that their photostimulation evoked inhibitory postsynaptic currents (IPSCs) in both TH-positive and TH-negative neurons; these currents were blocked by TTX and reinstated by 4-AP (Sup. Fig. 6B, C, E, F), demonstrating monosynaptic connections. The evoked IPSCs were abolished by the GABA$_A$ receptor antagonist bicuculine, suggesting a GABA$_A$ receptor-mediated response (Sup. Fig. 6B, C, E, F). VTA photostimulation of axons from VTA$^{GABA-only}$ neurons induced membrane hyperpolarization and inhibited action potentials in response to higher photostimulation frequencies and durations in TH-positive and TH-negative neurons (Sup. Fig. 6G–J). Collectively, the electrophysiological and ultrastructural results indicate that VTA$^{glutamate-only}$ neurons monosynaptically activate both VTA$^{dopamine}$ and VTA$^{non-dopamine}$ neurons, while VTA$^{GABA-only}$ neurons monosynaptically inhibit both VTA$^{dopamine}$ and VTA$^{non-dopamine}$ neurons.

To further characterize the VTA microcircuitry, we next determined the extent to which VTA photoactivation of VTA$^{glutamate-only}$ or VTA$^{GABA-only}$ neurons induced cFos (a marker of neuronal activation) in specific subpopulations of VTA neurons. We found that VTA photostimulation of VTA$^{glutamate-only}$ neurons resulted in ≈6 times more cFos-positive neurons in the VTA of mice injected with the Con-Foff-ChR2-eYFP viral vector (Glu-only-ChR2-eYFP mice, 1181.0 ± 23.8 neurons) than in the VTA of mice injected with the control Con-Foff-eYFP viral vector (Glu-only-eYFP mice, 203.3 ± 45.0 neurons; Sup Fig. 7A–H). By comparing cFos expression in different VTA neuronal subpopulations, we found cFos induction of 9.4-fold increase in Glu-only-ChR2-eYFP mice in neurons expressing VGluT2 mRNA alone (2137 neurons in Glu-only-ChR2-eYFP mice vs 227 neurons in Glu-only-eYFP mice), 8.9-fold induction in neurons co-expressing VGluT2 and VGaT mRNAs (454 neurons in Glu-only-ChR2-eYFP mice vs 51 neurons in Glu-only-eYFP mice), and 8.3-fold increase in neurons expressing TH protein (217 neurons in Glu-only-ChR2-eYFP mice vs 26 neurons in Glu-only-eYFP mice; Sup Fig. 7I, J; Sup. Table 4). These findings indicate that VTA release of glutamate from VTA$^{glutamate-only}$ neurons preferentially activates VTA$^{dopamine}$ and dual VTA$^{glutamate-GABA}$ neurons. By VTA photostimulation of VTA$^{GABA-only}$ neurons, we detected ≈ 3 times more cFos-positive neurons in the VTA of mice injected with the Coff-Fon-ChR2-eYFP viral vector (GABA-only-ChR2-eYFP mice, 525.0 ± 74.6 neurons) than in the VTA of mice injected with the control Coff-Fon-eYFP viral

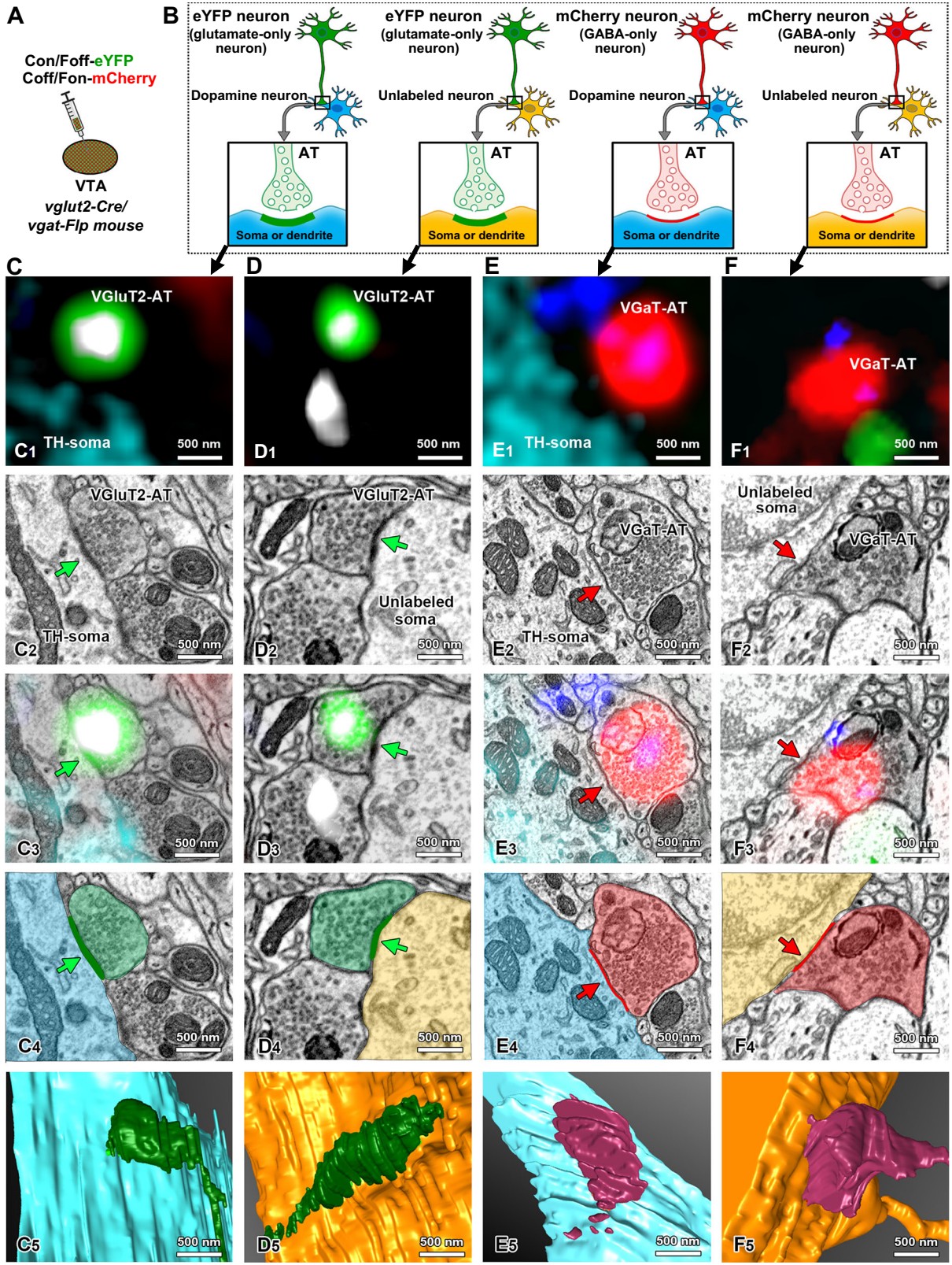

vector (GABA-only-eYFP mice, 162.7 ± 24.5 neurons; Sup Fig. 8A–F). In contrast to expression of cFos in different subpopulations of VTA resulting from local photoactivation of VTA$^{\text{glutamate-only}}$ neurons, cFos induction by VTA photoactivation of VTA$^{\text{GABA-only}}$ neurons was restricted to neurons expressing VGaT mRNA (14.1-fold increase, 1016 neurons in GABA-only-ChR2-eYFP mice vs 72 neurons in GABA-only-eYFP

mice, Sup. Fig. 8G, H, Sup. Table 5). These findings together, with the ultrastructural and electrophysiological recordings provide evidence of a VTA microcircuitry in which VTA$^{\text{glutamate-only}}$ neurons establish monosynaptic excitatory regulation mostly on VTA$^{\text{dopamine}}$ and dual VTA$^{\text{glutamate-GABA}}$ neurons, while VTA$^{\text{GABA-only}}$ neurons establish monosynaptic inhibitory regulation mostly on VTA$^{\text{dopamine}}$ neurons.

**Fig. 3 | Tridimensional ultrastructure of synapses between ATs from VTA<sup>glutamate-only</sup> or VTA<sup>GABA-only</sup> neurons and postsynaptic VTA<sup>dopamine</sup> or putative VTA<sup>glutamate-GABA</sup> neurons using correlative light and electron microscopy (CLEM). A** VTA injections of a viral cocktail with Con/Foff-eYFP (for the expression of eYFP in VTA<sup>glutamate-only</sup> neurons) and Coff/Fon-mCherry (for the expression of mCherry in VTA<sup>GABA-only</sup> neurons) in *vglut2-Cre/vgat-Flp* mice. **B–F** Diagrams of asymmetric synapses (created with Motifolio and BioRender [Zhang, S. (2025) https://BioRender.com/hksfm4r]) between ATs from VTA<sup>glutamate-only</sup> neurons (expressing eYFP) and postsynaptic VTA<sup>dopamine-only</sup> (TH, **C**) or VTA<sup>unlabeled</sup> neurons (without TH, eYFP, and mCherry signals, putative VTA<sup>glutamate-GABA</sup>, **D**); and symmetric synapses between ATs from VTA<sup>GABA-only</sup> neurons (expressing mCherry) and VTA<sup>dopamine-only</sup> neurons (TH, **E**) or VTA<sup>unlabeled</sup> neurons (without TH, eYFP, and mCherry signals, putative VTA<sup>glutamate-GABA</sup>, **F**). **C₁–F₁** VTA confocal micrographs showing one image-frame of ATs out of serial Z-stacks for simultaneous fluorescent detection of five proteins: TH (cyan), eYFP (green), VGluT2 (white), mCherry (red) and VGaT (blue) from four individual ATs. **C₂–F₂** Corresponding scanning electron microscopic (SEM) images of ATs imaged by confocal microscopy showing ATs with synaptic vesicles establishing asymmetric (green arrows) or symmetric (red arrows)

synapses. **C₃–D₃** Correlation of CLEM images showing asymmetric synapses (green arrows) between ATs from VTA<sup>glutamate-only</sup> neurons [VGluT2-AT, co-expressing eYFP (green) and VGluT2 (white)] and postsynaptic soma from a VTA<sup>dopamine-only</sup> neuron expressing TH, (cyan, **C₃**) or a putative VTA<sup>glutamate-GABA</sup> neuron (**D₃**). **E₃–F₃.** Correlation of CLEM images showing symmetric synapses (red arrows) between ATs from VTA<sup>GABA-only</sup> neurons [VGaT-AT, co-expressing mCherry (red) and VGaT (blue)] and postsynaptic soma from a VTA<sup>dopamine-only</sup> neuron expressing TH, (cyan, **E₃**) or a putative VTA<sup>glutamate-GABA</sup> neuron (**F₃**). **C₄–D₄** Segmentation of SEM images showing asymmetric synapses (green arrows) between ATs from VTA<sup>glutamate-only</sup> neurons (green) and postsynaptic soma from a VTA<sup>dopamine-only</sup> neuron (cyan) or a putative VTA<sup>glutamate-GABA</sup> neuron (yellow). **E₄–F₄.** Segmentation of SEM images showing symmetric synapses (red arrows) between ATs from VTA<sup>GABA-only</sup> neurons (red) and postsynaptic soma from a VTA<sup>dopamine-only</sup> neuron (cyan) or a putative VTA<sup>glutamate-GABA</sup> neuron (yellow). **C₅–F₅.** Tridimensional ultrastructural reconstruction of VTA local circuitry from serial SEM images showing ATs from VTA<sup>glutamate-only</sup> or VTA<sup>GABA-only</sup> neurons synapsing on postsynaptic VTA<sup>dopamine-only</sup> or putative VTA<sup>glutamate-GABA</sup> neurons.

## Local photostimulation of different classes of VTA neurons results in distinct behaviors

After showing the synaptic properties of VTA microcircuitry, we next determined the possible role of VTA photostimulation of VTA<sup>glutamate-only</sup>, VTA<sup>GABA-only</sup> and dual VTA<sup>glutamate-GABA</sup> neurons in behavior. By VTA INTRSECT AAV viral injections in different cohort of *vglut2-Cre/vgat-Flp* transgenic mice, we drove the expression of eYFP in control mice (Glu-GABA-eYFP, Glu-only-eYFP or GABA-only-eYFP, Fig. 4B, D, F, Sup. Fig. 9) or ChR2-eYFP in VTA<sup>glutamate-only</sup> neurons (Glu-only-ChR2-eYFP mice, Fig. 4B, Sup. Fig. 9A), VTA<sup>GABA-only</sup> neurons (GABA-only-ChR2-eYFP mice, Fig. 4D, Sup. Fig. 9B) or VTA<sup>glutamate-GABA</sup> neurons (Glu-GABA-ChR2-eYFP mice, Fig. 4F, Sup. Fig. 9C). We placed an optic fiber over the VTA for photostimulation (Sup. Fig. 9D–I) and tested the mice in a three-chamber apparatus, in which mice received continuous trains of 20 Hz photostimulation when they entered a photostimulation-paired chamber, but laser stimulation was terminated when mice exited the chamber (Fig. 4A). We found that Glu-only-ChR2-eYFP mice spent significantly more time in (and accessed more) the laser-paired chamber during photostimulation days, when compared with control Glu-only-eYFP mice (Fig. 4C, Sup. Fig. 10A–C). However, in a subsequent test in the absence of photostimulation (test day), Glu-only-ChR2-eYFP mice did not show preference for the photostimulation-paired chamber (Fig. 4C, Sup. Fig. 10C). In contrast to findings in Glu-only-ChR2-eYFP mice, GABA-only-ChR2-eYFP mice, but not control GABA-only-eYFP mice, avoided the chamber that was paired with laser stimulation (Fig. 4E). We found that when GABA-only-ChR2-eYFP mice were tested in a subsequent test in the absence of photostimulation (test day), they did not show avoidance to the photostimulation-paired chamber (Fig. 4E). The number of entries to the laser-paired chamber was similar for both GABA-only-ChR2-eYFP and control GABA-only-eYFP mice (Sup. Fig. 10D, E). Glu-GABA-ChR2-eYFP and Glu-GABA-eYFP control mice did not show preference or aversion to (and equally accessed) the photostimulation-paired chamber (Fig. 4G, Sup. Fig. 10F, G). We did not observe sex differences in the time spent on the different chambers for any of the experimental groups under study (Sup Fig. 11), and consequently, data from male and female mice were pooled together in subsequent experiments. These findings indicate that while glutamate release from VTA<sup>glutamate-only</sup> neurons drives place preference and GABA release from VTA<sup>GABA-only</sup> neurons drives place aversion, local photoactivation of VTA<sup>glutamate-GABA</sup> neurons does not seem to induce place preference or aversion. Given that VTA also has dual glutamate-dopamine neurons[3], we next tested the extent to which activation of these neurons participate in either place preference or aversion in the three-chamber apparatus. By intra-VTA injections of INTRSECT viral vectors, we drove the expression of eYFP or ChR2 tethered to eYFP in dual

VTA<sup>glutamate-dopamine</sup> neurons of *vglut2-Cre/th-Flp* transgenic mice and found that neither Glu-TH-ChR2-eYFP nor Glu-TH-eYFP control mice showed preference or avoidance for the photostimulation-paired chamber (Sup. Fig. 12).

To further characterize the rewarding or aversive properties of photoactivating VTA neurons, we evaluated the effects of their photostimulation on optical intracranial self-stimulation (oICSS). Both Glu-only-ChR2-eYFP and control Glu-only-eYFP mice were placed in chambers in which they were allowed to earn VTA photostimulation by rotating response wheels; a quarter-turn of one of the wheels (active wheel) resulted in photostimulation of VTA<sup>glutamate-only</sup> neurons (a 0.5 s train of 10 ms light pulses, 20 Hz). During a first period of six daily oICSS training sessions (D1 to D6), the right wheel was designated as the active wheel (Fig. 5A). Glu-only-ChR2-eYFP mice rotated the active wheel significantly more times than the inactive wheel from the third and subsequent sessions (Fig. 5B), and significantly more than control Glu-only-eYFP mice (Fig. 5B). To confirm that the frequency of wheel turning reflected the rewarding effect of the stimulation, we then switched the position of the active wheel (reversal D1 to D4) and found that Glu-only-ChR2-eYFP mice quickly changed their preference to the new active wheel on the second reversal training day (Fig. 5B). We found that Glu-GABA-ChR2-eYFP and Glu-GABA-eYFP mice rotated both wheels without any preference (Fig. 5C). Given that VTA photostimulation in GABA-only-ChR2-eYFP mice induced place aversion, we further determined whether these mice were able to stop experimenter-given photostimulation of VTA<sup>GABA-only</sup> neurons by turning a wheel (active wheel). We placed mice in an apparatus featuring two wheels in which mice were required to turn either the right wheel (D1 to D6) or the left wheel (reversal D1 to D4) to terminate (for 5 s) ongoing photostimulation of VTA<sup>GABA-only</sup> neurons (20 Hz, 0.5 s on/off, Fig. 5D). GABA-only-ChR2-eYFP mice rotated the 'active' wheel (which resulted in photostimulation timeout) significantly more than the 'inactive' wheel (Fig. 5E) and significantly more than the control GABA-only-eYFP mice. Thus, GABA-only-ChR2-eYFP mice discriminated the photostimulation as an aversive stimulus and learned to avoid the aversive stimulation by rotating the appropriate wheel. Next, we prepared another cohort of mice to determine the extent to which the operant response rates were affected by the photostimulation frequency (1–40 Hz) and the stimulation time (1–10 s) at 20 Hz. These photostimulation parameters were selected based on previous findings from our group, which demonstrated by slice or in vivo electrophysiological recordings that VTA<sup>GABA-only</sup>, VTA<sup>glutamate-only</sup>, and the entire population of VTA<sup>glutamate</sup> neurons exhibited firing rates ranging from 1 Hz to over 30 Hz[4,13]. In addition, our prior in vivo recordings demonstrated that appetitive or aversive stimuli resulted in neuronal activation of VTA<sup>GABA-only</sup> or VTA<sup>glutamate-only</sup> neurons with durations

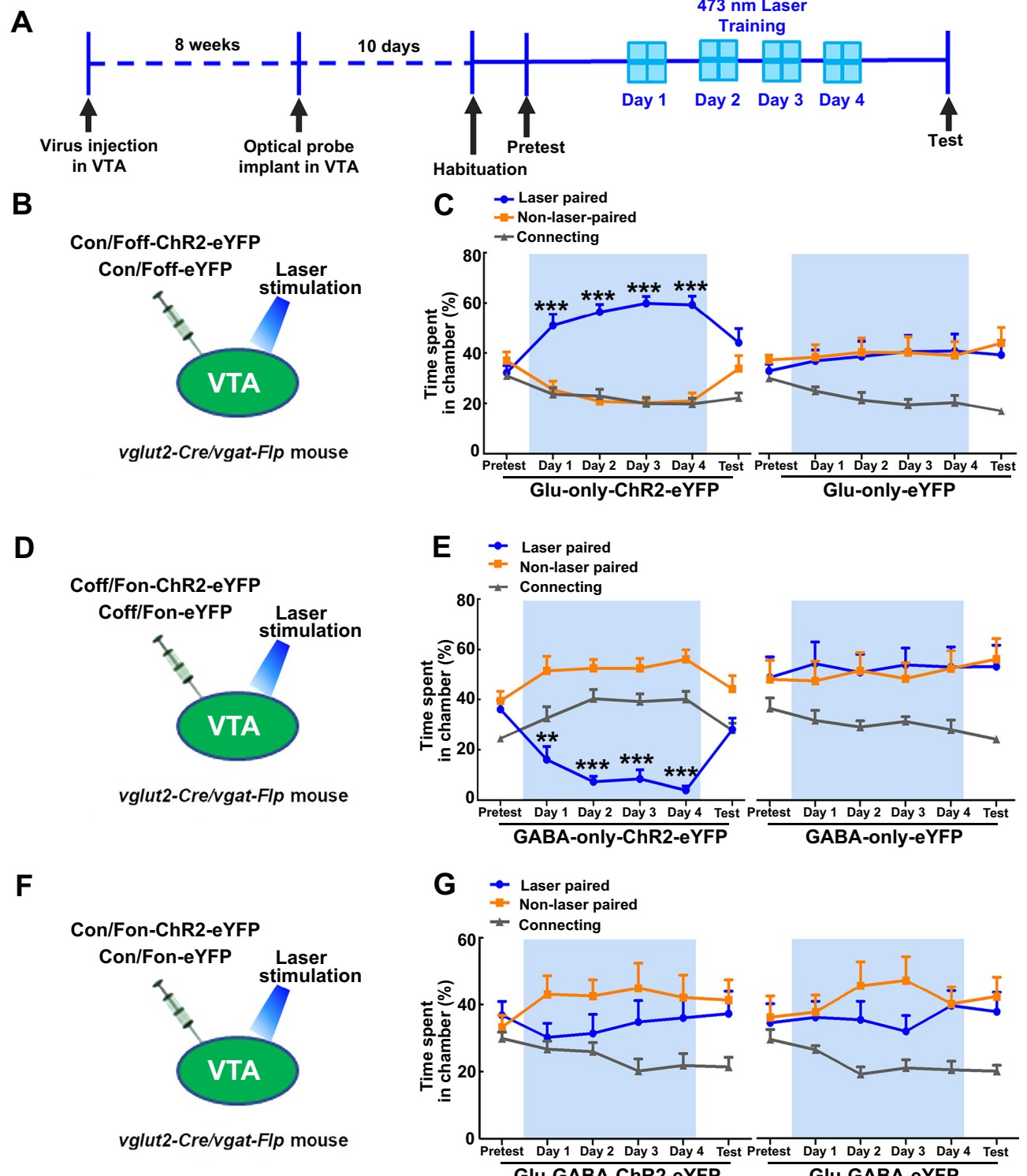

**Fig. 4 | Photostimulation of VTA<sup>glutamate-only</sup> neurons is rewarding, photostimulation of VTA<sup>GABA-only</sup> neurons is aversive, and photostimulation of dual VTA<sup>glutamate-GABA</sup> neurons does not induce reward or aversion. A** Timeline for behavioral testing. **B** VTA injection of Con/Foff viral vectors in *vglut2-Cre/vgat-Flp* mice (to target VTA<sup>glutamate-only</sup> neurons) and VTA photostimulation. **C** Glu-only-ChR2-eYFP mice (n = 7), but not Glu-only-eYFP control mice (n = 9), spent significantly more time in the laser paired chamber during the photostimulation sessions without developing conditioned place preference for the laser paired chamber (chamber × day × group: $F_{(10,140)} = 2.93$; $p = 0.002$). **D** VTA injection of Coff/Fon viral vectors in *vglut2-Cre/vgat-Flp* mice (to target VTA<sup>GABA-only</sup> neurons) and VTA photostimulation. **E** GABA-only-ChR2-eYFP mice (n = 12), but not GABA-only-eYFP

control mice (n = 9), spent significantly less time in the laser-paired chamber during the photostimulation sessions without developing conditioned place aversion for the laser-paired chamber (chamber × group × day: $F_{(10,190)} = 5.76$; $p = 0.00001$). **F** VTA injection of Con/Fon viral vectors in *vglut2-Cre/vgat-Flp* mice (to target VTA<sup>glutamate-GABA</sup> neurons) and VTA photostimulation. **G** Glu-GABA-ChR2-eYFP mice (n = 8) and Glu-GABA-eYFP control mice (n = 8) spent similar time in the laser paired and non-laser paired chambers in the presence or absence of VTA photostimulation (chamber × day × group: $F_{(10,140)} = 0.64$; $p = 0.77$). Data are presented as mean ± SEM. Three-way ANOVA with Tukey HSD post hoc test. **p < 0.01, ***p < 0.001, against non-laser paired chamber. Source data are provided as a Source Data file.

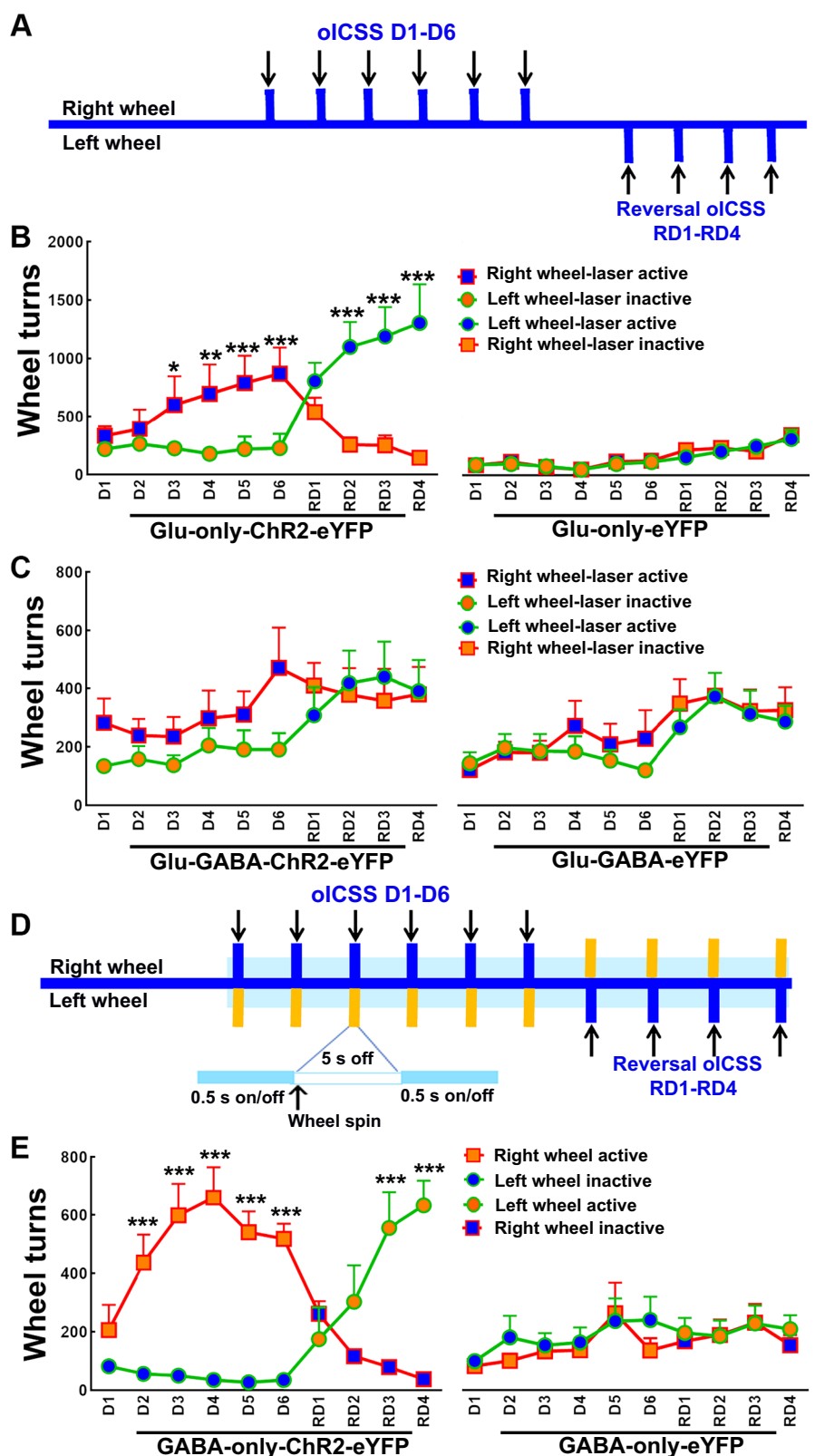

ranging from 1–20 seconds[5]. We found that Glu-only-ChR2-eYFP mice rotated the active wheel more times than the inactive wheel in response to photostimulation of VTA[glutamate-only] neurons (Sup. Fig. 13A) at 5, 10, 20, 40 Hz, but not at 1 Hz (Sup. Fig. 13B), and that they also increased the active wheel turns for all the tested stimulation durations (Sup. Fig. 13C). However, no dose-effect was observed for either the different photostimulation frequencies or durations. In addition, we found that GABA-only-ChR2-eYFP mice rotated the active wheel more times than the inactive wheel to stop photostimulation of VTA[GABA-only] neurons (Sup. Fig. 13D) for all the tested frequencies and stimulation durations (Sup. Fig. 13E, F), without a dose-effect observed for either parameter. While findings obtained after stimulating dual VTA[glutamate-GABA] neurons may reflect activation of long-range projections, evidence suggests that effects observed after stimulating

**Fig. 5 | Photostimulation of VTA$^{glutamate-only}$ neurons supports optical intracranial self-stimulation (oICSS), photostimulation of VTA$^{glutamate-GABA}$ neurons does not support oICSS, and photostimulation of VTA$^{GABA-only}$ neurons induces oICSS avoidance. A** Timeline for oICSS sessions (6 days, D1-D6) in which turning the right wheel resulted in laser activation, followed by reversal oICSS sessions (4 days, RD1-RD4) in which turning the left wheel resulted in laser activation. **B** Glu-only-ChR2-eYFP mice (n = 7) rotated the active wheel for oICSS of VTA$^{glutamate-only}$ neurons significantly more than the inactive wheel and significantly more than Glu-only-eYFP control mice (n = 7) during oICSS training and reversal training sessions (wheel × day × group: F$_{(9,108)}$ = 5.53; p = 0.00001). **C** Glu-GABA-ChR2-mice (n = 11) and Glu-

GABA-eYFP control mice (n = 11) rotated both wheels similarly for oICSS of VTA$^{glutamate-GABA}$ neurons (wheel × day × group: F$_{(9,180)}$ = 0.44; p = 0.91). **D** Timeline for oICSS avoidance sessions (6 days, D1-D6) in which turning the right wheel resulted in laser stimulation halt for 5 s, followed by reversal sessions (4 days, RD1-RD4) in which turning the left wheel resulted in laser stimulation halt for 5 s. **E** GABA-only-ChR2-eYFP mice (n = 11), but not GABA-only-eYFP control mice (n = 9), rotated the active wheel significantly more than the inactive wheel during training and reversal sessions (wheel × day × group: F$_{(9,162)}$ = 5.68; p = 0.00001). Data are presented as mean ± SEM. Three-way ANOVA with Tukey HSD post hoc test. *p < 0.05, **p < 0.01, ***p < 0.001, against inactive wheel. Source data are provided as a Source Data file.

VTA$^{glutamate-only}$, or VTA$^{GABA-only}$ neurons are primarily due to engagement of local VTA microcircuitry rather than long-range pathways (see Discussion section). Collectively, these findings further indicate that (a) VTA glutamate release from VTA$^{glutamate-only}$ neurons, but not from VTA$^{glutamate-GABA}$ or VTA$^{glutamate-dopamine}$ neurons, is rewarding; that (b) VTA GABA release from VTA$^{GABA-only}$ neurons, but not from VTA$^{glutamate-GABA}$ neurons, is aversive; that (c) these effects are not sexually dimorphic, and that (d) they do not depend on the photostimulation frequency or the stimulation time. Given that our anatomical studies indicated that dual VTA$^{glutamate-GABA}$ neurons do not establish local connections within the VTA, and that no effects on reward or aversion were found after VTA photostimulation of these neurons, we next conducted additional behavior tests focusing on VTA$^{glutamate-only}$ and VTA$^{GABA-only}$ neurons.

### Activation of VTA$^{glutamate-only}$ or VTA$^{GABA-only}$ neurons differentially affects feeding behavior

Microdialysis experiments have shown that feeding behavior induces release of dopamine within the VTA and the NAc[27,28]. As such, we next determined the extent to which VTA photostimulation of VTA$^{glutamate-only}$, or VTA$^{GABA-only}$ neurons plays a role in feeding behavior. For these studies, we conducted a free-feeding test consisting of two consecutive 3-min trials in which food-restricted mice were presented with 20 mg chocolate-flavored pellets. We measured both the feeding initiation latency and the amount of food eaten after each trial, and tested mice after 4 days of training, when VTA photostimulation was administered only during the first 3-min trial (day 5, Fig. 6A). We found that VTA photostimulation in Glu-only-ChR2-eYFP mice (targeting VTA$^{glutamate-only}$ neurons, Fig. 6B) during the first 3-min trial resulted in a significant increase in feeding initiation latency (Fig. 6C), together with a decrease in the amount of food eaten (Fig. 6D). These changes were not observed in control Glu-only-eYFP mice (Fig. 6C, D). The increase in feeding initiation latency and the decrease in the amount of food eaten found in Glu-only-ChR2-eYFP mice during the first 3-min trial was not observed in the second 3-min trial when photostimulation was no longer available (Fig. 6C, D). No changes in feeding initiation latency (Fig. 6F) or in the amount of food eaten (Fig. 6G) were observed in the first 3-min trial after VTA photostimulation in GABA-only-ChR2-eYFP mice (targeting VTA$^{GABA-only}$ neurons; Fig. 6E) when compared with control GABA-only-eYFP mice. GABA-only-ChR2-eYFP mice showed a decreased feeding initiation latency (Fig. 6F) along with a trend to eat more than control GABA-only-eYFP mice during the second 3-min trial of the experiment (Fig. 6G), when photostimulation was no longer administered. Collectively, these results indicate that (a) release of glutamate from VTA$^{glutamate-only}$ neurons decreases feeding behavior in food-restricted mice; and (b) terminating the stimulation of VTA$^{GABA-only}$ neurons seems to facilitate feeding behavior in food-restricted mice.

We next determined the extent to which VTA photostimulation of VTA$^{glutamate-only}$, or VTA$^{GABA-only}$ neurons plays a role in the acquisition of food self-administration. For these studies, we trained food-restricted mice to press a lever to obtain 20 mg chocolate-flavored pellets concomitant with a burst of white noise and a light cue while VTA photostimulation was administered throughout the session (5 s on/off; Fig. 7A, Sup. Fig. 14A). We found that VTA photostimulation in Glu-

only-ChR2-eYFP mice (Fig. 7B, Sup. Fig. 14B) significantly delayed learning of the food self-administration task when compared to Glu-only-eYFP control mice, which pressed the lever more times and obtained more pellets throughout the experimental sessions (Fig. 7C, Sup. Fig. 14C). However, the differences in lever presses or pellets obtained were not maintained by the end of the training period (Fig. 7C, Sup. Fig. 14C). In contrast, VTA photostimulation in GABA-only-ChR2-eYFP mice (Fig. 7E, Sup. Fig. 14D) substantially impaired the learning of the food self-administration task, impairment that remained by the end of the training period (Fig. 7F, Sup. Fig. 14E). These results indicate that release of glutamate from VTA$^{glutamate-only}$ has a temporary effect on delaying food self-administration learning, contrasting with VTA release of GABA from VTA$^{GABA-only}$ neurons which has a prolonged effect on delaying food self-administration learning.

In follow up studies, we explored the possible role of VTA neurons in the reinstatement of previously extinguished food-seeking behavior (Fig. 7A). We found that in all tested groups (Glu-only-ChR2-eYFP, Glu-only-eYFP, GABA-only-ChR2-eYFP, or GABA-only-eYFP mice) VTA photostimulation administered 5 min before an extinction session did not induce food-seeking behavior (Fig. 7D, G). In contrast, when VTA photostimulation was administered in the presence of food and cues (white noise and light), it blocked reinstatement of food-seeking behavior in Glu-only-ChR2-eYFP mice (in which VTA$^{glutamate-only}$ neurons were targeted, Fig. 7D). Glu-only-ChR2-eYFP and Glu-only-eYFP control mice showed reinstatement of food-seeking behavior induced by food and cues in the absence of photostimulation (Fig. 7D). GABA-only-ChR2-eYFP mice (in which VTA$^{GABA-only}$ neurons were targeted, Fig. 7E) did not learn the task and consequently reinstatement of food-seeking behavior was not observed (Fig. 7G). These findings indicate that release of glutamate from VTA$^{glutamate-only}$ neurons blocks the reinstatement of food-seeking behavior induced by the presentation of food pellets and cues.

### Photostimulation of VTA$^{glutamate-only}$ neurons decreases anxiety-like behaviors and photostimulation of VTA$^{GABA-only}$ neurons decreases locomotion

To determine the possible role of VTA$^{glutamate-only}$ (Sup. Fig. 15B), or VTA$^{GABA-only}$ neurons (Sup. Fig. 15E) in anxiety-like behaviors measured in an elevated plus maze (to study explicit approach-avoidance conflict), we evaluated the time mice spent in the closed and open arms of an elevated plus maze in three consecutive 5-min trials: before, during and after VTA photostimulation (Sup. Fig. 15A). We found that Glu-only-ChR2-eYFP mice, when compared with Glu-only-eYFP control mice spent significantly more time in the open arms and less time in the closed arms (Sup. Fig. 15C, D). In contrast, VTA photostimulation did not affect anxiety levels in GABA-only-ChR2-eYFP or GABA-only-eYFP control mice (Sup. Fig. 15F, G). These results suggest that release of glutamate from VTA$^{glutamate-only}$ neurons plays a role in the behavioral response of mice to threatening environments by decreasing their anxiety levels.

Next, by using an open field test, we evaluated the locomotor activity evoked by VTA photostimulation of VTA$^{glutamate-only}$, or VTA$^{GABA-only}$ neurons by measuring the total distance travelled and the average speed in three consecutive 5-min trials (before, during

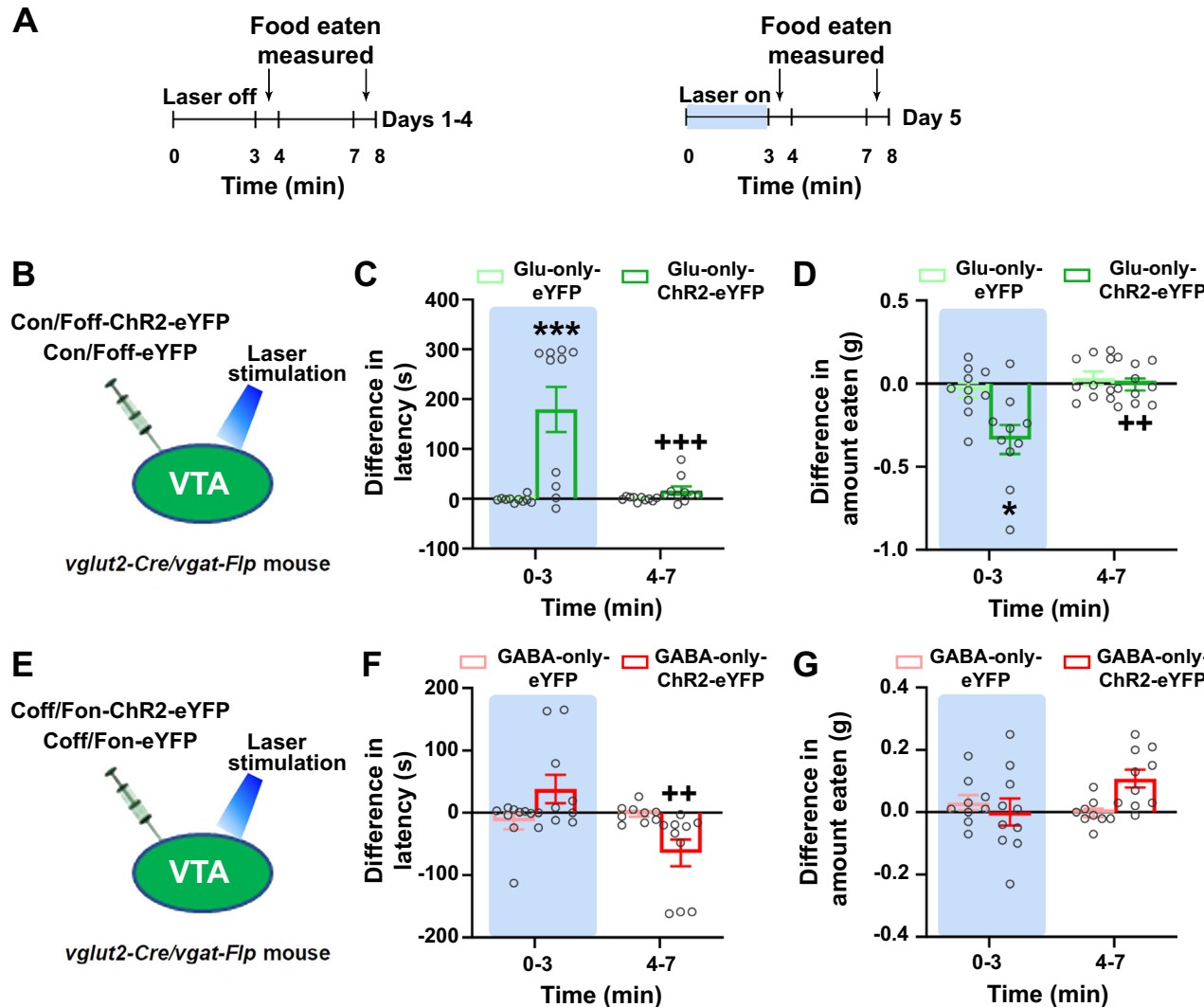

**Fig. 6 | Photostimulation of VTA^glutamate-only neurons decreases feeding behavior, while photostimulation of VTA^GABA-only neurons does not modify feeding behavior. A** Timeline for feeding experiment. **B** VTA injection of Con/Foff viral vectors in *vglut2-Cre/vgat-Flp* mice and VTA photostimulation. **C–D** Photostimulation of VTA^glutamate-only neurons increased feeding initiation latency (**C**; group × phase: $F_{(1,18)} = 15.93$; $p = 0.0009$) and decreased the amount of food eaten (**D**; group × phase: $F_{(1,18)} = 5.91$; $p = 0.03$) in Glu-only-ChR2-eYFP mice (n = 10) but not in Glu-only-eYFP control mice (n = 10). **E** VTA injection of Coff/Fon viral vectors in *vglut2-Cre/vgat-Flp* mice and VTA photostimulation. **F–G** Photostimulation of VTA^GABA-only neurons decreased the feeding initiation latency after termination of photostimulation

(**F**; group × phase: $F_{(1,17)} = 9.07$; $p = 0.008$) without modifying the amount of food eaten (**G**; group × phase: $F_{(1,17)} = 3.78$; $p = 0.07$) in GABA-only-ChR2-eYFP mice (n = 10) but not in GABA-only-eYFP control mice (n = 9). Light-blue rectangles indicate VTA photostimulation. Differences in the amount eaten or the latency to start eating were calculated as the values obtained the 5th day of the experiment minus the values obtained the 4th day of the experiment, and are presented as mean ± SEM. Two-way ANOVA with Tukey HSD post hoc test. *$p < 0.05$, ***$p < 0.001$, against eYFP mice; ++$p < 0.01$, +++$p < 0.001$, against the laser phase. Source data are provided as a Source Data file.

and after VTA photostimulation, Sup. Fig. 16A, B, E). We found that photostimulation did not affect locomotor activity of Glu-only-ChR2-eYFP or Glu-only-eYFP control mice (Sup. Fig. 16C, D). We also determined anxiety-like behavior by evaluating the time spent in the periphery versus the center of the open field arena (threat-induced avoidance test) and found that Glu-only-ChR2-eYFP mice tended to spend more time in the center of the open field during and after the laser stimulation trial when compared to Glu-only-eYFP control mice (Sup. Fig. 17A–D). Collectively, these findings indicate that while VTA^glutamate-only neurons do not appear to be involved in locomotion, they play a role in anxiety (measured by either the elevated plus maze or the open field tests). In clear contrast to the role of VTA^glutamate-only neurons, VTA photostimulation of VTA^GABA-only neurons decreased locomotor activity (Sup. Fig. 16F, G) without preventing mice from eliciting motor responses to avoid

VTA photostimulation of VTA^GABA-only neurons (Fig. 4E) or motor responses related to free-feeding behavior (Fig. 6G). Moreover, we observed that VTA photostimulation in GABA-only-ChR2-eYFP mice induced an increase in the time spent in the center of the open field (Sup. Fig. 17E–G), as an indication of decreased anxiety. This observed decrease in anxiety (by open field arena test) is different from the lack of responses by photostimulation of VTA^GABA-only neurons observed in the elevated plus maze test (Sup. Fig. 15F, G). These findings suggest that while activation of VTA^GABA-only neurons decreases locomotor activity, they do not prevent increases of motor responses elicited by environmental stimuli related to feeding or aversion, and that activation of VTA^GABA-only neurons appear to play a role in anxiety related to threat-induced avoidance but not in anxiety related to explicit approach-avoidance conflict. Alternatively, VTA photostimulation of VTA^GABA-only neurons may

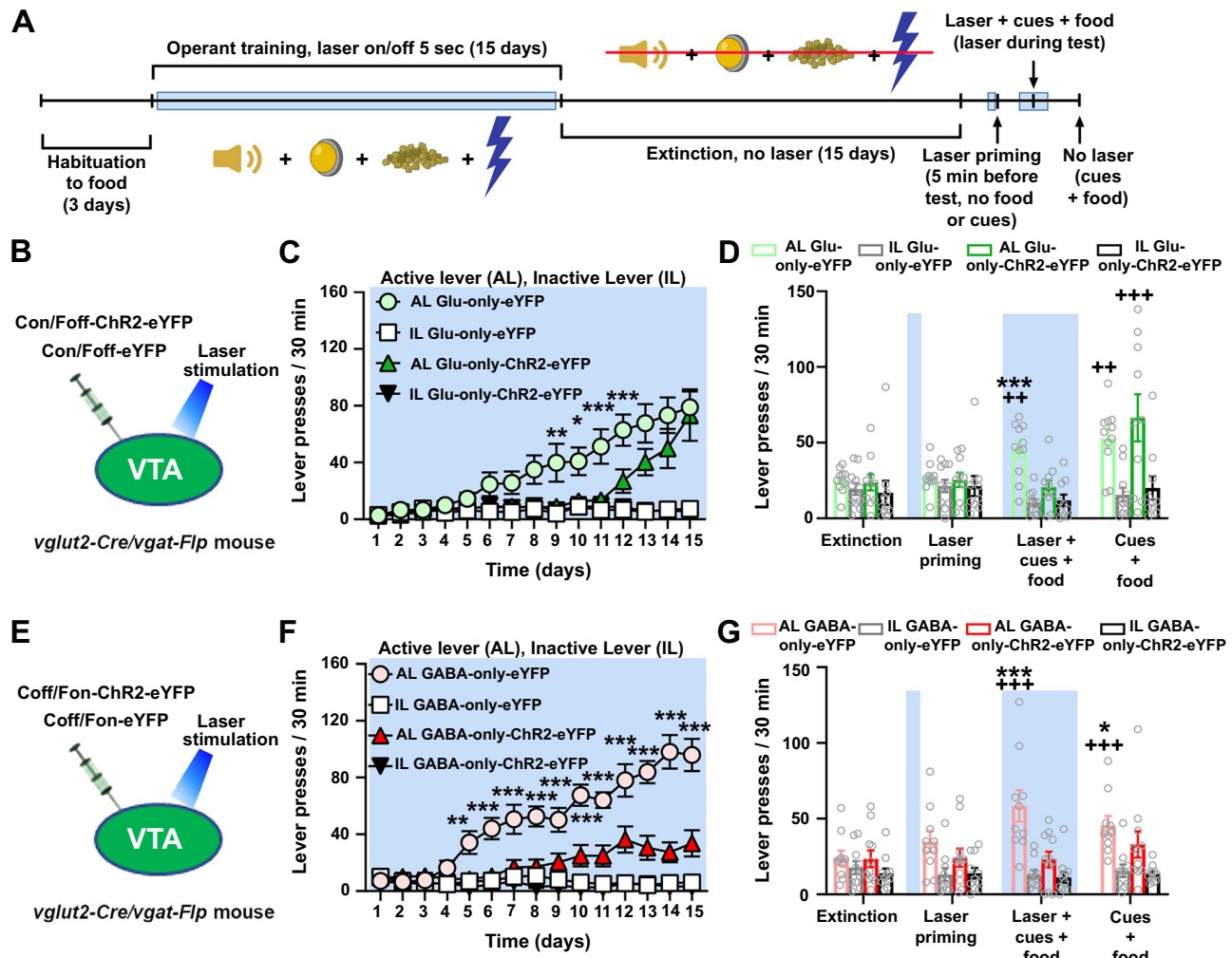

**Fig. 7 | Food self-administration and reinstatement of food-seeking behavior after photostimulation of VTA$^{glutamate-only}$ or VTA$^{GABA-only}$ neurons. A** Timeline for food self-administration and reinstatement conditions (created with BioRender. Barbano, F. (2025) https://BioRender.com/7ec597z). **B–D** VTA injection of Con/Foff viral vectors in *vglut2-Cre/vgat-Flp* mice and VTA photostimulation (**B**) delayed lever pressing for food reward in Glu-only-ChR2-eYFP mice (*n* = 10) when compared with Glu-only-eYFP mice (**C** n = 11, group × day × lever: F$_{(14,266)}$ = 2.20; p = 0.008). Photostimulation of VTA$^{glutamate-only}$ neurons for 5 min prior to the reinstatement test did not modify lever presses in Glu-only-ChR2-eYFP or Glu-only-eYFP mice (**D**, group × day × lever: F$_{(1,19)}$ = 0.11; p = 0.75), but lever presses in the presence of photostimulation during the reinstatement test were reduced in Glu-only-ChR2-eYFP mice when compared to Glu-only-eYFP mice (group × day × lever: F$_{(1,19)}$ = 9.86; p = 0.005). Reinstatement of food-seeking behavior in Glu-only-ChR2-eYFP mice was restored in the absence of VTA photostimulation (**D** group × day × lever:

F$_{(1,19)}$ = 0.48; p = 0.50). **E–G** VTA injection of Coff/Fon viral vectors in *vglut2-Cre/ vgat-Flp* mice and VTA photostimulation (**E**) decreased total lever presses for food reward in GABA-only-ChR2-eYFP mice (n = 11) when compared with GABA-only-eYFP mice (**F** n = 10, group × day × lever: F$_{(14,266)}$ = 8.62; p < 0.00001). Photostimulation of VTA$^{GABA-only}$ neurons 5 min prior (group × day × lever: F$_{(1,19)}$ = 2.95; p = 0.10) or during the reinstatement test did not modify lever presses in GABA-only-ChR2-eYFP mice, while GABA-only-eYFP mice increased their lever pressing during the reinstatement test (**G**, group × day × lever: F$_{(1,19)}$ = 16.90; p = 0.0006). Reinstatement of food-seeking behavior in GABA-only-ChR2-eYFP mice was not restored in the absence of VTA photostimulation (**G**, group × day × lever: F$_{(1,19)}$ = 4.28; p = 0.05). Light-blue rectangles indicate VTA photostimulation. Data are presented as mean ± SEM. Three-way ANOVA with Tukey HSD post hoc test. *p < 0.05, **p < 0.01, ***p < 0.001, against ChR2-eYFP mice; ⁺⁺p < 0.01, ⁺⁺⁺p < 0.001, against extinction values. Source data are provided as a Source Data file.

influence motivation or affect the ability to discriminate between the open and closed arms in the elevated plus maze test.

**Local inhibition of VTA$^{glutamate-only}$, VTA$^{glutamate}$ or VTA$^{GABA}$ neurons differentially affects behavior**

As a follow up to the gain-of-function studies described above, we set up loss-of-function studies by the selective inhibition of VTA$^{glutamate-only}$ or VTA$^{GABA-only}$ neuron by using INTRSECT viral vectors to drive the selective expression of halorhodopsin (Halo) or inhibitory DREADDs (designer receptors exclusively activated by designer drugs) in the VTA of dual *vglut2-Cre/vgat-Flp* mice. Following intra-VTA injection of INTRSECT viral vectors, we determined selective transfection of targeted VTA neurons by immunodetection of reporter fluorescent proteins (eYFP or mCherry) and RNAscope for the detection of both VGaT

and VGluT2 mRNAs (Sup. Fig. 18, 21, 25). For the selective inhibition of VTA$^{glutamate-only}$ neurons, we tested a commercially available INTRSECT Con/Foff-Halo viral vector tethered to eYFP that failed to selectively transfect VTA$^{glutamate-only}$ neurons (Sup. Fig. 18). While we identified several transfected neurons by eYFP expression, many of them lacked transcripts encoding VGluT2 or VGaT (Sup. Fig. 18E). Considering the lack of an available selective INTRSECT Con/Foff-Halo viral vector and given that axons from dual VTA$^{glutamate-GABA}$ neurons do not establish local synapses, we prepared cohorts of mice for the selective transfection of the total population of VTA$^{glutamate}$ neurons (Sup. Fig. 19A). We placed one optic fiber over the VTA for photoinhibition (Sup. Fig. 20A) and tested Glu-Halo-eYFP and control Glu- eYFP mice in the three-chamber apparatus in which one of the chambers was paired with the laser (Sup. Fig. 19B, C). We found that Glu-Halo-eYFP mice

spent significantly less time in the laser-paired chamber during most of the photoinhibition days when compared with Glu-eYFP control mice (Sup. Fig. 19B, C), contrary to the place preference to the laser paired chamber shown by Glu-ChR2-eYFP mice. In a subsequent test in the absence of photoinhibition, Glu-Halo-eYFP mice did not show avoidance for the laser-paired chamber (Sup. Fig. 19B). Next, we tested food restricted mice and found that photoinhibition of VTA^glutamate neurons in Glu-Halo-eYFP mice did not alter feeding initiation latency compared to Glu-eYFP controls during photoinhibition trials (0–3 min). However, in the absence of photoinhibition (4–7 min trials), Glu-Halo-eYFP mice showed an increased latency relative to controls (Sup. Fig. 19D). In addition, food intake did not differ between Glu-Halo-eYFP and Glu-eYFP control mice during photoinhibition of VTA^glutamate neurons; nonetheless Glu-Halo-eYFP mice consumed more food in the presence of photoinhibition compared to its absence (Sup. Fig. 19E). Furthermore, in the absence of photoinhibition, Glu-Halo-eYFP mice showed a non-significant trend towards reduced food consumption compared to Glu-eYFP control mice ($p = 0.06$, Sup. Fig. 19E). Locomotor activity and anxiety-like behaviors observed either in an open field arena or in an elevated plus maze were not modified by photoinhibition of VTA^glutamate neurons (Sup. Fig. 19F–K).

In a follow up study, we tested INTRSECT inhibitory DREADDs to selectively target VTA^glutamate-only neurons by intra-VTA injection of Con/Foff-Gi-DREADD-mCherry viral vector in dual *vglut2-Cre/vgat-Flp* mice (Sup. Fig. 21A). We detected many VTA mCherry transfected neurons that expressed VGluT2 mRNA but lacked VGaT mRNA (Sup. Fig. 21B, C), indicating the selective transfection of Gi-mCherry in VTA^glutamate-only neurons (Sup. Fig. 21C). We further validated the functionality of this viral vector through electrophysiological recordings, which demonstrated that VTA^glutamate-only neurons exhibited reduced firing rate following the bath application of the specific DREADD ligand JHU 37160 (J60, Sup. Fig. 21D, E). We used this INTRSECT vector to drive the expression of Gi-mCherry (Glu-only-Gi-mCherry mice) in VTA^glutamate-only neurons for behavioral studies in which these mice and respective controls (Glu-only-mCherry mice) were tested after intra-VTA administration of artificial cerebrospinal fluid (aCSF) or J60 (0.1 µg/µl, Sup. Fig. 22A, J). In a classic place conditioning experiment in the three-chamber apparatus, mice received an intra-VTA microinjection of J60 prior to confinement to one chamber (J60-paired chamber). The next day, mice received an intra-VTA aCSF microinjection and were confined to the chamber in which they did not receive J60 injections. When mice were tested in a J60-free state after the conditioning sessions, we found that Glu-only-Gi-mCherry mice spent less time in the chamber previously associated with the J60 administration, indicating the development of conditioned place aversion towards the chamber in which VTA^glutamate-only neurons were inhibited (Sup. Fig. 22B, C). We then evaluated feeding behavior in food restricted mice and found that Glu-only-Gi-mCherry mice showed no significant changes in feeding initiation latency (Sup. Fig. 22D) but consumed more food following chemogenetic inhibition of VTA^glutamate-only neurons (Sup. Fig. 22E). Locomotor activity and anxiety-like behaviors were not affected by chemogenetic inhibition of VTA^glutamate-only neurons (Sup. Fig. 22F–I). Collectively, these findings demonstrate that both local optogenetic inhibition of the total population of VTA^glutamate neurons and selective local chemogenetic inhibition of VTA^glutamate-only neurons mediate the same behaviors, further supporting the role of VTA microcircuitry in regulating distinct behavioral outcomes. The convergence of results across independent methodologies underscores the specificity and functional relevance of VTA^glutamate-only neurons in modulating reward and feeding behavior. Notably, our observations of local photoinhibition of VTA^glutamate neurons yielded results similar to those obtained using INTRSECT inhibitory DREADDs, reinforcing the consistency and reliability of these findings.

Next, we tested a commercially available INTRSECT Halo viral vector tethered to eYFP to target VTA^GABA-only neurons (Sup. Fig. 23A).

However, eYFP immunoreactivity was not detected in the VTA of *vglut2-Cre/vgat-Flp* mice, indicating transfection failure (Sup. Fig. 23B, C). Given the lack of a selective available INTRSECT viral vector to optogenetically inhibit VTA^GABA-only neurons and considering our findings showing that VTA^glutamate-GABA neurons do not establish local synaptic contacts, we next tested an alternative approach by driving the expression of Halo in the total population of VTA^GABA neurons (Sup. Fig. 24) and placed two optic fibers over the VTA for photoinhibition (Sup. Fig. 20). We prepared cohorts of mice targeting VTA^GABA neurons (Sup. Fig. 24A), tested them in the three-chamber apparatus and found that GABA-Halo-eYFP mice spent significantly more time in the laser-paired chamber during the photoinhibition days, when compared with GABA-eYFP control mice (Sup. Fig. 24B, C). In addition, GABA-Halo-eYFP mice developed a conditioned place preference for the laser-paired chamber that was evident in a subsequent test in the absence of photoinhibition (Sup. Fig. 24B). When feeding behavior was evaluated by the free-feeding test in food restricted mice, we found that during photoinhibition of VTA^GABA neurons, GABA-Halo-eYFP mice showed no significant changes in feeding initiation latency compared to GABA-eYFP control mice (Sup. Fig. 24D). However, in the absence of photoinhibition, GABA-Halo-eYFP mice showed reduced feeding initiation latency compared to both the laser trials and GABA-eYFP control mice (Sup. Fig. 24D). The amount of food eaten was unaffected by photoinhibition in GABA-Halo-eYFP and GABA-eYFP control mice. (Sup. Fig. 24E). Locomotion, represented by the distance travelled and the average speed, was increased in GABA-Halo-eYFP, but not in GABA-eYFP control mice during photoinhibition of VTA^GABA neurons (Sup. Fig. 24F, G). Anxiety-like behaviors were not modified by photoinhibition of VTA^GABA neurons in either GABA-Halo-eYFP or GABA-eYFP control mice (Sup. Fig. 24H–K). We then tested a custom-packaged INTRSECT inhibitory DREADD by injecting the Coff/Fon-Gi-DREADD-mCherry viral vector into the VTA of dual *vglut2-Cre/vgat-Flp* mice (Sup. Fig. 25A). Although selective expression of mCherry was observed in VTA^GABA-only neurons following injection of this vector (Sup. Fig. 25B, C), electrophysiological recordings of mCherry-expressing VTA neurons showed no response to J60 administration, indicating a functional failure of the viral vector (Sup. Fig. 25D, E).

Collectively, these loss-of-function findings, together with the gain-of-function (photostimulation) results, highlight the functional significance of an unexpected VTA microcircuitry, supporting a role for VTA^glutamate-only neurons in mediating reward and feeding behavior and for VTA^GABA-only neurons in aversion and locomotion, while leaving open the possibility that long-range projections may have also contributed to the observed effects.

## Discussion

The existence of cortical microcircuits regulating information processing and output is well established[29]. In contrast, the understanding of microcircuitry components, synaptic connectivity and roles in behavior is less clear for midbrain structures, such as the VTA. The VTA is best known for containing dopamine neurons that participate in diverse aspects of behavior mediated by their interactions with different brain areas, including synaptic regulation by local non-dopamine neurons. Here, we provide evidence that the VTA contains a local microcircuitry, in contrast to the well-studied effects of activating VTA long-range projections (Sup. Fig. 26). By characterizing the synaptic connectivity among VTA neurons and their behavioral contributions, we highlight the functional significance of local connectivity within this structure and switch the focus to future studies in considering the importance of both VTA microcircuitry and long-range projections in mediating different aspects of behavior. By implementing an innovative 3D ultrastructural reconstruction approach combined with INTRSECT viral vectors and dual recombinase transgenic mice to target specific subpopulations of VTA neurons, we uncovered the synaptic connectivity of VTA local circuitry established

by local glutamate-GABA, glutamate, and GABA neurons among them and on dopamine neurons (Sup. Fig. 26). This connectivity emphasizes the importance of local interactions within the VTA, as distinct from the effects mediated by long-range inputs. We further demonstrated that VTA[glutamate] and VTA[GABA] neurons play distinct or overlapping roles in reward, aversion, and motivated behaviors.

By in vivo recordings, we have previously shown that VTA[glutamate-GABA] neurons increase their activity in response to aversive or appetitive stimuli but not to the cues predicting them[5]. While it is unclear the origin of inputs that regulate the activity of VTA[glutamate-GABA] neurons, such regulation is likely to be mediated in part by VTA local neurons, as we found that VTA[glutamate-GABA] neurons receive major inputs from both VTA[glutamate-only] and VTA[GABA-only] neurons. Although VTA[glutamate-GABA] neurons partially exhibit common features with VTA[glutamate-only] and VTA[GABA-only] neurons[13], we observed that their local stimulation is not rewarding or aversive. Given that VTA[glutamate-GABA] neurons release both excitatory and inhibitory neurotransmitters, it is likely that they play a modulatory role in behavior. In this regard, recent findings showed that activation of VTA[glutamate-GABA] neurons enhanced the salience of both positive and negative behavioral experiences without inducing place preference or aversion[30].

In our previous in vivo recordings, we found that the activity of VTA[GABA-only] neurons increases in response to sucrose delivery, foot shock and cues predicting the absence of either reward or foot shock delivery[5]. Here, we extended these observations by demonstrating that either activation or inhibition of VTA[GABA-only] neurons had no effects on free food pellet feeding. In contrast, studies using Pavlovian conditioning or free sucrose consumption assays had reported that stimulation of the total population VTA[GABA] neurons disrupted sucrose licking responses[8]. In addition, studies comparing feeding behavior in sated and food-restricted mice presented with standard or palatable food after stimulation of VTA[GABA] neurons showed an increase in food intake in sated mice presented with palatable food, but no changes in food-restricted mice presented with standard chow. Furthermore, food-restricted mice did not change their food intake when presented with palatable food but decreased their intake when presented with standard chow during stimulation of VTA[GABA] neurons[31]. These apparent discrepancies in behaviors might result from the behavioral paradigm employed, the type of reward used, or the homeostatic state of the mice. We also found that activation of VTA[GABA-only] neurons severely impaired learning for food self-administration and given that a precise timing in the firing of VTA[GABA] neurons seems to be necessary to promote learning[9], it is likely that the asynchronous firing induced by photostimulation of VTA[GABA-only] neurons disrupted learning of the instrumental food self-administration task. Thus, we conclude that the lack of food-seeking behavior observed during the food reinstatement sessions was due to the lack of proper learning of the task during training. A limitation of the present study is the absence of time-locked manipulations during specific phases of feeding behavior (i.e., appetitive vs. consummatory phases), which could help uncover alternative interpretations of our findings, such as effects on motivation, disruption of action-outcome associations, or broader modulation of dopaminergic circuitry. Nonetheless, our approach provides an important step in characterizing the contributions of VTA[glutamate-only] and VTA[GABA-only] neurons to food-motivated behaviors, offering a foundation for future studies to build upon.

Prior studies have shown that VTA photoactivation of the total population of VTA[GABA] neurons is aversive[21,32]. We further extended these observations by demonstrating that VTA activation of VTA[GABA-only] neurons induced aversion, active avoidance, and is a strong negative reinforcer given that mice learned an instrumental response to stop such activation. We did not observe conditioned place aversion following VTA activation of VTA[GABA-only] neurons, as previously reported for photoactivation of the entire population of VTA[GABA] neurons targeted in GAD-Cre mice[21]. This apparent discrepancy is likely to reflect

differences in the photostimulation protocol employed, as well as the used of different transgenic mouse lines. Given that many VTA-GAD neurons lack VGaT, which is required for the vesicular accumulation of GABA[33], this raises the possibility that variability of behavioral outcomes may depend on the transgenic mouse lines tested (GAD-Cre vs VGaT-Cre mice). Our findings were strengthened by our observations that photoinhibition of VTA[GABA] neurons is rewarding and induces conditioned place preference. The aversive effects observed likely result from the direct inhibition of VTA[dopamine] neurons[19,21,34], as suggested by our connectivity studies and by previous work showing that VTA[dopamine] neurons receive GABAergic innervation from local neurons[23]. Nonetheless, the contribution of a multisynaptic mechanism in which VTA[GABA-only] neurons inhibit excitatory inputs from VTA[glutamate-only] neurons synapsing onto VTA[dopamine] neurons cannot be ruled out, especially in the light of our connectivity studies showing that VTA[GABA-only] neurons synapse on around 25% of neurons that are not GABAergic or dopaminergic. Given that, to date, activation of VTA[GABA] projections at terminal sites, such as the nucleus accumbens, the dorsomedial striatum or the lateral hypothalamus, has not been shown to be aversive[8,10,14,15], the aversion observed after activation of VTA[GABA-only] neurons in this study is best explained by a local microcircuitry mechanism. These findings reinforce the idea that the behaviors observed are governed by local interactions within the VTA, rather than long-range projections.

We showed that VTA[GABA-only] neurons play a critical role in locomotion, with their activation inducing decreases in speed and distance travelled and their inhibition inducing an increase on those responses. These results are in line with the notion that VTA[GABA] neurons participate in the inhibition of behavior by regulating neural circuits involved in reward processing and decision-making, specifically those mediated by VTA[dopamine] neurons. As with the aversion and avoidance findings, the locomotor effects observed after stimulation or inhibition of VTA[GABA-only] neurons are probably induced by local inhibition or disinhibition of VTA[dopamine] neurons based on our findings and on recent observations showing that inactivation of local VTA[GABA] neurons induced hyperlocomotion[2].

By in vivo recordings, we previously demonstrated that the activity of VTA[glutamate-only] neurons increases in response to sucrose delivery, foot shock or cues predicting these stimuli[5]. In addition, we and others have previously demonstrated that VTA photoactivation of the total population of VTA[glutamate] neurons is rewarding[18,24]. Here, we further extended these observations by showing that activation of local VTA[glutamate-only] neurons was rewarding and reinforcing, decreased feeding behavior, disrupted learning of an instrumental task to obtain food reward, and blocked reinstatement of food-seeking behavior in response to food priming- and cues associated with food delivery. In addition, we showed that chemogenetic inhibition of VTA[glutamate-only] neurons or local photoinhibition of VTA[glutamate] neurons is aversive and increased feeding behavior. While these observations suggest that local activation of VTA[glutamate-only] neurons play complex and multiple roles in regulating different behaviors, in vivo recordings studies indicate that different VTA[glutamate] neurons respond to different stimuli[4,6,7]. By in vivo electrophysiological recordings of the total population of VTA[glutamate] neurons, we found that most of these neurons increase their activity in response to an aversive air puff but some of them increase their activity in response to reward or remain unresponsive[4], and others increase their activity when mice approach threatening stimuli[6,7]. While collectively these results indicate heterogeneity among VTA[glutamate] neurons at the level of behavior, we had previously identified shared electrophysiological properties between VTA[glutamate-only] and VTA[glutamate-GABA] neurons[13], with similar heterogeneity and shared properties also identified by in vivo recordings of different VTA[glutamate] subpopulations containing or lacking GABAergic or dopaminergic markers. Indeed, the subpopulations studied varied in the phasic magnitude and sustained activity profiles of their responses to

consummatory rewards, preferences between rewards and aversive stimuli[30]. At the behavioral level, previous studies have led to the suggestion that mice regulate the time that they spent in a photostimulation-paired chamber within a two-chamber apparatus, frequently "shuttling" between compartments to receive brief stimulations. Results from these studies also led to infer that mice preferred shorter trains of photostimulation targeting the entire population of VTA[glutamate] neurons over longer ones[18]. While we also observed "shuttling" behavior in our experiments, photostimulation of VTA[glutamate-only] neurons led to a preference for the photostimulation-paired chamber, which is consistent with our previous findings showing that stimulation of the entire population of VTA[glutamate] neurons also induced preference for the photostimulation-paired chamber[24] but contrasts with previous findings[18] in which stimulation of the entire population of VTA[glutamate] neurons induced avoidance of the photostimulation-paired chamber[18]. A resolution of this apparent discrepancy is not possible at this stage, as differences in experimental conditions between our study and previous work[18] preclude a direct comparison. In addition, we did not observe differences in the operant responses to obtain photostimulation of VTA[glutamate-only] neurons at varying frequencies or durations, which may result from a ceiling effect or depolarization block. Thus, converging evidence indicates a great heterogeneity among VTA[glutamate] neurons, which is reflected, in part, in their electrophysiological properties and their participation in different behaviors.

Regarding the microcircuitry established by VTA[glutamate-only] neurons, we found that these neurons formed monosynaptic excitatory connections at similar frequency with both VTA[dopamine] and VTA[non-dopamine] neurons, and they also synapsed on both VTA[GABA-only] and VTA[glutamate-GABA] neurons. These findings suggest that VTA[glutamate-only] neurons have a broader intrinsic regulatory effect on different VTA neurons than VTA[GABA-only] neurons, whose connectivity appears more restricted. Thus, contrary to the widely accepted notion suggesting that all excitatory regulation on VTA[dopamine] neurons is mediated by long range excitatory inputs synapsing on them, we demonstrated the presence of a distinct VTA excitatory microcircuit originated by local glutamatergic neurons that modulates reward and motivated behaviors. These findings redefine the understanding of VTA functionality as being not solely dependent on afferent inputs but also intrinsically governed by local neuronal interactions.

Activation of VTA[glutamate-only] neurons reduced anxiety-like behavior, as evidenced by increased time in the open arms of the elevated plus maze and a trend toward greater center occupancy in the open field, supporting a role for these neurons in regulating both approach-avoidance conflict and threat-induced avoidance. In contrast, activation of VTA[GABA-only] neurons suppressed locomotor activity without impairing feeding- or context-specific motor behaviors and reduced anxiety-like responses in the open field but not the elevated plus maze. This pattern suggests that VTA[GABA-only] neurons may selectively influence anxiety-related responses tied to generalized arousal or avoidance, or alternatively, modulate motivation or arm discrimination in the elevated plus maze, possibilities that merit further investigation. Inhibition experiments further distinguished these populations: silencing VTA[glutamate-only] or VTA[glutamate] neurons had no significant behavioral effect on anxiety-like behaviors or locomotion, whereas inhibition of VTA[GABA-only] neurons increased locomotion without altering anxiety measures, data that underscore the behavioral specificity of the VTA microcircuits. Notably, both neuron types affected behavior in the open field, a test more sensitive to general arousal and exploration, highlighting that shared behavioral outcomes may arise through distinct neural pathways. Overall, these findings reveal complementary yet distinct contributions of VTA[glutamate-only] and VTA[GABA-only] neurons to anxiety regulation and motor control, advancing our understanding of VTA microcircuit specialization.

Although brain areas targeted by VTA[dopamine] neurons have been widely characterized[35,36], the outputs of the different VTA non-dopaminergic neurons are far less studied. Behavioral studies aimed at understanding the VTA neuronal circuitry participating in different aspects of behavior have demonstrated differential behaviors induced by local activation of VTA neurons versus activation of their axons outside the VTA. For instance, while VTA activation of either the total population of VTA[glutamate] neurons[18,24] or specifically VTA[glutamate-only] neurons is rewarding, we have found that release of glutamate from VTA neurons at terminal sites, such as the nucleus accumbens[17] or the lateral habenula[37], induces avoidance and aversion in place conditioning and operant paradigms. Similarly, VTA activation of VTA[GABA 21,32] or VTA[GABA-only] neurons promotes active avoidance and aversion but activation of long-range projections at specific terminal sites, such as the nucleus accumbens, the dorsomedial striatum or the lateral hypothalamus, is not aversive[8,10,14,15], with some reports suggesting promotion of reward instead, in the case of projections to the ventral nucleus accumbens and ventral pallidum[11,38]. Thus, detailed brain mapping of synapses from the diverse VTA interneurons or VTA long-range neurons is necessary to better address the behavioral participation of VTA interneurons (forming a VTA microcircuitry) versus VTA long-range neurons.

Several studies have identified afferents to VTA[dopamine] and VTA[GABA] neurons[3,23,35,39,40] but further work is needed regarding mapping the afferents to specific VTA non-dopaminergic neurons, such as VTA[glutamate-only], VTA[GABA-only], and VTA[glutamate-GABA] neurons, and how the regulation provided by those afferents modulate diverse behavioral outputs. We recently described the first known excitatory input to the total population of VTA[glutamate] neurons arising from lateral hypothalamic glutamatergic neurons[6] and found that activation of this pathway is aversive and promotes innate defensive behaviors[6]. Similar findings were observed after VTA stimulation of basal forebrain inputs that targeted VTA[glutamate] neurons[41]. Given that local photostimulation of VTA[glutamate-only] neurons is rewarding, we conclude that the VTA[glutamate] neurons targeted by lateral hypothalamic or basal forebrain inputs are most likely to be long range projection neurons instead of VTA[glutamate] neurons belonging to the local microcircuitry. Nonetheless, a subset of VTA[glutamate-only] neurons may also receive innervation from lateral hypothalamic glutamatergic neurons since we recently demonstrated that activation of the LH-VTA glutamatergic pathway mediates decreases in food intake in response to threats[7]. Future work should explore the distinct contributions of VTA local microcircuitry versus long-range projections to diverse behavioral outputs in the same study. Mapping the afferents and efferents of these local circuits in greater detail will help delineate how intrinsic connectivity within the VTA contributes to specific behaviors, in contrast to the roles played by long-range VTA projections.

In summary, by combination of specific targeting of VTA neurons (glutamate-GABA, glutamate, or GABA neurons) and correlative light and electron microscopy, we identified a complex VTA microcircuitry among VTA[dopamine] and non-dopamine neurons. Our findings demonstrate that activation of the VTA local microcircuitry elicits behaviors that are distinct from those mediated by its long-range projections, as widely reported in the literature. This underscores the functional importance of intrinsic VTA connectivity in regulating behaviors and highlights how local circuits, rather than long-range projections, play a key role in reward, aversion, and motivated behaviors. Given the profound implications of dopamine system dysregulations in mental health, our results identifying alternative sources of regulation to VTA[dopamine] neurons are critical to pinpoint future targets to treat dopamine-dependent disorders.

## Methods

### Subjects

*vglut2-IRES-Cre* mice (JAX # 016963, The Jackson Laboratories, Bar Harbor, ME; on a mixed C57BL/6;FVB;129S6 genetic background) and *slc32a1-IRES2-FlpO-D* mice (*vgat-FlpO-D*, Jax # 031331, The Jackson

Laboratories, on a mixed 129S6/SvEvTac × C57BL/6NCrl genetic background, deposited[42]) were crossed to produce the male and female *vglut2-Cre/ vgat-Flp* mice (20–30 g) used in this study. *vglut2-IRES-Cre* and *th-2A-Flpo* mice (C57BL/6N-Thtm1Awar/Mmmh, in C57BL/6 J background from the Mutant Mouse Resource and Research Centers, Davis, CA) were crossed to produce the *vglut2-Cre/th-Flp* male and female mice (20–30 g) used in this study. *vglut2-Cre* mice (Slc17a6tm2(cre)Lowl/J, in C57BL/6 J background from The Jackson Laboratories) and *vgat-Cre* mice (Slc32a1tm2(cre)Lowl/J, in C57BL/6 J background from The Jackson Laboratories) were bred in the NIDA/IRP animal facility (20–30 g) and were used in photoinhibition experiments. Groups of 2–5 mice were housed in a temperature- and humidity-controlled vivarium (at a constant temperature of 23 °C and 35–55% humidity) under a 12 h light/dark cycle (lights on at 7:00 am) with *ad libitum* access to food and water. Mice (2–3 months old before any manipulation) were kept undisturbed for at least one week before the start of each experimental procedure and were handled and weighed daily to minimize handling stress during experiments, which were conducted in accordance with the *Guide for the Care and Use of Laboratory Animals* and approved by the Animal Care and Use Committee of the National Institute on Drug Abuse, Intramural Research Program (ASP 24-INRB-2). All the experiments were performed during the light phase of the diurnal cycle.

## Viral vector construction

For the construction of INTRSECT viral vectors expressing mCherry, we used plasmids and methods published[43]. Sequence analysis of the mCherry coding region was performed to identify candidate locations for the insertion of the artificial intron necessary for the INTRSECT functionality. Possible sites were ranked using a neural network (Net-Gene2-2.42[44,45]) to predict splice sites. We broke mCherry between amino acids K20 and V21, encoded by "AAGGtg", since this location was predicted to have a confidence of 1.0 (100%) when interrupted with the sequence corresponding to an INTRSECT intron. Amplicons encoding mCherry (M1-K20) and (V21-236) were used to replace the first and second exons of eYFP in pAAV-hSyn Con/Fon eYFP (a gift from Karl Deisseroth, Addgene plasmid 55650; RRID: Addgene_55650; Addgene, Watertown, MA) to produce pAAV-Syn1-Con/Fon-mCherry (pOTTC2153, Addgene 217506). This construct was then used as the backbone and template for the construction of pAAV-Syn1-Coff/Fon-mCherry (pOTTC2190, Addgene 217507), where the orientation of the first exon and splice donor has been inverted (anti-parallel orientation). pAAV-nEF-Con/Fon-hChR2(H134R)-mCherry (pOTTC2147, Addgene 202545), an intermediate step on the construction of the other vectors, was constructed by isolating the eYFP sequences in pAAV-nEF-Con/Fon-hChR2(H134R)-eYFP (a gift from Karl Deisseroth, Addgene plasmid # 55644; RRID: Addgene_55644) and replacing those elements with corresponding sequences from mCherry. We modified "exon 3" by removing the eYFP sequences bounded by SalI and AscI restriction sites and inserting a synthetic fragment encoding the C-terminal portion of mCherry (V21-K236). Then we modified the "exon2" by removing the hChR2-eYFP sequences bounded by SpeI and KpnI restriction sites and inserting a synthetic fragment encoding the C-terminal portion of hChR2 and the N-terminal portion of mCherry (M1-K20). pAAV-nEF-Con/Foff-hChR2(H134R)-mCherry (pOTTC2185, Addgene 202546) was constructed by amplifying the region flanked by the Flp recombination sites (encompassing the entire hChR2-mCherry coding region) using pOTTC2147 as a template and using it to replace the corresponding sequence bounded by the AscI and XbaI restriction sites in pOTTC2147, in an anti-parallel orientation. pAAV-nEF-Coff/Fon-hChR2(H134R)-mCherry (pOTTC2186, Addgene 202547) was constructed by amplifying the region flanked by the lox recombination sites (encompassing "exon2" of the Con/Fon-hChR2-mCherry and adjacent intron elements) using pOTTC2147 as a template and using it to replace the corresponding sequences by the SpeI and KpnI

restriction sites in pOTTC2147, in an anti-parallel orientation. All constructs were made using ligation-independent cloning (In-Fusion, Takara Bio USA, Inc., Mountain View, CA), and transformed into a recombination-deficient bacterial strain (NEB Stable competent cells, New England Biolabs, Ipswich, MA). Insert containing clones were confirmed by restriction digest analysis and DNA sequencing. Each viral vector was packaged as previously described[46] with minor modifications. HEK293 cells were transfected using calcium phosphate precipitation to deliver plasmids encoding the adenovirus helper genes and the adeno-associated replicase and capsid genes for AAV serotype 1 (pHelper and pXR1, respectively)[47], and a plasmid encoding the vector genome. Cell lysates were collected 40 h post-transfection, treated with SAN HQ nuclease (ArcticZymes Technologies ASA, Norway), and passed through an AVB columns (GE Healthcare, Silver Spring, MD) using a fast protein liquid chromatography (FPLC) machine (AKTA, Cytiva, Wilmington, DE). Purified viral particles were eluted from the column with sodium citrate (pH -3.0) and titered using digital PCR (QIAcuity, QIAGEN, Germantown, MD) using a probe-based assay that recognizes the WPRE sequence. pAAV-nEF-Con/Foff-Gi-mCherry and pAAV-nEF-Coff/Fon-Gi-mCherry DREADDs were gifts from Karl Deisseroth (Addgene plasmid # 177673; RRID: Addgene_177673 and Addgene plasmid # 177669; RRID: Addgene_177669, respectively) and were packaged by the NIDA Genetic Engineering and Viral Vector Core (NIDA GEVV Core, Baltimore, MD), as described above.

## Surgical procedures

Each mouse was anesthetized with isoflurane (1–4% induction; 1–2% maintenance) and secured to a stereotaxic frame where its skull was exposed and leveled. INTRSECT (INTronic Recombinase Sites Enabling Combinatorial Targeting) viral vectors encoding the light-sensitive protein ChR2 tethered to eYFP or eYFP alone under control of the human synapsin promoter were used (Stanford University Gene Vector and Virus Core, Stanford, CA). Briefly, 250 nl of AAV-DJ-hSyn-Con/Fon-ChR2-eYFP (corresponding Glu-GABA-ChR2-eYFP mice), AAV-DJ-hSyn-Con/Fon-eYFP (corresponding control Glu-GABA-eYFP mice), AAV-DJ-hSyn-Con/Foff-ChR2-eYFP (corresponding Glu-only-ChR2-eYFP mice), AAV-DJ-hSyn-Con/Foff-eYFP (corresponding control Glu-only-eYFP mice), AAV-DJ-hSyn-Coff/Fon-ChR2-eYFP (corresponding GABA-only-ChR2-eYFP mice), or AAV-DJ-hSyn-Coff/Fon-eYFP (corresponding control Coff-Fon-eYFP mice) were injected into the VTA (AP: −3.2, ML: ± 0.0 to 0.3, DV: −4.3)[48]. AAV-nEF-Con/Foff-ChR2-mCherry or AAV-nEF-Coff/Fon-ChR2-mCherry (Addgene) were injected into the VTA of mice for ex vivo electrophysiology. AAV1- EF1α-DIO-eNpHR3.0-eYFP (UNC Vector Core Facility, Chapel Hill, NC), AAV8-nEF-Con/Foff-2.0-NpHR3.3-eYFP, AAV8-nEF-Coff/Fon-NpHR3.3-eYFP (Addgene), AAV1-nEF-Con/Foff-Gi-mCherry, AAV1-Con/Foff-SYN1-mCherry, AAV1-nEF-Coff/Fon-Gi-mCherry, or AAV1-Coff/Fon-SYN1-mCherry (NIDA GEVV Core) were injected into the VTA of mice for the loss-of-function experiments (AP: −3.2, ML: ± 0.0 to 0.3, DV: −4.3). Injections were done at a flow rate of 100 nl/min, using a Micro 4 controller and Ultra-MicroPump (WPI Inc., Sarasota, FL), 5 µl syringes, and 33-gauge needles (Hamilton, Reno, NV). Needles were left in place for 5 min following injections to minimize diffusion and to prevent reflux. Additional cohorts of mice were injected with a cocktail of AAV-DJ-hSyn-Con/Foff-eYFP and pAAV-nEF-Coff/Fon-ChR2-mCherry (pOTTC2186, NIDA GEVV Core) viral vectors for the selective and concurrent targeting of VTA$^{glutamate-only}$ and VTA$^{GABA-only}$ neurons. For photostimulation of local VTA$^{glutamate-only}$, VTA$^{GABA-only}$, or VTA$^{glutamate-GABA}$ axons, and for photo-inhibition of local VTA$^{glutamate}$ and VTA$^{GABA}$ axons, one or two optic fibers (200 mm diameter, BFL37-200, Thorlabs, Newton, NJ) were implanted above the VTA (AP: −3.2, ML: ± 1.0 to 1.3 with 10° angle, DV: −4.0) and secured to the skull using stainless steel screws (#000-120×1/16; Fasteners and Metal products Corp, Waltham, MA) and dental cement. For microinjection studies, a guide cannula (22 gauge, PlasticsOne,

Roanoke, VA) was lowered at a 10° angle toward the VTA (AP: −3.2, ML: +1.0 to 1.3, DV: −3.3). Following surgery, mice were given the analgesic meloxicam (5 mg/kg; Covetrus, Dublin, OH) to prevent post-surgical pain or discomfort and recovered on a warm heating pad before being transferred back to the vivarium. Optic fibers were implanted 8 weeks after viral injections and thereafter mice recovered for at least 10 days before the beginning of any experimental manipulation.

### RNAscope in situ hybridization combined with immunolabeling

Eight weeks following viral injections mice were anesthetized with chloral hydrate (80 mg/ml, 0.5 ml/kg) and perfused transcardially with 4% (w/v) paraformaldehyde (PFA) in 0.1 M phosphate buffer (PB), pH 7.3. Brains were left in 4% PFA for 2 h and transferred to 18% sucrose in DEPC-treated PB overnight at 4 °C. For RNAscope in situ hybridization experiments, coronal serial cryosections (16 μm) were prepared to phenotype VTA$^{glutamate-only}$, VTA$^{GABA-only}$, and VTA$^{glutamate-GABA}$ neurons. Briefly, free-floating VTA sections were incubated for 2 h at 30 °C with mouse anti-GFP antibody (1:500, 632381, Takara Bio USA, Inc.) or a cocktail of mouse anti-GFP and rabbit anti-DsRed (1:500, 632496, Takara Bio USA, Inc., Mountain View, CA) antibodies in DEPC-treated phosphate buffer (PB) with 0.5% Triton X-100 supplemented with RNasin (Promega, Madison, WI). Sections were rinsed 3 ×10 min with DEPC-treated PB and incubated with secondary donkey anti-mouse Alexa Fluor 488 (1:100, 715-545-151, Jackson ImmunoResearch Laboratories Inc., West Grove, PA) or a cocktail of secondary donkey anti-mouse Alexa Fluor 488 and secondary donkey anti-rabbit Cy3 (1:100, 711-165-152, Jackson ImmunoResearch Laboratories Inc.) antibodies for 1 h at 30 °C. Sections were rinsed with DEPC-treated PB, mounted onto Fisher SuperFrost slides and dried overnight at 60 °C. The phenotyping of VTA neurons expressing cFos after photostimulation of VTA$^{glutamate-only}$ or VTA$^{GABA-only}$ fibers was done by combination of immunolabeling using rabbit anti-phospho-c-Fos antibody (1:400, MAB5348, Cell Signaling Technology, Inc., Danvers, MA), mouse anti-TH antibody (1:1000, MAB318 at 1:1000 dilution, Millipore, Burlington, MA) and the detection of VGluT2 and VGaT mRNAs by RNAscope. Midbrain coronal free-floating sections (16 μm) were rinsed with DEPC-treated PB, and incubated with a cocktail of anti-cFos and anti-TH antibodies for 2 h at 30 °C, rinsed and incubated with secondary donkey anti-rabbit Alexa Fluor 647 (1:100, 711-605-152, Jackson ImmunoResearch) and donkey anti-mouse Alexa Fluor 750 (1:100, ab175738, Abcam, Cambridge, MA) for 1 h at 30 °C. Sections were rinsed with DEPC-treated PB and then were mounted onto Fisher SuperFrost slides and dried overnight at 60 °C. Then, RNAscope in situ hybridization was performed according to the manufacturer's instructions. Briefly, sections were treated with heat and protease digestion followed by hybridization with a cocktail of probes for detection of transcripts encoding both VGluT2 (319171, Atto 550, Advanced Cell Diagnostics, Newark, CA or Opal Polaris 780, Akoya Biosciences, Marlborough, MA) and VGaT mRNAs (319191-C3, Atto 647, or TSA Vivid Fluorophore 650, Advanced Cell Diagnostics). Additional sections were hybridized with the bacterial gene DapB as a negative control, which did not exhibit fluorescent labeling. RNAscope in situ hybridization and immunolabeled sections were viewed and photographed with a Zeiss confocal microscope with Airyscan/CY7.5 (LSM880, Zeiss, White Plains, NY). Negative control hybridizations showed negligible fluorophore expression. Neurons were counted when the stained cell was at least 5 μm in diameter. Pictures were adjusted to match contrast and brightness by using Adobe Photoshop (Adobe Systems, San Jose, CA). The number of mice (n = 3/group) analyzed was based on previous studies in our lab using radioactive detection of VGluT2 mRNA from rat VTA neurons[16,49].

### Fluorescence microscopy

We used cohorts of mice injected with AAV-DJ hSyn-Con/Fon-eYFP, AAV-DJ hSyn-Con/Foff-eYFP, or AAV-DJ hSyn-Coff/Fon-eYFP (Stanford University Gene Vector and Virus Core) for the detection of eYFP and VGluT2 or VGaT proteins in axon terminals. To study the distribution of concurrently transfected VTA$^{glutamate-only}$, VTA$^{GABA-only}$, and VTA$^{glutamate-GABA}$ neurons in the VTA, mice were injected with a mixture of three INTRSECT2.0 vectors: AAV8-Ef1α-Con/Fon-mCherry, AAV8-Ef1α-Con/Foff-eYFP, and AAV8-Ef1α-Coff/Fon-BFP (gifts from Karl Deisseroth, Addgene). We used another group of mice injected with AAV-DJ-Ef1α-Con/Fon-ChR2-eYFP, AAV-DJ-Ef1α-Con/Foff-ChR2-eYFP, or AAV-DJ-Ef1α-Coff/Fon-ChR2-eYFP (Addgene) for the detection of VTA$^{glutamate-GABA}$, VTA$^{glutamate-only}$, and VTA$^{GABA-only}$ axon terminals and their distribution in the VTA. Mice were deeply anesthetized with chloral hydrate (35 mg per 100 g), and perfused transcardially with 4% (w/v) PFA with 0.15% (v/v) glutaraldehyde and 15% (v/v) picric acid in 0.1 M PB (pH 7.3). Brains were left in this fixative solution for 2 h at 4 °C; the solution was replaced with 2% PFA and left overnight at 4 °C. Brains were rinsed with PB and cut into coronal serial sections (40 μm thick) with a vibratome (VT1200, Leica, Nussloch, Germany). Coronal VTA sections were incubated for 1 h in PB supplemented with 4% BSA and 0.3% Triton X-100, and then incubated with a cocktail of mouse anti-synaptophysin antibody (1:2000, MABN1193; MilliporeSigma, Billerica, MA) + rabbit anti-VGaT antibody (1:500, MSFR106160, Nittobo Medical, Japan) + guinea pig anti-VGluT2 antibody (1:500, MSFR106290, Nittobo Medical), or mouse anti-tyrosine hydroxylase antibody (1:1000, AB318, MilliporeSigma) overnight at 4 °C. The specificity of primary antibodies was examined and validated through western blot analysis in a previous study[33]. After PB rinsing, sections were incubated in cocktails of fluorescent secondary antibodies (1:100, all raised in donkey; Jackson ImmunoResearch Laboratories Inc.): DyLight 405 anti-mouse (715-475-151) + Alexa Fluor 594 anti-rabbit (711-585-152) + Alexa Fluor 647 anti-guinea pig (706-605-148), or Alexa Fluor 647 anti-mouse (715-605-151). After rinsing, sections were mounted with Vectashield mounting medium (H-1000-10; Vector Laboratories, Burlingame, CA) on slides and were air-dried. Fluorescent images were collected with a Zeiss confocal microscope with Airyscan/CY7.5 (Zeiss). Low magnification images using tile scan were taken sequentially with different lasers with ×20 objectives. High magnification images using Airyscan detector were taken sequentially with different lasers with ×63 oil immersion objectives and z-axis stacks were collected. Imaris microscopy software (Bitplane Inc., South Windsor, CT) was used to analyze z-stacks of confocal images from 3 mice for each phenotype (n = 9, 4 males, 5 females) to obtain three-dimensional quantification of axon terminals expressing GFP, synaptophysin, VGluT2, or VGaT.

### Electron microscopy

We used cohorts of mice injected with AAV-DJ hSyn-Con/Foff-hChR2(H134R)-eYFP, or AAV-DJ hSyn-Coff/Fon-hChR2(H134R)-eYFP (Stanford University Gene Vector and Virus Core) for the detection of eYFP and VGluT2, or VGaT in axon terminals by electron microscopy. Mice were deeply anesthetized with chloral hydrate (35 mg per 100 g), and perfused transcardially with 4% (w/v) PFA with 0.15% (v/v) glutaraldehyde and 15% (v/v) picric acid in 0.1 M PB (pH 7.3). Brains were left in this fixative solution for 2 h at 4 °C; the solution was replaced with 2% PFA and left overnight at 4 °C. Brains were rinsed with PB and cut into coronal serial sections (40 μm thick) with a vibratome (VT1200, Leica). Vibratome tissue sections were rinsed with PB, incubated with 1% sodium borohydride in PB for 30 min to inactivate free aldehyde groups, rinsed in PB, and then incubated with blocking solution [1% normal goat serum (NGS), 4% BSA in PB supplemented with 0.02% saponin] for 30 min. Sections were then incubated with primary antibodies, as follows: rabbit anti-GFP antibody (1:2000, MSFR101900, Nittobo Medical) + guinea pig anti-VGluT2 antibody + mouse anti-tyrosine hydroxylase antibody (1:1000, AB318, MilliporeSigma), or rabbit anti-GFP antibody + guinea pig anti-VGaT antibody (1:500, MSFR106160, Nittobo Medical) + mouse anti-tyrosine hydroxylase antibody. All primary antibodies were diluted with 1% normal goat

serum (NGS), 4% BSA in PB supplemented with 0.02% saponin and incubations were for 24 h at 4 °C. Sections were rinsed and incubated overnight at 4 °C in the corresponding secondary antibodies: biotinylated goat-anti-rabbit (GFP detection) + anti-guinea pig-IgG Fab' fragment coupled to 1.4-nm gold (VGluT2 or VGaT detection, 1:100, 2055-1 ML, Nanoprobes Inc., Yaphank, NY) + anti-mouse-IgG coupled to 1.4-nm gold (TH detection, 1:100, 2001-1 ML, Nanoprobes Inc.). Sections were rinsed in PB, and then incubated in avidin-biotinylated horseradish peroxidase complex in PB for 2 h at room temperature. Sections were rinsed in PB and postfixed with 1.5% glutaraldehyde at room temperature for 10 min. Sections were rinsed again in PB and in double-distilled water, followed by silver enhancement of the gold particles with the Nanoprobe Silver Kit (2012, Nanoprobes Inc.) for 7 min at room temperature. Next, peroxidase activity was detected with 0.025% 3,3'-diaminobenzidine (DAB) and 0.003% $H_2O_2$ in PB for 5–10 min. Sections were rinsed with PB and fixed with 0.5% osmium tetroxide in PB for 25 min, washed in PB, followed by double distilled water wash, and then contrasted in freshly prepared 1% uranyl acetate for 35 min. Sections were dehydrated through a series of graded alcohols and with propylene oxide, and were then flat embedded in Durcupan ACM epoxy resin (14040, Electron Microscopy Sciences, Hatfield, PA). Resin-embedded sections were polymerized at 60 °C for 2 days, and sections of 60 nm were cut from the outer surface of the tissue with an ultramicrotome UC7 (Leica Microsystems, Deerfield, IL) using a diamond knife (Diatome, Hatfield, PA). The sections were collected on formvar-coated single slot grids and counterstained with Reynolds lead citrate to be examined and photographed using a Tecnai G2 12 transmission electron microscope (Fei Company, Hillsboro, OR) equipped with the OneView digital micrograph camera (Gatan, Pleasanton, CA).

### Ultrastructural analysis
Serial ultrathin sections of the VTA (bregma −2.92 mm to −3.64 mm) from Glu-only-ChR2-eYFP (n = 3, 2 males, 1 female) or GABA-only-ChR2-eYFP (n = 3, 2 males, 1 female) mice were analyzed. These mice were deeply anesthetized, perfused, and their brain tissue was processed for electron microscopy study as described above. Synaptic contacts were classified according to their morphology and immunolabel and photographed at a magnification of ×6800–13,000. The morphological criteria used for identification and classification of cellular components or type of synapse observed in these thin sections were as previously described[50]. Briefly, type I synapses, here referred as asymmetric synapses, were defined by the presence of contiguous synaptic vesicles in the presynaptic axon terminal and a thick postsynaptic density (PSD) greater than 40 nm. Type II synapses, here referred as symmetric synapses, were defined by the presence of contiguous synaptic vesicles in the presynaptic axon terminal and a thin PSD. Serial sections were obtained to determine the type of synapse. In the serial sections, a terminal containing more than 5 immunogold particles was considered as immunopositive. Pictures were adjusted to match contrast and brightness by using Adobe Photoshop (Adobe Systems). This experiment was successfully repeated three times. Electron microscopy and confocal analysis quantification were blinded.

### Correlative light and electron microscopy (CLEM)
*vglut2-IRES-Cre × vgat-FlpO* mice (n = 7, 4 males, 3 females) were double injected with AAV-DJ hSyn-Con/Foff-eYFP (Stanford University Gene Vector and Virus Core) and AAV1-Syn1-Coff/Fon-mCherry (NIDA Genetic Engineering and Viral Vector Core), or with AAV-DJ hSyn-Con/Foff-eYFP (Stanford University Gene Vector and Virus Core) and AAV1-Syn1-Con/Fon-mCherry (NIDA Genetic Engineering and Viral Vector Core) in the VTA. Mice were deeply anesthetized with chloral hydrate (35 mg per 100 g), and perfused transcardially with 4% (w/v) PFA with 0.15% (v/v) glutaraldehyde and 15% (v/v) picric acid in 0.1 M PB (pH 7.3). Brains were left in this fixative solution for 2 h at 4 °C; the solution was

replaced with 2% PFA and left overnight at 4 °C. Brains were rinsed with PB and cut into coronal serial sections (100 µm thick) with a vibratome (VT1200, Leica). Vibratome sliced brain sections were rinsed with PB, and then incubated with blocking solution [1% normal goat serum (NGS), 4% BSA in PB supplemented with 0.02% saponin] for 30 min. Sections were then incubated with primary antibodies as follows: rabbit anti-VGluT2 antibody (1:500, MSFR106310, Nittobo Medical) + guinea pig anti-VGaT antibody + mouse anti-tyrosine hydroxylase antibody). All primary antibodies were diluted with 1% normal goat serum (NGS), 4% BSA in PB supplemented with 0.02% saponin and incubations were for 24 h at 4 °C. Sections were rinsed and incubated overnight at 4 °C in the corresponding secondary antibodies: DyLight 405 donkey anti-guinea pig IgG (1:100, 706-475-1148, Jackson ImmunoResearch Laboratories Inc.) + Alexa Fluor 647 donkey anti-rabbit IgG (1:100, 711-605-152, Jackson ImmunoResearch Laboratories Inc.) + Cy7 anti-mouse IgG (1:100, ab194808, Abcam). Sections were rinsed in PB, and then mounted on glass slides, cover-slipped and imaged with a Zeiss confocal microscope with Airyscan/CY7.5 (Zeiss). Low magnification images using tile scan were taken sequentially with 5 different lasers including 405, 488, 594, 647, and Cy7 with ×20 objectives. High magnification images using Airyscan detector were taken sequentially with different lasers with ×63 oil immersion objectives and z-axis stacks were collected. After imaging the fluorescent signals, the same VTA sections were transferred to 2.5% glutaraldehyde in PB overnight at 4 °C. After being rinsed with PB and then with ice cold 0.15 M cacodylate buffer containing 2 mM calcium chloride 10 min, sections were stained in the following heavy metal solutions in the following order: (1) 5% potassium ferrocyanide-2.0% osmium tetroxide solution in cacodylate buffer on ice for 60 min and covered with foil to block the light; (2) 1% TCH in double distilled water for 20 min at room temperature; (3) 2% osmium tetroxide in double distilled water for 30 min. Sections were rinsed 4 times (10 min each time) with double distilled water after each staining. Sections were then rinsed with degassed water and transferred into 1% uranyl acetate in degassed water overnight at 4 °C. Next, sections were rinsed with double distilled water, transferred to a lead aspartate solution inside a 60 °C oven for 30 min, and then they were rinsed with double distilled water and dehydrated using ice-cold solutions of freshly prepared 30%, 50%, 70%, 90%, 100%, 100% ethanol (anhydrous) for 8 min in each solution. Sections were incubated in 1% phosphotungstic acid (PTA) in 100% ethanol on ice for one hour, then were rinsed with ice cold 100% ethanol 3 times for 10 min each time and were treated with ice cold 100% acetone for 10 min. Next, sections were transferred to 50% Durcupan ACM resin in acetone for 2 h on a rotator, and then transferred to 100% Durcupan resin, overnight at room temperature. Then, sections were transferred into freshly prepared 100% Durcupan resin for 2 h, where they were embedded. Resin-embedded sections were polymerized at 60 °C for 2 days. Serial sections of 60 nm were cut with an Automated Tape Collecting Ultramicrotome (ATUMtome, RMC Boeckeler, Tucson, AZ). Serial sections were cut on Kapton tape and were placed on wafers with double stick conductive tape. Then, sections were coated on the wafer with Leica ACE600 (Leica Microsystems). The carbon coating thickness was set at 5.5 nm. Serial electron microscopic images were collected using Atlas 5 software with Zeiss Sigma VP scanning electron microscope (SEM, Zeiss, White Plains, NY). The serial SEM images were aligned with the Amira software (Thermo Fisher Scientific, Waltham, MA). The serial confocal images and serial SEM images were correlated with the Imaris software (Oxford Instruments, Abingdon, United Kingdom) and then analyzed for 2D segmentation and 3D reconstruction movies with the Dragonfly software (Comet Technologies Canada Inc., Canada).

### Laser photostimulation and apparatus for behavioral studies
Photostimulation was administered with a frequency of 20 Hz, a pulse duration of 10 ms and an intensity of 8 mW at 473 nm wavelength, and

photoinhibition was administered continuously at 8–10 mW at 532 nm wavelength. A three-chamber apparatus (Anymaze, Stoelting, Wood Dale, IL) consisting of two chambers (20 × 18 × 35 cm) and a connecting chamber (20 × 10 × 35 cm) with distinct walls patterns and non-reflective floors was used for place conditioning tests. Both optical intracranial self-stimulation (oICSS) and oICSS avoidance were conducted in light- and sound-attenuating operant chambers (Med Associates, St. Albans, VT) equipped with metal grid floor, two operant response wheels (1.9 cm W × 5.7 cm diameter) installed on a sidewall, a house light, and a cue light situated above each of the two wheels. Wheel turns were monitored by Med-PC software (Med Associates); each quarter-turn of the designated response wheel resulted in the delivery of optical stimulation (1 s train of photostimulation) for oICSS, and in a 2-s time out from optical stimulation for oICSS avoidance. Free-feeding studies were conducted in clear acrylic chambers (34 × 25 × 19 cm) containing regular bedding. All food self-administration tests were conducted in operant chambers (Model ENV-307W-C, Med Associates) equipped with two retractable levers located at each side of a food port. One of the levers was selected as the reinforced lever for delivering the food reward and the other was selected as the non-reinforced lever. Active pressing on the reinforced lever resulted in the delivery of a 20 mg chocolate-flavored food pellet (F05301, Bio-Serv, Flemington, NJ), while pressing on the non-reinforced lever did not result in a pellet delivery. Chambers were made of aluminum and clear acrylic, had grid floors, and were housed in sound- and light-attenuating boxes equipped with fans to provide ventilation and ambient noise. A stimulus light, located above the reinforced lever, and a 75 dB burst of white noise were paired contingently with the delivery of the food pellet. Mice were habituated to the food pellets for three days in their home cage prior to self-administration session. Optical stimulation was delivered continuously during food self-administration training and during reinstatement sessions (see food self-administration studies below). An open field arena made of opaque acrylic walls and a non-reflective base plate (40 × 40 × 35 cm; AnyBox, Stoelting) was used to test mice locomotor activity. An elevated plus maze made of grey acrylic (arm length:35 cm, lane width: 5 cm, wall height: 15 cm, elevated 50 cm from the floor, Stoelting) was used to test anxiety-like behaviors. Fiber optic cables were attached via FC/PC connector to 473 nm lasers (Opto Engine LLC, Midvale, UT) for photostimulation. For all behavioral experiments, the position of the animal was monitored via an overhead closed-circuit camera interfaced with video tracking software (Anymaze, Stoelting).

### Place preference or avoidance induced by photostimulation

Using the three-chamber place conditioning apparatus, Glu-GABA-ChR2-eYFP (n = 8, 5 males, 3 females), Glu-GABA-eYFP (n = 8, 5 males, 3 females), Glu-only-ChR2-eYFP (n = 7, 5 males, 2 females), Glu-only-eYFP (n = 9, 5 males, 4 females), GABA-only-ChR2-eYFP (n = 12, 8 males, 4 females), and GABA-only-eYFP (n = 9, 3 males, 6 females) *vglut2-Cre/vgat-Flp* mice, Glu-TH-ChR2-eYFP (n = 8, 5 males, 3 females), Glu-TH-eYFP (n = 7, 4 males, 3 females) *vglut2-Cre/th-Flp* mice, Glu-Halo-eYFP (n = 11, 6 males, 5 females), Glu-eYFP (n = 8, 2 males, 6 females) *vglut2-Cre* mice, and GABA-Halo-eYFP (n = 8, 3 males, 5 females), GABA-eYFP (n = 8, 3 males, 5 females) *vgat-Cre* mice were tested to determine whether VTA photostimulation or photoinhibition resulted in place preference or place aversion. During habituation and pretest sessions, mice were connected to a fiber optic cable and were allowed to freely explore the entire apparatus for 15 min without laser administration. Mice that spent 80% or more of the total session time in one of the chambers during the prestest session were excluded from the study. During the four photostimulation or photoinhibition days, one chamber was randomly assigned as the laser-paired chamber (counterbalanced across all mice): entry to this chamber by the mouse triggered continuous trains of VTA photostimulation (10 ms, 8 mW, 20 Hz) or photoinhibition (continuously administered at 8–10 mW).

The photostimulation or photoinhibition remained on for as long as the mouse was within the chamber. Entry to the other chamber was without consequences. Each test lasted 30 min and was repeated for 4 days. Mice were tested 24 h after the last photostimulation or photoinhibition session for 15 min in the absence of stimulation to determine if they developed conditioned preference or avoidance for any of the chambers.

### Place preference or aversion induced by chemogenetic inhibition

Using the three-chamber place conditioning apparatus, Glu-only-Gi-mCherry (n = 10, 6 males, 4 females), and Glu-only-mCherry (n = 8, 3 males, 5 females) *vglut2-Cre/vgat-Flp* mice were tested to determine whether VTA chemogenetic inhibition resulted in place preference or place aversion. During habituation and pretest sessions, mice were allowed to freely explore the entire apparatus for 15 min. Mice that spent 80% or more of the total session time in one of the chambers during the prestest session were excluded from the study. During the eight conditioning days, mice received intra-VTA microinjections (one per day) of either JHU37160 (J60, 200 nl, 0.1 µg/µl, in aCSF, HB6261, Hello Bio, Princeton, NJ) or aCSF, 3 min before being confined for 30 min to one chamber that was randomly assigned as the J60- or aCSF-paired chamber, respectively (counterbalanced across all mice). Mice were tested 24 h after the last microinjection for 15 min in the absence of drugs to determine if they developed conditioned preference or avoidance for any of the chambers.

### Optical intracranial self-stimulation (oICSS)

Glu-only-ChR2-eYFP (n = 7, 4 males, 3 females), Glu-only-eYFP (n = 7, 4 males, 3 females), Glu-GABA-ChR2-eYFP (n = 11, 6 males, 5 females) and Glu-GABA-eYFP (n = 11, 5 males, 6 females) mice were connected to the fiber optic cable and laser and were placed in operant chambers equipped with two response wheels (left and right) for daily 30-min self-stimulation sessions. Mice were trained on a fixed ratio (FR1) schedule of reinforcement. Each session began with illumination of the house light, which remained on for the entire session. Quarter-turns on one wheel ("active" wheel) activated a cue light above the wheel and resulted in a 1 s train of photostimulation at 20 Hz (8 mW, 10 ms, 473 nm). Responses on the other ("inactive") wheel were without consequences. Total wheel turns in the active and inactive wheel were recorded. For the first 6 oICSS training sessions, the right wheel was designated as the "active wheel", and for the next 4 oICSS training sessions, the left wheel became the "active wheel" (reversal training). A different cohort of Glu-only-ChR2-eYFP mice (n = 7, 3 males, 4 females) was used in the parametric studies testing a range of stimulation frequencies or stimulation durations. Mice were first assessed for stable performance with 1-s trains of photostimulation at 20 Hz, and then they were tested in morning and afternoon sessions, in which they received either different stimulation frequencies (1, 5, 10, 20 and 40 Hz) or different stimulation durations (1 s, 2.5 s and 5 s). Each stimulation parameter was given twice to each mouse in either ascending or descending order to control for potential learning effects.

### Avoidance of oICSS

GABA-only-ChR2-eYFP (n = 11, 7 males, 4 females), and GABA-only-eYFP (n = 9, 4 males, 5 females) mice were trained in the light- and sound-attenuated operant chambers described for the oICSS experiments. Light stimulation (0.5 s on/off at 20 Hz) was delivered in daily sessions (30 min) and mice were able to stop light stimulation for 5 sec by turning the active wheel. Responses on the inactive wheel were without consequences. During the first 6 days of training, the right wheel was designated as the active wheel, and during the subsequent 4 days of training, the left wheel was designated as the active wheel (reversal training). A different cohort of GABA-only-ChR2-eYFP mice (n = 9, 5 males, 4 females) was used in the parametric studies testing a

range of stimulation frequencies or stimulation durations. Mice were first assessed for stable performance at stopping 20 Hz photo-stiomulation for 5 s, and then they were tested in morning and after-noon sessions, in which they received either different stimulation frequencies (1, 5, 10, 20 and 40 Hz) or different durations of laser off (1 s, 2.5 s, 5 s and 10 s). Each stimulation parameter was given twice to each mouse in either ascending or descending order to control for potential learning effects.

**Open field test**
Glu-only-ChR2-eYFP (n = 8, 5 males, 3 females), Glu-only-eYFP (n = 13, 7 males, 6 females), GABA-only-ChR2-eYFP (n = 20, 12 males, 8 females), GABA-only-eYFP (n = 11, 5 males, 6 females) *vglut2-Cre/vgat-Flp* mice, as well as Glu-Halo-eYFP (n = 11, 6 males, 5 females), Glu-eYFP (n = 8, 2 males, 6 females) *vglut2-Cre* mice, GABA-Halo-eYFP (n = 8, 3 males, 5 females), and GABA-eYFP (n = 8, 3 males, 5 females) *vgat-Cre* mice were tested in an open field arena for 15 min to evaluate locomotor activity and anxiety-like behavior. Mice were connected to the fiber optic cable and laser and were placed in the center of the arena. Three 5-min trials were consecutively conducted: before, during and after VTA photo-stimulation (10 ms, 10 mW, 20 Hz) or photoinhibition (continuously administered at 8–10 mW). Total distance travelled (in meters), aver-age speed (in m/s) and time spent in the center, or the periphery of the arena were recorded for each of the 5-min trials. Glu-only-Gi-mCherry (n = 10, 6 males, 4 females), and Glu-only-mCherry (n = 8, 3 males, 5 females) *vglut2-Cre/vgat-Flp* mice were also tested in the open field arena for 5 min over 2 consecutive days. They receive intra-VTA microinjections of aCSF or J60 (counterbalanced across mice and days), 3 min before the start of the test, with total distance travelled, average speed and time spent in the center, or the periphery of the arena being recorded.

**Elevated plus maze test**
Glu-only-ChR2-eYFP (n = 20, 10 males, 10 females), Glu-only-eYFP (n = 10, 7 males, 3 females), GABA-only-ChR2-eYFP (n = 11, 7 males, 4 females), GABA-only-eYFP (n = 11, 5 males, 6 females) *vglut2-Cre/vgat-Flp* mice, as well as Glu-Halo-eYFP (n = 11, 6 males, 5 females), Glu-eYFP (n = 8, 2 males, 6 females) *vglut2-Cre* mice, GABA-Halo-eYFP (n = 8, 3 males, 5 females), and GABA-eYFP (n = 8, 3 males, 5 females) *vgat-Cre* mice were tested in an elevated plus maze for 15 min to evaluate anxiety-like behaviors. Mice were connected to the fiber optic cable and laser and were placed in the center of the maze. Three 5-min trials were consecutively conducted: before, during and after VTA photo-stimulation (10 ms, 10 mW, 20 Hz) or photoinhibition (continuously administered at 8–10 mW). Total time (in seconds) spent in the open and closed arms of the maze was recorded for each of the 5-min trials.

**Free-feeding studies**
Glu-only-ChR2-eYFP (n = 10, 5 males, 5 females), Glu-only-eYFP (n = 10, 7 males, 3 females), GABA-only-ChR2-eYFP (n = 10, 7 males, 3 females), GABA-only-eYFP (n = 9, 4 males, 5 females) *vglut2-Cre/vgat-Flp* mice, as well as Glu-Halo-eYFP (n = 11, 6 males, 5 females), Glu-eYFP (n = 8, 2 males, 6 females) *vglut2-Cre* mice, GABA-Halo-eYFP (n = 8, 3 males, 5 females), and GABA-eYFP (n = 8, 3 males, 5 females) *vgat-Cre* mice were food-restricted at 90% of their free-feeding weight and were placed in a chamber with a pre-weighed (≈1 g) amount of food (20 mg chocolate-flavored pellets, F05301, Bio-Serv) in a ceramic bowl (3.8 cm H × 7.6 cm diam.) for 3 days to habituate them to the environment and the testing procedure. Each session lasted 6 min and was divided into two 3-min trials. Food was weighed after every trial. On day 4, mice were con-nected to the fiber optic cable, but VTA photostimulation or photo-inhibition remained off. On day 5, mice were connected to the fiber optic cable and VTA photostimulation (10 ms, 10 mW, 20 Hz) or pho-toinhibition (continuously administered at 8–10 mW) were given

during the first 3-min trial of the session. No photostimulation or photoinhibition were administered during the second 3-min trial. The results were expressed as the difference between day 5 and day 4. Latency to start eating and the amount of food eaten were recorded for each experimental trial. Glu-only-Gi-mCherry (n = 10, 6 males, 4 females), and Glu-only-mCherry (n = 8, 3 males, 5 females) *vglut2-Cre/vgat-Flp* mice were food-restricted and habituated to the testing environment and procedure for 3 days, as described above. Then, they receive intra-VTA microinjections of aCSF or J60 (counterbalanced across mice and days), 3 min before the start of the feeding test, which lasted 5 min and in which they were presented with a pre-weighed (≈1 g) amount of food (20 mg chocolate-flavored pellets). The latency to start eating and the amount of food eaten were also recorded for each day, and the results were presented as the difference between the day in which mice received the microinjection of J60 and they day in which they received the aCSF microinjection.

**Food self-administration studies**
*Training sessions*. Glu-only-ChR2-eYFP (n = 10, 5 males, 5 females), Glu-only-eYFP (n = 11, 7 males, 4 females), GABA-only-ChR2-eYFP (n = 11, 7 males, 4 females), and GABA-only-eYFP (n = 10, 5 males, 5 females) mice were food-restricted at 90% of their free-feeding weight and were placed daily in operant chambers for food self-administration sessions that lasted 30 min and were conducted 7 days per week. Mice were connected to the fiber optic cable and laser, and VTA photostimulation (10 ms, 10 mW, 20 Hz, 5 s on/off) was delivered throughout the entire session. Each training session started with the presentation of both the reinforced and non-reinforced levers. The house light was on during the entire session. Presses on the reinforced lever resulted in a food pellet delivery concomitant with the presentation of a burst of white noise and the light cue (located above the reinforced lever) for 3 s. Mice were trained to lever press for food pellets under a fixed ratio 1 (FR1) schedule of reinforcement. A new food pellet could only be obtained after consumption of the previous food pellet. Consumption of the pellet in the food port was determined by head entry detectors (Med Associates). Lever presses on the reinforced lever while a food pellet was present in the food port were recorded but did not result in the delivery of a new food pellet. All the responses performed on the reinforced and non-reinforced levers (rewarded and non-rewarded) were recorded. The training sessions were conducted over 15 days. *Extinction sessions*. During the extinction sessions, presses on the reinforced lever did not result in a food pellet delivery. The light cue, white noise and VTA photostimulation were off during the extinction sessions that lasted 30 min and were conducted 7 days/week for 15 days. After this period, the number of responses on the reinforced lever was equal or <30% of the responses performed during the last day of the training sessions, and the reinstatement sessions started. Additional extinction sessions were run between reinstatement ses-sions. *Reinstatement sessions*. We tested three different experimental conditions to induce reinstatement of food-seeking behavior: a session with VTA photostimulation (10 ms, 10 mW, 20 Hz, 5 s on/off) admi-nistered for 5 min before the start of the extinction session; a session with cues and food priming, in the presence of VTA photostimulation (10 ms, 10 mW, 20 Hz, 5 s on/off) during the experimental session, in which lever presses on the reinforced lever resulted in the pre-sentation of the cues (light and burst of white noise) for 3 s but did not result in food pellets delivery; and a session with cues and food priming in the absence of VTA photostimulation, in which lever presses on the reinforced lever resulted in the presentation of the cues (light and burst of white noise) for 3 s but did not result in food pellets delivery. For both food priming sessions, 20 food pellets were placed in the food port immediately before the start of the session by the experimenter. The order of the sessions was counterbalanced across the mice and lasted 30 min.

## cFos induction

Glu-only-ChR2-eYFP (n = 3, 2 males, 1 female), Glu-only-eYFP (n = 3, 1 male, 2 females), GABA-only-ChR2-eYFP (n = 3, 2 males, 1 female) and GABA-only-eYFP (n = 3, 2 males, 1 female) mice were tested in their home cage. They were connected to the fiber optic cable and VTA photostimulation (10 ms, 8 mW, 20 Hz) was administered for 10 min. Mice were perfused 2 h later and brain tissue was dissected for RNAscope and immunohistochemical studies. Given that RNAscope procedures damage endogenous eYFP, rendering this fluorescent tag undetectable in VTA tissue following RNAscope processing, we first performed immunostaining with anti-GFP antibody on VTA sections. Once eYFP expression was confirmed, we used adjacent tissue sections from the same mouse for RNAscope analysis.

## Ex vivo electrophysiology

Six weeks after virus injection into the VTA, Glu-only-ChR2-mCherry and Glu-only-eYFP mice (n = 15, 5 males, 10 females) or GABA-only-ChR2-mCherry and GABA-only-eYFP mice (n = 12, 7 males, 5 females) were deeply anesthetized with isoflurane, were decapitated and their brains were quickly removed into oxygenated (95% O2/5% CO2), ice-cold high sucrose-based cutting solution (in mM): 220 sucrose, 2.5 KCl, 0.5 $CaCl_2$, 7 $MgSO_4$, 1.25 $NaH_2PO_4$, 26 $NaHCO_3$, 20 glucose (pH 7.2–7.4). Coronal slices containing the VTA (220 μm) were cut using a vibratome (VT1200, Leica), which were transferred into an oxygenated N-Methyl-D-glucamine (NMDG)-based recovery solution at 33 °C for 7 min (in mM): 93 NMDG, 3 KCl, 10 $MgSO_4$, 0.5 $CaCl_2$, 1.2 $NaH_2PO_4$, 30 $NaHCO_3$, 25 glucose, 20 HEPES, 5 sodium ascorbate, 3 sodium pyruvate (pH 7.3–7.4, -310 mOsm$^{-1}$). After recovery, slices were incubated in oxygenated aCSF at room temperature (in mM): 126 NaCl, 2.5 KCl, 2.4 $CaCl_2$, 1.4 $NaH_2PO_4$, 25 $NaHCO_3$, 11 glucose, $MgCl_2$ (pH 7.3–7.4, -320 mOsm$^{-1}$). For electrophysiological recordings, slices were transferred to a recording chamber continuously perfused with fully oxygenated aCSF at 33 °C. Patch pipettes (4–6 MΩ) were pulled from filamented borosilicate glass capillaries (World Precision Instruments, Sarasota, FL) with a PC-100 micropipette puller (Narishige, Tokyo, Japan) and backfilled with an internal solution containing (in mM): 140 potassium gluconate, 2 NaCl, 1.5 $MgCl_2$, 10 HEPES, 4 Mg-ATP, 0.3 $Na_2$-GTP, 10 Tris-phosphocreatine, 0.1 ethylene glycol-bis (2-aminoethyl ether)-N,N,N′,N′-tetraacetic acid (EGTA) with 0.08–0.1 % biocytin (pH 7.2, 280–290 mOsm$^{-1}$). Cells were visualized on an upright microscope using infrared differential interference contrast video microscopy. Whole-cell voltage-clamp recordings were made using a MultiClamp 700B amplifier (Molecular Devices, Sunnyvale, CA), low-pass filtered at 2 kHz and digitized at 10 kHz with pClamp 11.2 software (Molecular Devices). To measure EPSCs (held at − 60 mV) or IPSCs (held at 0 mV) in VTA$^{dopamine}$ neurons and VTA$^{non-dopamine}$ neurons, VTA neurons surrounded by mCherry-expressing fibers arising from VTA$^{glutamate-only}$ and VTA$^{GABA-only}$ neurons, without eYFP expression, were selected. The optical-evoked EPSCs or IPSCs were obtained with single-pulses of 473 nm wavelength blue light (5 ms). To verify monosynaptic connections, TTX (0.5 μM) followed by 4-aminopyridine (4-AP, 200 μM) was bath-applied after evoking EPSCs or IPSCs. To determine whether EPSCs were evoked by AMPA receptors, CNQX (10 μM) was successively applied. Bicuculline (10 μM) was added to aCSF during recordings of EPSCs to avoid synaptic inhibition induced by GABA$_A$ receptors. The synaptic properties of IPSCs were examined by bath-application of bicuculline (10 μM) after administration of TTX and 4-AP. Current amplitudes were measured by differences of peak amplitudes from baselines. To measure firing activity evoked by photostimulation of axon terminals from VTA$^{glutamate-only}$ or VTA$^{GABA-only}$ neurons, 10 or 20 pulses of blue light were applied with different stimulation frequencies (10, 20, 40, and 100 Hz for VTA$^{glutamate-only}$ fibers or 1, 5, 10, and 20 Hz for VTA$^{GABA-only}$ fibers) for different stimulation durations (1, 5, 7, and 11 ms). To test for the selective expression and function of the INTRSECT inhibitory DREADD viral vector in VTA$^{glutamate-only}$ or VTA$^{GABA-only}$ neurons,

we injected depolarizing currents (600 ms duration) in mCherry-positive VTA neurons and measured the evoked spiking responses before and after application of J60 (10 μM). After recordings, slices were fixed with 4% PFA, overnight at 4 °C. Then, they were rinsed with PB and were incubated with mouse anti-tyrosine hydroxylase antibody (1:500, MAB318, MilliporeSigma) overnight at 4 °C. After being rinsed in PB, slices were incubated with DyLight 405 donkey anti-mouse (1:100; 715-475-151, Jackson ImmunoResearch Laboratories Inc.) and Cy5 Streptavidin (1:100; 016-170-084, Jackson ImmunoResearch Laboratories Inc.) overnight at 4 °C. mCherry and eYFP labeling were observed by their endogenous fluorescence. Images were taken with a Zeiss confocal microscope with Airyscan/CY7.5 (Zeiss).

## Quantification, statistical analyses and reproducibility

Behavioral data were analyzed using two- or three-way ANOVAs with group (eYFP vs ChR2-eYFP) as the between-subjects factor, and sessions, phases of testing, etc., as within-subject factors, in the case where more than two variables were to be compared. When the same mice were tested under different conditions, a repeated-measures ANOVA was used instead. For significant overall interactions, further analyses of partial interactions were carried out. Post hoc analyses were performed using the Tukey HSD test when the initial p-value was significant. A result was considered significant if p < 0.05. All data were analyzed using Statistica software (Cloud Software Group Inc., Fort Lauderdale, FL). All the experiments were successfully repeated three times to ensure reproducibility of the results.

## Reporting summary

Further information on research design is available in the Nature Portfolio Reporting Summary linked to this article.

## Data availability

The source data generated in this study have been deposited in the Zenodo repository under https://doi.org/10.5281/zenodo.15765068 (https://doi.org/10.5281/zenodo.15765069). Source Data are provided with this paper[51].

## Code availability

No unreported custom computer code or algorithm was used to generate the results that are described in this study.

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

## Acknowledgements

This research was supported by the Intramural Research Program of the National Institute on Drug Abuse, National Institutes of Health (NIH). The contributions of the NIH authors are considered Works of the United States Government. The findings and conclusions presented in this paper are those of the authors and do not necessarily reflect the views of the NIH or the U.S. Department of Health and Human Services. Some plasmids and AAV vectors were produced by the National Institute on Drug Abuse Genetic Engineering and Viral Vector Core (RRID: SCR_022969). Some of the histological material was processed by the National Institute on Drug Abuse Histology and Imaging Core. We thank Rucha Kulkarni for image analysis assistance.

## Author contributions

M.M., M.F.B., H.W. and S.Z. conceptualized the project. MFB and HW performed behavioral and pharmacological studies, and data analysis. H.W., S.Z. and B.L. performed neuroanatomical studies. S.Z., A.V.S., K.J.Y. and R.Y. performed anatomical and ultrastructural studies and data analysis. S.Z., K.J.Y. and A.V.S. performed CLEM. S.H. performed ex vivo recordings and data analysis. C.T.R. prepared custom viral vectors. M.M. and M.F.B. prepared the manuscript with contributions from all co-authors.

## Funding

## Competing interests

The authors declare no competing interests.
