## [Transparent Peer Review file · Nature Communications]

VTA monosynaptic connections by local glutamate and GABA neurons and their distinct roles in behavior

Corresponding Author: Dr Marisela Morales

Version 0:

Reviewer comments:

Reviewer #1

(Remarks to the Author)

VTA neurons show remarkable heterogeneity and recently have been shown to have subpopulations expressing only GABA, only glutamate, or co-expressing both GABA and glutamate. In this manuscript, Barbano and colleagues explore the roles of these specific subpopulations in behavior using the INTERSECT viral approach. The authors show that optogenetic stimulation of VTA glutamate-only neurons is reinforcing, yet decreases obtaining rewards and slows appetitive operant learning. By contrast, VTA GABA-only stimulation is aversive, yet does not impair reward consumption, blocks appetitive operant learning and decreases locomotion. Finally, dual GABA/glutamate neuron stimulation is largely inert, but delays appetitive operant learning. Additionally, they report that glutamate-only cells synapse onto dopamine and glutamate cells, while GABA neurons target mostly dopamine cells. While the viral approach to dissecting neural subpopulations and microcircuit tracings are elegant and despite presenting some new interesting data, the study falls short of integrating the behavioral observations with the connectivity data into a cohesive story.

Major comments:

The optogenetic stimulation experiments do not yield a consistent picture. Broadly, glutamate only neurons are presented as rewarding and anxiolytic and GABA only neurons are presented as aversive, yet stimulating glutamate neurons disrupts reward consumption and slows appetitive learning and vexingly, GABA stimulation does not disrupt reward consumption (even though multiple authors have previously reported this effect) but has a profound effect on preventing appetitive learning. How is it that GABA stimulation is aversive, yet time in the open arms is not affected (stress tends to decrease time in open arms). Why does stimulating glutamate or GABA neurons lead to real time preference or avoidance, but not conditioned place preference or avoidance (especially since dopamine stimulation and inhibition leads to CPP/CPA)? The authors do not give a satisfying explanation of these discrepancies.

Similarly, the authors have previously reported that GABA only and glutamate only neurons respond to both rewarding and aversive stimuli, along with cues that predict these outcomes, while dual GABA/glutamate neurons respond only to the stimuli and not to the predicting cues (Root et al 2020). It is unclear how the stimulation experiments in this manuscript builds on those findings. For example, one can come up with explanations for why consuming rewards evokes glutamate neuron activity but stimulating those neurons reduces reward consumption, but these explanations don't shed obvious light on the normal function of these neurons.

This problem is rendered more challenging by the fact that there is no evidence that the optogenetic stimulation delivered (20 Hz for anywhere from 0.5 seconds to 5 minutes depending on the experiment) resembles the natural activity patterns these neural populations would produce. While optogenetic stimulation experiments are always somewhat artificial, they can be useful in demonstrating the causal impact of mimicking naturally observed activity patterns. From Root et al, it would seem that reward activates glutamate only neurons for a shorter period than GABA or dual GABA/glutamate neurons but these dynamics were not taken into account. To attempt to understand what these neurons do by activating them requires recording the normal activity rates and temporal dynamics of the different populations and tailoring the stimulation accordingly.

By contrast, optogenetically inhibiting the neural activity of these different populations selectively during the different phases of behavioral tasks in which they are known to be active would provide some insight into what that neural activity contributes.

Characterizing the synaptic connectivity of VTA glutamate and GABA neurons is an important contribution to the field. However, it is unclear what these results imply for the behavioral experiments. Does the fact that glutamate neurons synapse on 25% of TH+ neurons explain why stimulating these neurons do not induce hyperactivity (while stimulating GABA neurons

which synapse on 65% of TH+ neurons induces hypoactivity)? Are any of the other discrepant behavioral results explained by the synaptic connections? Conversely, now that we know the connectivity, are there experiments to test the relationship between the microcircuitry and behavior?

Line 325:326. Why should a lack of dual GABA/glutamate synapses within the VTA explain the lack of behavioral responses? Don't these neurons synapse elsewhere?

Minor.

Line 119: reward is mediated.... Aversion is mediated should be "photostimulation induces ..."

The manuscript switches between referring to the neural populations by the INTERSECT strategy (Coff/Fon), the promoter (VGAT-only), or the neuron type (GABA-only). I would pick the latter and stick with it throughout.

Reviewer #2

(Remarks to the Author)

This is an interesting paper with extensive and useful anatomical data on local connectivity within the VTA. The authors are well aware of the heterogeneity of VTA neurons. In this case, using a clever cross-sectional molecular approach in mice they address differences in connectivity between VTA neurons that express GABA only, Glutamate only and GABA-Glutamate. A major anatomical finding is that VTAGlut-only neurons frequently establish synapses on VTAdopamine and VTAGlut-only neurons, and that VTAGABA-only neurons mostly synapse on VTAdopamine neurons. Importantly, VTAGlut-only neurons do not establish synapses on dual VTAdopamine-glut neurons, but do provide a major input to dual VTAGlut-GABA neurons. Their ultrastructural data is quite convincing and novel.

1) In ex-vivo recording they demonstrate convincingly that VTAGlut-only neurons monosynaptically connect with unidentified VTA neurons projecting to the nucleus accumbens. It's likely that the majority of the excited neurons were dopaminergic but a significant number could be non-dopaminergic (given their results, this omission is critical). Furthermore, the authors present no direct data to demonstrate that this projection produces the rewarding behavior of activation of the VTAGlutamate only population of neurons.

2) In addition to the anatomy, the authors undertook a variety of behavioral studies using channel rhodopsin stimulation of the various subpopulations of VTA neurons. Although, interesting differences in effect were observed when the different subpopulations were differentially activated those results are difficult to interpret, for a variety of reasons.

a) As the authors point out, within a given subgroup, neurons project to different target regions with different functions.

b) Simultaneous activation of an entire subpopulation by any perturbation is probably quite rare in intact mice under physiological conditions. So while showing that some members of the subpopulation are sufficient to produce a given effect they may not be necessary. To establish that, the authors would have to use an inhibitory opsin, ideally in specific axon terminal regions.

c) The authors did not demonstrate the effectiveness of their optogenetic stimulation, i.e. what per cent of the visualized neuron subtype ex vivo, was activated by the optogenetic stimulation. This might be particularly important for neurons that release more than one neurotransmitter, since different stimulation parameters can result in differential release of transmitter.

Reviewer #3

(Remarks to the Author)

The study by Barbano et al focuses on the contribution of different subclasses of VTA neurons and their roles in reward-seeking and learning. This is one of a handful of studies to examine VTA microcircuitry and uses cutting-edge CLEM imaging to define the local connectivity. The paper includes several very nicely run studies that help to differentiate the roles of separate VTA subpopulations and their connectivity. However, there is also a disconnect between the way the paper is framed and the limitations of the experimental designs employed in the paper.

Major

1. The framework of the paper focuses on local connectivity within the VTA and VTA microcircuitry. However, most of the experiments (e.g., the Real-time place conditioning) involve stimulating what is almost certainly a mixture of local and projection neurons. While I recognize that the field lacks experimental methods that would allow the authors to overcome this hurdle, this still creates a problem for the overall framing, and I felt the authors could be more forward about the limitations of the place conditioning and optical ICSS experiments. If the authors wish to focus on the microstructure, they might consider leading with the CLEM and anatomical experiments that describe the microstructure and then framing the remaining experiments accordingly.

2. Several experiments employ the Newman-Keuls post hoc test, which poorly corrects for the familywise error rate and may decrease the reliability of the observed findings. This post hoc is typically found in GraphPad, whose documentation indicates that it should not be used.

3. While male and female mice are both indicated in the methods, sex as a biological variable appears to be otherwise ignored and several experiments do not appear sufficiently powered to even detect sex differences, were they to exist.

4. The authors follow the real-time place conditioning experiments with optical ICSS confirmatory experiments. These experiments include both a stimulation-seeking and a stimulation-avoidance protocol, depending on the outcome of the

CPP. While it is clear why a stimulation-seeking protocol was used for the VTA glutamate-only neurons, and a stimulation-avoidance protocol was used for the VTA GABA-only neurons, it isn't clear why the choice was made to test VTA glutamate-GABA mice in a stimulation-seeking paradigm. While these mice showed no preference or aversion in the real-time place conditioning paradigm, they showed a bias towards stimulation avoidance, if anything. Repeating this experiment in the stimulation-avoidance protocol would be valuable.

5. The number of CreOn/FlpON mice included in the food self-administration experiments (Figure 5) is about half of that included for every other group and low enough to raise questions about the experimental power. With only 6 mice included, subtle differences in early operant behavior could contribute to the observed delays in learning. Did the authors conduct a power analysis to determine that these experiments were sufficiently powered? If not, it seems that these experiments should include a similar number of mice to those in the same figure.

6. The purpose of the final recording experiment is unclear. Is the mesoaccumbal pathway the only pathway where VTA VGluT2-only and VTA TH-only neurons project? It is simply not clear why this pathway was chosen above all of the other VTA-VGluT2 projections, and without that, this experiment is confusing and may even detract from the straightforward narrative the authors maintain up to this point.

Minor

1. While perhaps more technically accurate, the use of "Con/Off-ChR2-eYFP" when referencing the different stimulation groups on Page 4 and beyond decreases readability. The authors establish that the viral vectors are specific in the first section, after which I would encourage them to switch to the nomenclature "VTA VGluT2-Only", as they use in the introduction, and later in the electron microscopy sections.

2. For Figure 4 and the associated experiments, it is unclear what the differences in latency and the amount of food eaten are in comparison to. Is this the session prior?

3. On page 5, line 178, the authors conclude that "Termination of GABA release from VTA GABA-only neurons seems to facilitate feeding behavior in food-restricted mice". However, this statement is not quite accurate. The authors are not terminating GABA release here but rather terminating the stimulation of GABA-only neurons. These neurons would presumably return to their normal firing patterns following the end of stimulation.

Reviewer #4

(Remarks to the Author)

This manuscript by Barbano et al. uses an intersectional viral approach to isolate neurons in the VTA that are glutamatergic only, GABAergic only, or express both glutamatergic and GABAergic markers. They find that optogenetic stimulation of Glu-only neurons is appetitive, GABA-only neurons is aversive, and Glu/GABA neurons is largely neutral. They also find that activating Glu only neurons inhibits feeding, and activating Glu only or Glu/GABA neurons during an operant task (lever press for food) delays learning on the task, while activating GABA neurons strongly inhibits learning. Using EM, the authors quantify the synaptic connectivity between these populations and with dopamine neurons in the VTA.

Overall, this paper increases our understanding of the different populations of glutamate and GABA neurons in the VTA, and represents an important contribution to our knowledge of the complexity of VTA microcircuits. The EM work in particular is an impressive and elegant analysis of the connectivity of these cell types. However, I have concerns regarding the design of some of the optogenetic experiments, which makes it difficult to interpret those results and limits our understanding of the functional role of these neurons.

Major points:

1) It has been established that some glutamate, GABA, and Glu/GABA neurons also co-express dopamine neuron markers (i.e. Th and/or DAT; see PMID: 37243808, PMID: 35385745). The authors should acknowledge that some of the results they see with optogenetics could be due to release of dopamine from these populations.

2) This lab previously found that stimulation of all VTA Glu neurons (not just Glu only) could drive a place preference (PMID: 26631475). However, another group found that stimulation of all VTA Glu neurons (not just Glu only) was moderately aversive in a RTPP task, but that mice made repeated entries in and out of the paired chamber to stimulate brief activation of VTA Glu neurons (PMID: 27976722). To facilitate a better comparison to this previous study, it would be interesting to plot the number of entries made into the paired chamber for each group in Fig 2. The authors should also discuss what might account for the differences between these studies and between stimulation of all Glu neurons vs the smaller Glu only or Glu/GABA populations.

3) A major concern is with the design of the optogenetic stimulation during the operant task. The authors deliver 5s on/5s off stimulation during the entirety of the operant session. This makes it difficult to interpret what effect the different VTA populations have on the specific aspects of this task. For example, given that stimulation of Glu only neurons is reinforcing, it is not particularly surprising that giving the animals continual reinforcement delays them from learning to press a lever to

receive additional reinforcement (food pellet). And given that stimulation of GABA only neurons is aversive, it is not particularly surprising that animals in a highly aversive state would show impaired behavioral performance. But in neither case does this provide much insight into the specific role of these neurons in regulating motivated behavior. This could potentially be addressed with a study time-locking optogenetic stimulation of these neurons to specific behavioral events, or better yet (at least in the case of the Glu and Glu/GABA populations) using chemogenetic or optogenetic inhibition of these neurons to test their necessity in this task.

4) In supplementary Fig 1 the authors show example images of the different cell types in the VTA. If images are available, a more thorough description/analysis of the spatial distribution of these cells (and comparison to previous in situ work looking at the distribution of these cell types, PMID: 37243808) would be beneficial.

5) In supplementary Fig 3 there appear to be substantial differences at baseline (no light stim) between the YFP control group and the Chr2 group in some cases. Was the difference in the first light off period in C and D tested for significance? And in I and J, is there an explanation for why the Chr2 mice overall spend more time in the open arms even with no stimulation? The same question applies to Supplementary Fig 4 C-D.

6) It would be helpful to plot edge time during the open field assay (Supp. Fig. 4), as this is another measurement of the anxiety-like behavior seen in the elevated plus assay.

Minor Points

1) A schematic summarizing the connectivity patterns observed between different VTA cell types with the EM experiments would be very helpful.

2) The finding that VTA GABA neurons make few synaptic contacts with other VTA GABA neurons is confirmatory of a previous study (PMID: 32541962), which used local or distal knockout of Vgat to examine GABAergic connectivity in the VTA. This paper should be referenced.

3) In several places the authors specifically refer to identifying axon terminals contacting the soma of other cell types, without referencing synapses found on dendrites. Were somatic synapses the only type detected?

4) Both in figures and in the text, the use of "Con/Off-ChR2-eYFP mice" etc. to denote the different groups makes the paper somewhat difficult to follow. Authors should consider using a shorthand designation (i.e. "Glu-only mice") to make the paper easier to read.

Reviewer #5

(Remarks to the Author)

Version 1:

Reviewer comments:

Reviewer #1

(Remarks to the Author)

I appreciate that the authors have done a lot of experiments, including new ones in response to review, but I continue to have concerns that it is hard to draw collective conclusions about the role of these VTA subpopulations.

Major comments:

1. It is worth noting that the authors show that in their hands INTERSECT method has an error of at least 10-15% which limits the interpretability of findings.
2. One of the most striking findings is that when using Vglut-cre mice to target neurons in the VTA 65% of them will be dual Glu-GABA neurons and when using vGAT-cre mice, 57% of the neurons will be dual Glu-GABA. The authors do not explicitly comment on this finding which would seem to be an important consideration for the field in interpreting such experiments. Yet in their own inhibition experiments (supplemental figures 18-19), the authors essentially ignore that using vglut and vgat cre mice will respectively include these large subpopulations. Apparently, INTERSECT did not work for selectively inhibiting specific subpopulations (why?) but global inhibition does not answer the scientific question.
3. Another important finding is that combined Glu-GABA neurons do not form local synapses. However, when considering the behavioral consequences of exciting/inhibiting them, the authors seem to discount the possibility that they could exert influence by means of their distal projections. Moreover, they do not characterize where this major subpopulation project to or what their role in behavior is.
4. Similarly, the justification given for omitting dual GABA-Glu neurons in further optogenetic experiments is that optogenetic behavioral experiments are intended to understand the impact of local VTA. However, there is no reason to believe that GABA -only and Glu-only do not exert their impact on conditioned place, self-stimulation, or feeding via long-range actions.
5. The findings are not well integrated with prior literature or internally. For example, Tan et al found that VTA GABA stimulation leads to real time conditioned place avoidance and persistent avoidance the following day unlike the current

study. Similarly, there is no explanation of why VTA GABA-only neurons have divergent effects in the EPM and OF.

6. The conclusions drawn sometimes outpace the evidence. For example, in addition to “anxiety” changes in behavior during the EPM (a conflict task) could reflect changes in motivation or ability to discriminate between open and closed arms.

7. The authors report non-uniform distribution of the different subpopulations, but did not specify the bregma coordinate borders of “medial-lateral” or “lateral”. Could the absence of effect of optogenetic stimulation on dual Glu-GABA neurons reflect poor placement of the fiber relative to the population?

8. Similarly, given the non-uniform distribution of the subpopulations, can the authors be certain that the behavioral effects of GABA-only or Glu-only optogenetic stimulation did not differ based on the distribution of the fiber placements?

9. While the addition of subpopulation superscripts was a welcome addition, the manuscript remains very densely written and challenging to read.

Minor:

There is a typo on lines 221-222

Reviewer #2

(Remarks to the Author)

The authors have addressed my concerns.

Reviewer #3

(Remarks to the Author)

Overall, the authors addressed all of my points and seemed to adequately address those raised by other reviewers. A few minor edits or clarification points:

Minor Comment:

For supplementary figures 7 and 8, it would be nice to separate the c-Fos plots for the Glutamate (Figure S7) and GABA (Figure S8). Since these are the populations being stimulated, it seems that they should be depicted separately. Related to this, the authors don't indicate whether they counted eYFP-positive cells from the ChR2 or explicitly avoided these. This is important, because even using a 4-color imaging system, it seems that one of the RNAscope colors would have overlapped with the eYFP from the viral injections. Some clarification in the results and methods would be

Supplemental Figure 13- Title sentence says “glutamate” instead of “glutamate”

Reviewer #4

(Remarks to the Author)

The revised version of the manuscript by Barbano et al. makes some improvements on the original submission, including better organization starting with the EM data, and nice corroborating data with the Fos analysis. However, I feel that some of my concerns and those of other reviewers were not adequately addressed, particularly regarding the optogenetic behavioral experiments, and the results are not adequately discussed in the context of the previous literature.

1) 3 out of 4 reviewers commented that using the INTERSECT virus nomenclature (i.e. ConFoff) throughout the paper was confusing, and suggested that they instead refer to groups by the neuron type (i.e. Glu-only). The authors declined to do so for the behavioral studies, and the manuscript continues to suffer from poor readability due to this choice.

2) The authors now report entries into the paired chamber in the RTPP experiment, but fail to put these results in the context of the previous literature as requested. The authors do in fact see greatly elevated entries into the paired side (Sup Fig 10c), similar to what was reported by Yoo et al (2016) with stimulation of all VTA-Glu neurons. These increased entries are an interesting phenotype, as this is not what is typically seen with activation of dopamine neurons (mice will enter the paired side and spend more time there but will not repeatedly enter and exit). How do the authors interpret these results? Why might the all-Glu vs Glu-only results be different in terms of total time spent on the paired side? What might this mean for the role of these neurons in reward signaling?

Relatedly, the authors add data looking at ICSS with different frequency and duration stimulation (Sup. Fig 13). (Side note: The diagram in this figure appears to be mislabeled, showing vglut2-Cre/th-Flp instead of vgat Flp.) In contrast to Yoo et al, they do not see a frequency dependence for stimulation of VTA-Glu neurons. This is surprising but is not discussed. Yoo et al also found that mice preferred shorter stimulation of VTA-Glu neurons (1 or 5 s) relative to long duration stimulation (20 or 40 s). Unfortunately, the authors in this manuscript only test 1, 2.5, and 5 seconds, so it is impossible to know whether the Glu-only neurons act similarly to the all-Glu neurons tested by Yoo et al. (They likely are similar, given the high number of entries in the RTPP experiment, which indicates the mice are likely “shuttling” to turn the light on and off for brief durations.) Again, this is one of the most interesting phenotypes of these VTA-Glu neurons in the context of their role in reward learning, but unfortunately it is not discussed at all in the manuscript, and the experimental design did not take into account these previous findings.

3) The added experiments with inhibitory opsins are disappointing. While I appreciate the apparent technical issues with the INTERSECT inhibitory viruses, doing these experiments in the total VTA GABA and Glu populations is of limited benefit, and does not address the original concerns of myself and the other reviewers regarding the interpretation of the optogenetic results. Both Reviewer 1 and my comments suggested that time-locked manipulation of these neurons, particularly during

the operant task, would be a much better way to test their function, but this was not done either with stimulation or inhibition. As it stands, the operant task remains uninterpretable and in my opinion does not add anything of benefit to the paper.

4) A summary schematic of connectivity of the different cell types would still be helpful.

Reviewer #5

(Remarks to the Author)

Version 2:

Reviewer comments:

Reviewer #1

(Remarks to the Author)

The revised manuscript by Barbano et al. includes an impressive amount of work which is now clearer. Overall, the manuscript will be a substantive addition to the field. There are still some points that require clarification:

Major

1. While the authors clearly demonstrate the existence of local connections among VTA GABAergic and glutamatergic (Glu) subpopulations, the current framing of the results unintentionally implies that the effects observed with optogenetic and chemogenetic manipulations are mediated solely via these local connections. However, the authors have not provided evidence that these subpopulations lack distal projections. The discussion section nicely addresses the distal vs local distinction and prior evidence about the topic. Importantly, the innovative intersectional approach of this study means that prior manipulations in distal regions targeted mixed populations, making it difficult to directly infer the impact manipulating distal projections of pure populations. As a result, the manuscript's results section language should be revised to accurately reflect the findings (local and possibly distal as well) without giving the impression that they demonstrate local manipulations only.

2. The authors state that inhibition of VTA Glu neurons increases food intake, but it is not clear from how that the data presented in Fig. 19E demonstrates that conclusion. Could the authors please clarify?

3. Due to the manuscript's richness in experimental results, some interesting findings are unfortunately under-discussed. For instance, none of the anxiety-related experiments are mentioned in the discussion—despite an entire section of results (including Supplementary Figures 15–17, 19H–K, 22H–I, and 24H–K). Notably, it is quite striking that stimulation of both Glu-only and GABA-only neurons produces similar changes in the open field test.

4. I agree with Reviewer 4 that the experiments presented in fig 7 remain difficult to interpret, because the stimulation is not timed. Perhaps this caveat could be added to the discussion.

minor

1. Abstract: should mention clearly that the study was done on mice (+strain/background, sex)

2. Line 342: vta stim of gaba-only induced an increase in the time spent in the center (not decrease)

3. Typo: page 8, line 286 "not"

Reviewer #2

(Remarks to the Author)

No further comments

Reviewer #3

(Remarks to the Author)

I appreciate that the authors have changed the nomenclature to help with readability. I was willing to give the Authors creative autonomy after receiving the last version. However, this was a consistent response among reviewers and therefore a necessary change.

Two minor comments:

1) In the introduction, the authors might consider defining the rheobase, as many don't know what this term means. "Baseline excitability level (i.e., rheobase)" or "activation threshold (i.e., rheobase)" could both be used.

Second, on line 341, the authors indicate "Moreover, we observed that VTA photostimulation in GABA-only-ChR2-eYFP mice induced a decrease in the time spent in the center of the open field (Sup. Fig. 17E-G), as an indication of decrease anxiety." Unless I am missing something, less time in the center would be an indication of increased conflict avoidance and increased anxiety-like behavior. Based on the graph, I think the authors mean that time in the periphery was decreased. However, they should verify and make the correction.

Reviewer #4

(Remarks to the Author)

The second revision of this manuscript by Barbano et al. makes textual changes to improve clarity and adds additional discussion as well as additional experiments with inhibitory DREADD receptors in VTA-Glu only neurons. The strength of this paper remains the elegant EM work, accompanied by corroborating physiology and Fos studies, that provide an important characterization of local connectivity between different types of VTA neurons, which is of great benefit to the field. The behavioral experiments attempting to define the function of these subgroups, and the accompanying interpretation, are less clear. The authors argue in their rebuttal that “Our study serves as a pioneering step in characterizing the contributions of these VTA subpopulations, and we anticipate that future research will build upon our findings to further dissect their specific roles in food-motivated behaviors.” Viewed in this light, these experiments do indeed provide an initial characterization, but many unanswered questions remain. I additionally have some concerns about discrepancies between the text and the figures they describe.

Specific comments:

1) Authors should update all figures to include the clarifying “VTAglutamate only” etc. language. Many figures still just use the ConFoff language.

2) The authors state at line 341: “Moreover, we observed that VTA photostimulation in GABA-only-ChR2-eYFP mice induced a decrease in the time spent in the center of the open field (Sup. Fig. 17E-G), as an indication of decrease anxiety.”

The figures show an increase, not a decrease, in center time.

3) Line 489: “These findings reinforce the idea that the behaviors observed are governed by local interactions within the VTA, rather than long-range projections. This challenges the widely accepted notion, upheld for over 50 years, that VTA neurons regulate behavior through their distant connections.”

This is overstating the novelty of the results, particularly as this statement is made in the context of discussing GABAergic connectivity. Yes, VTA dopamine neurons regulate behavior through their distant connections, but the local inhibition of dopamine neurons by VTA GABA neurons is extremely well described (Tan et al. 2012 and others) and would certainly qualify as a “widely accepted notion.”

4) Some of the descriptions of results in the text don't appear to match the data in the supplemental figures, which is quite confusing.

Line 369: “Next, we tested food restricted mice and found that photoinhibition of VTAglutamate neurons in Glu-Halo-eYFP mice resulted in a decrease in the feeding initiation latency (Sup. Fig. 19D), together with an increase in the amount of food eaten (Sup. Fig. 19E), effects that were not observed in Glu-eYFP control mice.”

Line 506: “...local photoinhibition of VTAglutamate neurons is aversive and increased feeding behavior.”

Supplementary Figure 19D shows an increase (not decrease) in feeding latency, and a decrease (not increase) in food eaten. (These effects are also only during the post stimulation period, which is not mentioned in the text.) These effects are in the same direction (though much much smaller) as the effects with stimulation of VTA-Glu only neurons (Fig 6C-D), which seems to contradict the conclusions drawn.

5) Another discrepancy:

Line 389: “We then evaluated feeding behavior in food restricted mice and found that Glu-only-Gi-mCherry mice showed decrease in their feeding initiation latency (Sup. Fig. 22D), while they increased the amount of food they consumed after chemogenetic inhibition of VTAglutamate-only neurons (Sup. Fig. 22E).”

Supplementary Figure 22D shows no significant change in initiation latency.

6) A third discrepancy: “When feeding behavior was evaluated by the free-feeding test in food restricted mice, we found that photoinhibition of VTAGABA neurons slightly increased the feeding initiation latency (Sup. Fig. 24D) without modifying the amount of food eaten (Sup. Fig. 24E) in GABA-Halo-eYFP when compared with GABA-eYFP control mice.”

Sup. Fig. 24D shows a decrease, not an increase in latency (and again, this is only during the post-stimulation period, which is not mentioned in the text).

7) The authors seemed to take offense to my suggestion to more thoroughly discuss their RTPP results and how they contrast with the study by Yoo et al, stating in their rebuttal that “it is concerning that the reviewer evaluates our findings solely in the context of a single prior study, without considering the broader scientific contributions of our work, somehow overlooking critical insights that our study provides to the field.”

My intention was to prompt the authors to take one step further in the interpretation of their results, to speculate on why VTA-Glu stimulation generates this unique “shuttling” behavior, and compare where their results differ from other findings in order to benefit the field as a whole. It does not seem out of line to ask for a comparison to a specific study when that study has so many similarities to the current one. My focus on this specific study was because it describes interesting behavioral

responses to stimulation of VTA-Glu neurons that likely have important functional implications, and I believed that the current paper could be strengthened by discussing this.

I am confused by their comments that Yoo et al use “stimulation durations that exceed the natural activity patterns of neurons” (20 or 40 seconds), when in the previous paragraph they state that they themselves have seen activation durations in these neurons “varying from 1 to 20 seconds” in response to stimuli. So the reported preference in Yoo et al. for 1 or 5 s stimulation of VTA Glu neurons over 20 s stimulation seems perfectly valid.

The argument in the discussion that the discrepancy in the place preference studies could be due to “the high frequency used” for stimulation is inaccurate, as Yoo et al. tested frequencies from 1 to 40 Hz and did not see a preference at any frequency. The authors also state in the discussion that they “did not observe differences in the operant responses to obtain photostimulation of VTAglytamate-only neurons at varying frequencies or durations” without acknowledging that the range of durations tested did not include the longer durations tested by Yoo et al, making the comparison impossible.

I did not think it was an unusual or unreasonable request that the authors acknowledge where results differ with other reports and speculate as to why they differ, or to acknowledge where conclusions cannot be drawn because the same conditions were not tested. This type of honest discussion does not detract at all from the current study but moves the field forward. No offense was intended.

8) Finally, my previous concerns about the interpretability of the operant behaviors remain. The authors are arguing that VTA-Glu-only neurons are rewarding, but VTA-GABA-only neurons are aversive. And yet in both cases constant stimulation of these neurons in an operant chamber impairs learning (in slightly different ways). Why? Does Glu-only stimulation feel rewarding to the animals so they have no need to pursue further reward? Is it their motivation that is impacted? Or their ability to link actions and outcomes? Or is the entire dopamine system short-circuited by this stimulation? Of course the authors don't need to answer all questions related to these neurons in this one experiment, but it is difficult to see how our understanding of their function is increased here.

Reviewer #5

(Remarks to the Author)

Version 3:

Reviewer comments:

Reviewer #1

(Remarks to the Author)

The authors have addressed my concerns. I congratulate them on their interesting and comprehensive manuscript which will be of great use to the field.

Reviewer #3

(Remarks to the Author)

No further comments.

Reviewer #4

(Remarks to the Author)

I have one remaining concern regarding the over-interpretation of non-significant findings in this manuscript.

In the previous round of comments I pointed out Supplemental Figs. 19D-E, 22D, and 24D-E. In all cases the authors are making claims about results that are not statistically significant. In fact, I was quite confused during the last round of revision because where there are statistical differences (in Fig. 19 and 24) they are only during the post-stimulation period, and are going in the opposite direction of what the authors claim in the text. I now realize that my confusion was because the authors are referring in the text to “differences” during the stimulation period, but none of these are statistically significant, and frankly in most cases they don't look like anything close to a “trend”.

The revised text continues to claim differences in these figures that do not exist, saying things like “tended to” or “slight increase” without acknowledging the lack of statistical significance (except regarding Fig 22). For example, the legend for 24D simply says that photoinhibition “increased feeding initiation latency” but there is no significance during the stim period in the figure, and no post-hoc P value provided.

It is occasionally appropriate to identify trends in data that do not reach significance, but in these cases it must be made abundantly clear in the text that the difference is not significant, and the exact post-hoc P value should be provided. If that P value is larger than 0.1, I would argue that it is inappropriate to claim even a non-significant trend. The authors need to edit their text and figure legends to more accurately reflect the data as they are.

A final note. The revised text regarding Sup. Fig. 19D states: "This effect was more pronounced when compared to the increased latency observed after photostimulation of VTAglutamate-only neurons (Sup. Fig. 6C)."

First, Sup. Fig. 6C is a slice electrophysiology figure. I believe the authors are referring to Main Fig. 6C. Second, this comparison is not appropriate. The question is whether inhibiting these neurons has an effect compared to control animals, which it quite clearly does not.

Reviewer #5

(Remarks to the Author)

REVIEWER COMMENTS

Reviewer #1 (Remarks to the Author):

VTA neurons show remarkable heterogeneity and recently have been shown to have subpopulations expressing only GABA, only glutamate, or co-expressing both GABA and glutamate. In this manuscript, Barbano and colleagues explore the roles of these specific subpopulations in behavior using the INTERSECT viral approach. The authors show that optogenetic stimulation of VTA glutamate-only neurons is reinforcing, yet decreases obtaining rewards and slows appetitive operant learning. By contrast, VTA GABA-only stimulation is aversive, yet does not impair reward consumption, blocks appetitive operant learning and decreases locomotion. Finally, dual GABA/glutamate neuron stimulation is largely inert, but delays appetitive operant learning. Additionally, they report that glutamate-only cells synapse onto dopamine and glutamate cells, while GABA neurons target mostly dopamine cells. While the viral approach to dissecting neural subpopulations and microcircuit tracings are elegant and despite presenting some new interesting data, the study falls short of integrating the behavioral observations with the connectivity data into a cohesive story.

Major comments:

The optogenetic stimulation experiments do not yield a consistent picture. Broadly, glutamate only neurons are presented as rewarding and anxiolytic and GABA only neurons are presented as aversive, yet stimulating glutamate neurons disrupts reward consumption and slows appetitive learning and vexingly, GABA stimulation does not disrupt reward consumption (even though multiple authors have previously reported this effect) but has a profound effect on preventing appetitive learning. How is it that GABA stimulation is aversive, yet time in the open arms is not affected (stress tends to decrease time in open arms). Why does stimulating glutamate or GABA neurons lead to real time preference or avoidance, but not conditioned place preference or avoidance (especially since dopamine stimulation and inhibition leads to CPP/CPA)? The authors do not give a satisfying explanation of these discrepancies.

Regarding the points raised here by the reviewer, we have now included additional discussion regarding the role of VTA GABA neurons on reward consumption, which appears to depend on several factors, such as the test used to measure food intake, the hedonic properties of the food and the homeostatic state of the mice (page 10, line 399). We have shown that local GABA release from VTA^{GABA-only} neurons is aversive and inhibiting this release is rewarding when associated to a specific context. In the case of the elevated plus maze test, the stimulation of inhibition is administered throughout the entire “laser on” trial and mice are experiencing the aversive nature of GABA release from VTA^{GABA-only} neurons or the rewarding nature of inhibiting this release in both the open and closed arms of the maze. The fact that we did not observe any effect when photoinhibiting VTA^{GABA-only} neurons on the time spent in either open or closed arms confirm our photostimulation findings suggesting that VTA^{GABA-only} neurons do not play a role in anxiety-like behaviors. Regarding the real time experiments, our observations rely on the nature of the test. In contrast with the classical conditioned place preference or aversion experiments in which the mice are confined to the chamber while receiving the stimulus, the mice in the real time experiments here are free to leave the laser-paired chamber when they want, which does not account as a conditioning session.

Similarly, the authors have previously reported that GABA only and glutamate only neurons respond to both rewarding and aversive stimuli, along with cues that predict these outcomes, while dual GABA/glutamate neurons respond only to the stimuli and not to the predicting cues (Root et al 2020). It is unclear how the stimulation experiments in this manuscript builds on those findings. For example, one can come up with explanations for why consuming rewards evokes glutamate neuron activity but stimulating those neurons reduces reward consumption, but these explanations don't shed obvious light on the normal function of these neurons.

Findings from the Root et al. 2020 paper showed the response of VTA^{GABA-only}, VTA^{glutamate-only} and VTA^{glutamate-GABA} neurons when mice were presented with appetitive or aversive stimuli, or the cues predicting them. However, manipulations of those neurons to evaluate their role in behavior were not conducted on that study. The effects of locally activating or

inhibiting VTA^{GABA-only}, VTA^{glutamate-only} and VTA^{glutamate-GABA} neurons to probe their participation in different motivated behaviors are reported in this study.

This problem is rendered more challenging by the fact that there is no evidence that the optogenetic stimulation delivered (20 Hz for anywhere from 0.5 seconds to 5 minutes depending on the experiment) resembles the natural activity patterns these neural populations would produce. While optogenetic stimulation experiments are always somewhat artificial, they can be useful in demonstrating the causal impact of mimicking naturally observed activity patterns. From Root et al, it would seem that reward activates glutamate only neurons for a shorter period than GABA or dual GABA/glutamate neurons but these dynamics were not taken into account. To attempt to understand what these neurons do by activating them requires recording the normal activity rates and temporal dynamics of the different populations and tailoring the stimulation accordingly.

We (Miranda-Barrientos et al. 2021, Root et al. 2018) and others (Sagheddu et al. 2024, Li et al. 2012) have shown that the firing rate of the three VTA subpopulations under study can vary from 1 Hz to more than 30 Hz. As such, we have conducted parametric electrophysiological recordings (Sup. Fig. 5G-J, 6G-J), as well as parametric behavioral studies (Sup. Fig. 13) to elucidate the effects of varying the frequency and the duration of the stimulation on VTA^{GABA-only} and VTA^{glutamate-only} neurons. While a dose-response effect was evident only when stimulating VTA^{GABA-only} neurons at the electrophysiological level, increasing either the frequency or the duration of the photostimulation did not induce a dose-response effect on wheel turning to receive or to stop VTA photostimulation at the behavioral level.

By contrast, optogenetically inhibiting the neural activity of these different populations selectively during the different phases of behavioral tasks in which they are known to be active would provide some insight into what that neural activity contributes.

We have now included optogenetic inhibitory experiments to better elucidate the roles played by the different VTA subpopulations on behavior (page 9, line 336, Sup. Fig. 18-20).

Characterizing the synaptic connectivity of VTA glutamate and GABA neurons is an important contribution to the field. However, it is unclear what these results imply for the behavioral experiments. Does the fact that glutamate neurons synapse on 25% of TH+ neurons explain why stimulating these neurons do not induce hyperactivity (while stimulating GABA neurons which synapse on 65% of TH+ neurons induces hypoactivity)? Are any of the other discrepant behavioral results explained by the synaptic connections? Conversely, now that we know the connectivity, are there experiments to test the relationship between the microcircuitry and behavior?

We have now re-structured the manuscript to show the anatomical findings first. The behavioral experiments were designed based on the anatomical results. In addition, we have indicated in the discussion section the relationship between the anatomical and the behavioral findings (page 10, line 419; page 10, line 424; page 11, line 427).

Line 325:326. Why should a lack of dual GABA/glutamate synapses within the VTA explain the lack of behavioral responses? Don't these neurons synapse elsewhere?

Photostimulating at the level of neuronal cell bodies directly affects the neuron itself, influencing its firing rate and downstream signaling, while stimulating at the level of the neuronal target affects the activity of the receiving neuron by directly activating its synapses, potentially impacting the overall circuit function rather than the individual neuron's intrinsic properties. As such, the lack of behavioral responses may be related to the lack of local innervation by VTA^{glutamate-GABA} neurons. However, it may also be related to the nature of the neurotransmitters being co-released (excitatory and inhibitory), a fact that we have now acknowledged in the discussion section (page 10, line 391).

Minor.

Line 119: reward is mediated.... Aversion is mediated should be "photostimulation induces ..."

The subheading has been corrected following the reviewer's suggestion.

The manuscript switches between referring to the neural populations by the INTERSECT strategy (Coff/Fon), the promoter (VGaT-only), or the neuron type (GABA-only). I would pick the latter and stick with it throughout.

As suggested, we have now changed the nomenclature to refer to the different types of neurons and we have based it on the neuron subtype.

Reviewer #2 (Remarks to the Author):

This is an interesting paper with extensive and useful anatomical data on local connectivity within the VTA. The authors are well aware of the heterogeneity of VTA neurons. In this case, using a clever cross-sectional molecular approach in mice they address differences in connectivity between VTA neurons that express GABA only, Glutamate only and GABA-Glutamate. A major anatomical finding is that VTAglut-only neurons frequently establish synapses on VTAdopamine and VTAglut-only neurons, and that VTAGABA-only neurons mostly synapse on VTAdopamine neurons. Importantly, VTAglut-only neurons do not establish synapses on dual VTAdopamine-glut neurons, but do provide a major input to dual VTAglut-GABA neurons. Their ultrastructural data is quite convincing and novel.

1) In ex-vivo recording they demonstrate convincingly that VTAglut-only neurons monosynaptically connect with unidentified VTA neurons projecting to the nucleus accumbens. It's likely that the majority of the excited neurons were dopaminergic but a significant number could be non-dopaminergic (given their results, this omission is critical).

Although we are no longer including the electrophysiological findings related to the VTA-NAc glutamatergic pathway in the manuscript, we followed the suggestion of the reviewer and included the number of dopaminergic and non-dopaminergic neurons recorded after stimulation of VTA^{glutamate-only} and VTA^{GABA-only} neurons (Sup. Fig. 5-6).

Furthermore, the authors present no direct data to demonstrate that this projection produces the rewarding behavior of activation of the VTAglutamate only population of neurons.

Given that the identification of the downstream pathway/s underlying the behaviors observed after activation of the different subpopulations of VTA neurons is beyond the scope of the present work, we are no longer including findings related to downstream targets outside the VTA.

2) In addition to the anatomy, the authors undertook a variety of behavioral studies using channel rhodopsin stimulation of the various subpopulations of VTA neurons. Although, interesting differences in effect were observed when the different subpopulations were differentially activated those results are difficult to interpret, for a variety of reasons.

a) As the authors point out, within a given subgroup, neurons project to different target regions with different functions.

Although the scope of the manuscript was not to discuss downstream projections, a paragraph in the discussion (page 11, lines 458-469) highlights the sometimes-paradoxical findings obtained when stimulating local versus distal VTA targets.

b) Simultaneous activation of an entire subpopulation by any perturbation is probably quite rare in intact mice under physiological conditions. So while showing that some members of the subpopulation are sufficient to produce a given effect they may not be necessary. To establish that, the authors would have to use an inhibitory opsin, ideally in specific axon terminal regions.

As mentioned previously, the scope of the manuscript was not to focus on the downstream projections of the different subpopulations of VTA neurons but on the local microcircuits existing within the VTA. As such, we have now included experiments run with the inhibitory opsin Halorhodopsin (Sup. Fig. 18-20). We incorporated the description of the findings in the Results section (page 9, line 336) and we extended the discussion regarding these findings accordingly.

c) The authors did not demonstrate the effectiveness of their optogenetic stimulation, i.e. what per cent of the visualized neuron subtype *ex vivo*, was activated by the optogenetic stimulation. This might be particularly important for neurons that release more than one neurotransmitter, since different stimulation parameters can result in differential release of transmitter.

We have now included cFos quantification in combination with RNAscope phenotyping of VTA neurons, which demonstrates the effectiveness of our photostimulation protocols (page 6, line 186, Sup. Fig. 7-8).

Reviewer #3 (Remarks to the Author):

The study by Barbano et al focuses on the contribution of different subclasses of VTA neurons and their roles in reward-seeking and learning. This is one of a handful of studies to examine VTA microcircuitry and uses cutting-edge CLEM imaging to define the local connectivity. The paper includes several very nicely run studies that help to differentiate the roles of separate VTA subpopulations and their connectivity. However, there is also a disconnect between the way the paper is framed and the limitations of the experimental designs employed in the paper.

Major

1. The framework of the paper focuses on local connectivity within the VTA and VTA microcircuitry. However, most of the experiments (e.g., the Real-time place conditioning) involve stimulating what is almost certainly a mixture of local and projection neurons. While I recognize that the field lacks experimental methods that would allow the authors to overcome this hurdle, this still creates a problem for the overall framing, and I felt the authors could be more forward about the limitations of the place conditioning and optical ICSS experiments. If the authors wish to focus on the microstructure, they might consider leading with the CLEM and anatomical experiments that describe the microstructure and then framing the remaining experiments accordingly.

Following the reviewer's suggestion, we have re-structured the manuscript with a focus first on the anatomical, CLEM and electrophysiological findings, followed by the behavioral findings.

2. Several experiments employ the Newman-Keuls post hoc test, which poorly corrects for the familywise error rate and may decrease the reliability of the observed findings. This post hoc is typically found in GraphPad, whose documentation indicates that it should not be used.

Although the Newman-Keuls test controls the Family-Wise Error Rate (FWER) in a weak sense, it has been shown that with two or three groups, the Newman-Keuls procedure has strong control over the FWER (Keselman et al. 1991, *Psychological Bulletin*. 110 (1): 155–161). Following the reviewer's request, we have now use Tuckey HSD as a post hoc test that controls for FWER throughout the manuscript.

3. While male and female mice are both indicated in the methods, sex as a biological variable appears to be otherwise ignored and several experiments do not appear sufficiently powered to even detect sex differences, were they to exist.

We have now included a new supplementary figure (Sup. Fig. 11) showing that the behavioral effects observed in the place conditioning experiments were not sexually dimorphic and, as such, male and female mice were used together in

the rest of the study (page 7, line 225).

4. The authors follow the real-time place conditioning experiments with optical ICSS confirmatory experiments. These experiments include both a stimulation-seeking and a stimulation-avoidance protocol, depending on the outcome of the CPP. While it is clear why a stimulation-seeking protocol was used for the VTA glutamate-only neurons, and a stimulation-avoidance protocol was used for the VTA GABA-only neurons, it isn't clear why the choice was made to test VTA glutamate-GABA mice in a stimulation-seeking paradigm. While these mice showed no preference or aversion in the real-time place conditioning paradigm, they showed a bias towards stimulation avoidance, if anything. Repeating this experiment in the stimulation-avoidance protocol would be valuable.

As the reviewer indicated, activation of VTA^{glutamate-GABA} neurons induced an apparent, non-significant bias towards stimulation avoidance. However, when running the oICSS experiment targeting VTA^{glutamate-GABA} neurons, we observed an apparent, non-significant bias towards stimulation-seeking. As such and considering recent findings in the literature that suggest a lack of participation of these neurons in reward or aversion (reference 30 of the manuscript), we decided not to run the avoidance oICSS experiment.

5. The number of CreOn/FlpON mice included in the food self-administration experiments (Figure 5) is about half of that included for every other group and low enough to raise questions about the experimental power. With only 6 mice included, subtle differences in early operant behavior could contribute to the observed delays in learning. Did the authors conduct a power analysis to determine that these experiments were sufficiently powered? If not, it seems that these experiments should include a similar number of mice to those in the same figure.

Given our observations that VTA^{glutamate-GABA} neurons did not establish synapses locally and that their photostimulation was not rewarding nor aversive, we are no longer including other findings related to this VTA subpopulation in the manuscript.

6. The purpose of the final recording experiment is unclear. Is the mesoaccumbal pathway the only pathway where VTA VGlut2-only and VTA TH-only neurons project? It is simply not clear why this pathway was chosen above all of the other VTA-VGlut2 projections, and without that, this experiment is confusing and may even detract from the straightforward narrative the authors maintain up to this point.

Following the reviewer's suggestion, we have now removed the electrophysiological findings related to the VTA-NAc glutamatergic pathway.

Minor

1. While perhaps more technically accurate, the use of "Con/Foff-ChR2-eYFP" when referencing the different stimulation groups on Page 4 and beyond decreases readability. The authors establish that the viral vectors are specific in the first section, after which I would encourage them to switch to the nomenclature "VTA VGlut2-Only", as they use in the introduction, and later in the electron microscopy sections.

As we indicated in our reply to minor point 2 by reviewer 1, we have now updated the nomenclature to refer to the different subpopulations of VTA neurons throughout the manuscript.

2. For Figure 4 and the associated experiments, it is unclear what the differences in latency and the amount of food eaten are in comparison to. Is this the session prior?

As stated in the Methods section (page 22, line 1027), "The results were expressed as the difference between day 5 and day 4." We have now included this information in the figure legend.

3. On page 5, line 178, the authors conclude that “Termination of GABA release from VTA GABA-only neurons seems to facilitate feeding behavior in food-restricted mice”. However, this statement is not quite accurate. The authors are not terminating GABA release here but rather terminating the stimulation of GABA-only neurons. These neurons would presumably return to their normal firing patterns following the end of stimulation.

We thank the reviewer for raising this point. As suggested, the sentence now reads “terminating the stimulation of VTA^{GABA-only} neurons seems...”.

Reviewer #4 (Remarks to the Author):

This manuscript by Barbano et al. uses an intersectional viral approach to isolate neurons in the VTA that are glutamatergic only, GABAergic only, or express both glutamatergic and GABAergic markers. They find that optogenetic stimulation of Glu-only neurons is appetitive, GABA-only neurons is aversive, and Glu/GABA neurons is largely neutral. They also find that activating Glu only neurons inhibits feeding, and activating Glu only or Glu/GABA neurons during an operant task (lever press for food) delays learning on the task, while activating GABA neurons strongly inhibits learning. Using EM, the authors quantify the synaptic connectivity between these populations and with dopamine neurons in the VTA.

Overall, this paper increases our understanding of the different populations of glutamate and GABA neurons in the VTA, and represents an important contribution to our knowledge of the complexity of VTA microcircuits. The EM work in particular is an impressive and elegant analysis of the connectivity of these cell types. However, I have concerns regarding the design of some of the optogenetic experiments, which makes it difficult to interpret those results and limits our understanding of the functional role of these neurons.

Major points:

1) It has been established that some glutamate, GABA, and Glu/GABA neurons also co-express dopamine neuron markers (i.e. Th and/or DAT; see PMID: 37243808, PMID: 35385745). The authors should acknowledge that some of the results they see with optogenetics could be due to release of dopamine from these populations.

Following the reviewer’s suggestion, we have included a new experiment and figure (Sup. Fig. 12) showing that co-release of dopamine and glutamate after local stimulation of VTA^{glutamate-dopamine} neurons does not play a role in reward or aversion.

2) This lab previously found that stimulation of all VTA Glu neurons (not just Glu only) could drive a place preference (PMID: 26631475). However, another group found that stimulation of all VTA Glu neurons (not just Glu only) was moderately aversive in a RTPP task, but that mice made repeated entries in and out of the paired chamber to stimulate brief activation of VTA Glu neurons (PMID: 27976722). To facilitate a better comparison to this previous study, it would be interesting to plot the number of entries made into the paired chamber for each group in Fig 2. The authors should also discuss what might account for the differences between these studies and between stimulation of all Glu neurons vs the smaller Glu only or Glu/GABA populations.

As suggested, we have included an additional supplementary figure (Sup. Fig. 10) depicting the number of chamber entries made by the different groups in each phase of the place conditioning studies. In addition, we have expanded the discussion on the heterogeneity of VTA^{glutamate} neurons and their role in behavior (page 10, line 391; page 11, lines 442-449).

3) A major concern is with the design of the optogenetic stimulation during the operant task. The authors deliver 5s on/5s off stimulation during the entirety of the operant session. This makes it difficult to interpret what effect the

different VTA populations have on the specific aspects of this task. For example, given that stimulation of Glu only neurons is reinforcing, it is not particularly surprising that giving the animals continual reinforcement delays them from learning to press a lever to receive additional reinforcement (food pellet). And given that stimulation of GABA only neurons is aversive, it is not particularly surprising that animals in a highly aversive state would show impaired behavioral performance. But in neither case does this provide much insight into the specific role of these neurons in regulating motivated behavior. This could potentially be addressed with a study time-locking optogenetic stimulation of these neurons to specific behavioral events, or better yet (at least in the case of the Glu and Glu/GABA populations) using chemogenetic or optogenetic inhibition of these neurons to test their necessity in this task.

We have now included experiments run with inhibitory viral vectors (Sup. Fig. 18-20), which support our findings obtained with stimulatory viral vectors.

4) In supplementary Fig 1 the authors show example images of the different cell types in the VTA. If images are available, a more thorough description/analysis of the spatial distribution of these cells (and comparison to previous in situ work looking at the distribution of these cell types, PMID: 37243808) would be beneficial.

Following the reviewer's request, we have now included 2 additional figures showing the rostrocaudal and mediolateral distribution of the different cell types in the VTA (Sup. Fig. 1) and, more relevant to this study, the VTA rostrocaudal and mediolateral distribution of the axon terminals from VTA^{glutamate-GABA}, VTA^{glutamate-only} and VTA^{glutamate-GABA} neurons (Sup. Fig. 2).

5) In supplementary Fig 3 there appear to be substantial differences at baseline (no light stim) between the YFP control group and the Chr2 group in some cases. Was the difference in the first light off period in C and D tested for significance? And in I and J, is there an explanation for why the Chr2 mice overall spend more time in the open arms even with no stimulation? The same question applies to Supplementary Fig 4 C-D.

We have increased the number of experimental mice, and no differences are observed now at baseline in Sup. Fig. 15 (former Sup. Fig. 3). With regard to panels I and J, and given our observations that VTA^{glutamate-GABA} neurons did not establish synapses locally and that their photostimulation was not rewarding nor aversive, we are no longer including other findings related to this VTA subpopulation in the manuscript. Regarding Sup. Fig. 16 (former Sup. Fig. 4), the values at baseline are not significantly different, even when considering the less conservative of the post hoc tests (Fisher LSD, $p = 0.36$).

6) It would be helpful to plot edge time during the open field assay (Supp. Fig. 4), as this is another measurement of the anxiety-like behavior seen in the elevated plus assay.

As suggested, we have included an additional figure (Sup. Fig. 17) depicting the time spent in the periphery and center of the open field by the different experimental groups.

Minor Points

1) A schematic summarizing the connectivity patterns observed between different VTA cell types with the EM experiments would be very helpful.

Although we agree with the reviewer that a figure summarizing the results would be helpful, the fact that we have already added 15 new additional figures to the manuscript prevents us from adding additional material.

2) The finding that VTA GABA neurons make few synaptic contacts with other VTA GABA neurons is confirmatory of a previous study (PMID: 32541962), which used local or distal knockout of Vgat to examine GABAergic connectivity in the

VTA. This paper should be referenced.

We thank the reviewer for raising this point. The article by Soden et al. 2020 is now referenced (#23 in the references list).

3) In several places the authors specifically refer to identifying axon terminals contacting the soma of other cell types, without referencing synapses found on dendrites. Were somatic synapses the only type detected?

We found axon terminals contacting both the soma and the dendrites of postsynaptic neurons. We have added an example of a synapse on a dendrite in for Sup Fig. 4.

4) Both in figures and in the text, the use of “Con/Foff-ChR2-eYFP mice” etc. to denote the different groups makes the paper somewhat difficult to follow. Authors should consider using a shorthand designation (i.e. “Glu-only mice”) to make the paper easier to read.

We have updated the nomenclature to refer to the different subpopulations of VTA neurons but we consider that the nomenclature referring to the different experimental groups is specific and more accurate as is.

Reviewer #5 (Remarks to the Author):

We appreciate that the reviewer took part in this initiative to facilitate training in peer review and to provide appropriate recognition for Early Career Researchers who co-review manuscripts.

REVIEWER COMMENTS

Reviewer #1 (Remarks to the Author):

I appreciate that the authors have done a lot of experiments, including new ones in response to review, but I continue to have concerns that it is hard to draw collective conclusions about the role of these VTA subpopulations.

Major comments:

1. It is worth noting that the authors show that in their hands INTERSECT method has an error of at least 10-15% which limits the interpretability of findings.

It is well recognized that dual transgenic mouse lines may exhibit 10-20% off-target effects, which falls within the accepted range and align with the established standards in the field, as confirmed through personal communication with Dr. Lief Fenno, a leading authority in the development and application of INTRSECT viral vectors.

2. One of the most striking findings is that when using Vglut-cre mice to target neurons in the VTA 65% of them will be dual Glu-GABA neurons and when using vGAT-cre mice, 57% of the neurons will be dual Glu-GABA. The authors do not explicitly comment on this finding which would seem to be an important consideration for the field in interpreting such experiments. Yet in their own inhibition experiments (supplemental figures 18-19), the authors essentially ignore that using vglut and vgat cre mice will respectively include these large subpopulations. Apparently, INTERSECT did not work for selectively inhibiting specific subpopulations (why?) but global inhibition does not answer the scientific question.

It is unclear to us where the reviewer obtained the information on the proportions of dual VTA^{glutamate-GABA} neurons, as such, we can't comment on this issue. However, by conducting unique quantitative ultrastructural analysis of synapses established by local neurons, we demonstrated that dual VTA^{glutamate-GABA} neurons infrequently establish synapses with local VTA neurons. Therefore, the local activation or inhibition of VTA^{glutamate-GABA} neurons would be unlikely to have a significant effect at the microcircuit level, which is the primary focus of our study.

To address the reviewer's concerns, we have now included additional figures (Sup. Fig. 18, 21, 23, and 25) showing the results from testing the specificity of both custom and commercially available INTRSECT inhibitory viral vectors. While we identified specific INTRSECT inhibitory viral vectors for the targeting of VTA^{glutamate-only} neurons, we found that some of the tested INTRSECT inhibitory viral vectors for the targeting of VTA^{GABA-only} neurons failed to transfect these neurons or resulted in a non-functional DREADD receptor after transfection.

Regarding inhibition of the subpopulation of VTA^{glutamate-only} neurons, we added a new figure (Sup. Fig. 22) describing behavior in response of the specific inhibition of VTA^{glutamate-only} neurons by chemogenetics, or the specific inhibition of the total population of VTA^{glutamate} neurons by optogenetics (Sup. Fig. 19). Notably both types of inhibition procedures resulted in similar behavioral outcomes, further reinforcing our conclusions.

Regarding inhibition of the subpopulation of VTA^{GABA-only} neurons, as indicated above, none of the available INTRSECT inhibitory vectors specifically targeted this subpopulation of VTA neurons. Thus, given the limitation of available means for the specific inhibition of VTA^{GABA-only} neurons and considering that (a) we demonstrated that axons from VTA^{glutamate-GABA} neurons rarely establish local synapses and (b) our results from our gain-of-function studies clearly demonstrated that while activation of VTA^{glutamate-GABA} neurons DOES NOT affect behavior (but activation of VTA^{GABA-only} neurons induces changes in

behavior), it is reasonable to infer that in our reported loss-of-function experiment (targeting the whole population of VTA^{GABA} neurons) the changes in behavior were mediated by VTA^{GABA-only} neurons.

3. Another important finding is that combined Glu-GABA neurons do not form local synapses. However, when considering the behavioral consequences of exciting/inhibiting them, the authors seem to discount the possibility that they could exert influence by means of their distal projections. Moreover, they do not characterize where this major subpopulation project to or what their role in behavior is.

We have clarified in several parts of the discussion (i.e. page 11, lines 437-441, lines 444-445) that our study is focused on exploring local connections rather than long-range projections.

The reviewer stated that “Moreover, they do not characterize where this major subpopulation (sic, “VTA^{glutamate-GABA} neurons”) project (sic “s”) to or what their (sic “its”) role in behavior is”. This is obviously out of the scope of the current manuscript given that the focus of our study is on the characterization of an unrecognized microcircuitry within the VTA. As point of clarification, we have previously demonstrated that VTA^{glutamate-GABA} neurons mostly innervate the Lateral Habenula (Root et al, 2018), and we have unpublished results indicating that this mesohabenular pathway is not involved in the behaviors that we tested in the current study. We are willing to share with the editor these unpublished results, if needed.

Root DH, Zhang S, Barker DJ, Miranda-Barrientos J, Liu B, Wang HL, Morales M. 2018. Selective Brain Distribution and distinctive synaptic architecture of dual glutamatergic-GABAergic neurons. *Cell Reports*. 23:3465-3479.

4. Similarly, the justification given for omitting dual GABA-Glu neurons in further optogenetic experiments is that optogenetic behavioral experiments are intended to understand the impact of local VTA. However, there is no reason to believe that GABA -only and Glu-only do not exert their impact on conditioned place, self-stimulation, or feeding via long-range actions.

We indeed conducted several time-consuming experiments showing that VTA optical stimulation of VTA^{glutamate-GABA} neurons did not alter behavior, as shown in Figures 4 and 5. Moreover, we previously demonstrated that while dual VTA^{glutamate-GABA} neurons respond to aversive or rewarding stimuli, they do not respond to cues predicting these stimuli. Consistent with our findings, a recent study showed that activation of dual VTA^{glutamate-GABA} neurons does not induce place preference or aversion (McGovern et al., 2024), a point we have incorporated into the discussion section. Taken together, this evidence did not justify conducting additional costly and unnecessary experiments merely to confirm the lack of involvement of VTA^{glutamate-GABA} neurons in the behaviors under study.

Root DH, Barker DJ, Estrin DJ, Miranda-Barrientos JA, Liu B, Zhang S, Wang HL, Vautier F, Ramakrishnan C, Kim YS, Fenno L, Deisseroth K, Morales M. 2020. Distinct signaling by Ventral Tegmental Area glutamate, GABA, and combinatorial glutamate-GABA neurons in motivated behavior. *Cell Reports*. 32:1-14.

McGovern DJ, Phillips A, Ly A, Prévost ED, Ward L, Siletti K, Kim YS, Fenno LE, Ramakrishnan C, Deisseroth K, Ford CP, Root DH. (2024). Saliency signaling and stimulus scaling of ventral tegmental area glutamate neuron subtypes. *bioRxiv* [Preprint]. doi: 10.1101/2024.06.12.598688.

This reviewer has indicated several times that “there is no reason to believe that GABA -only and Glu-only do not exert their impact on conditioned place, self-stimulation, or feeding via long-range actions”. We are very aware of the literature on roles ascribed to VTA efferents in behavior, which we and others have been exploring for decades, as we detailed in one of our reviews (Morales and Margolis, *Nature Reviews in Neuroscience* 2017). In contrast, a detailed analysis at the synaptic and functional levels of a

VTA microcircuitry hasn't been done before, and we have highlighted this point in the discussion. Also in the discussion section, we provided a careful (unbiased) assessment of thoroughly collected data in the literature (vs speculation) demonstrating differential behavioral outputs as a result of local stimulation of VTA neurons vs stimulation of VTA efferents.

Morales M and Elyssa B Margolis (2017). *Ventral Tegmental Area: cellular heterogeneity, connectivity and behavior*. **Nature Reviews Neuroscience**. 18:73-85.

5. The findings are not well integrated with prior literature or internally. For example, Tan et al found that VTA GABA stimulation leads to real time conditioned place avoidance and persistent avoidance the following day unlike the current study. Similarly, there is no explanation of why VTA GABA-only neurons have divergent effects in the EPM and OF.

We have added sentences in the discussion (page 11, lines 475-480) acknowledging that the discrepancies between our study and that of Tan et al. are likely due to differences in the photostimulation protocol employed (30 s on/off in their case vs 20 Hz continuous stimulation in ours) or variations in the transgenic mouse lines used. While we used VGAT-cre mice, Tan et al. used GAD-cre mice resulting in the targeting of neurons that make GABA (as metabolite) and some neurons that utilize GABA as neurotransmitter. In this framework, we had previously demonstrated that in the VTA, many GAD neurons lack VGAT for the vesicular accumulation of GABA (Root et al. 2018). This clarification helps integrate our findings within the context of prior literature while highlighting potential methodological factors that may contribute to the observed differences. In addition, we have further elaborated on the key differences between the OF (threat-induced avoidance test) and the EPM (approach-avoidance conflict test, page 8, lines 322-323, page 9, lines 334-338). Moreover, we have expanded our interpretation of the EPM and OF findings by considering the differences between these behavioral tests in the context of VTA-induced GABA release from VTA^{GABA-only} neurons (page 9, lines 338-348).

6. The conclusions drawn sometimes outpace the evidence. For example, in addition to "anxiety" changes in behavior during the EPM (a conflict task) could reflect changes in motivation or ability to discriminate between open and closed arms.

We have included a sentence in the Results section (page 9, lines 348-349) acknowledging that, in addition to anxiety, behavioral changes observed in the elevated plus maze may also be influenced by factors such as motivation or the animal's ability to discriminate between the open and closed arms. This addition provides a more nuanced interpretation of our findings and ensures that our conclusions remain aligned with the available evidence.

7. The authors report non-uniform distribution of the different subpopulations, but did not specify the bregma coordinate borders of "medial-lateral" or "lateral". Could the absence of effect of optogenetic stimulation on dual Glu-GABA neurons reflect poor placement of the fiber relative to the population?

We have now specified the boundaries for the medial, mediolateral and lateral aspects of the VTA in Supplementary Tables 1, 2 and 3.

Based on our previously published anatomical findings, dual VTA^{glutamate-GABA} neurons are predominantly located in the medial aspect of the VTA, spanning from bregma -3.08 mm to -4.03 mm. The optic fiber was precisely placed at coordinates A/P = -3.2, M/L = 1.0 (with 10° angle), and D/V = -4.0. Additionally, the fiber's specifications (200 μm diameter and 0.39 numerical aperture) ensure that the laser stimulation adequately covers the medial aspect of the VTA, effectively targeting VTA^{glutamate-GABA}

neurons. Importantly, mice with fiber placements in the lateral aspects of the VTA were excluded from the analysis of this experimental group. To further ensure appropriate irradiance within our area of interest, we generally use an open-source tool provided by Stanford University (<http://web.stanford.edu/group/dlab/cgi-bin/graph/chart.php>).

8. Similarly, given the non-uniform distribution of the subpopulations, can the authors be certain that the behavioral effects of GABA-only or Glu-only optogenetic stimulation did not differ based on the distribution of the fiber placements?

We are confident that the behavioral effects observed are not due to artifacts from optic fiber placements. For VTA^{glutamate-only} neurons, their location within the VTA is similar to that of the dual VTA^{glutamate-GABA} neurons. We used identical coordinates to implant the optic fibers for this subpopulation, ensuring that laser stimulation fully covers the medial aspect of the VTA. Mice with fiber placements in the lateral aspects of the VTA were excluded from the analysis of this experimental group. Regarding VTA^{GABA-only} neurons, which are more diffusely distributed throughout the VTA, optic fibers were implanted at coordinates A/P = -3.2, M/L = +1.3 for half of the mice and -1.3 for the other half (with 10° angle), and D/V = -4.0. The fiber's specifications (200 μm diameter and 0.39 numerical aperture) ensured that the laser stimulation covered the entire VTA in one hemisphere and part of it in the other hemisphere, effectively targeting VTA^{GABA-only} neurons. Moreover, we obtained consistent results regardless of whether VTA^{GABA-only} neurons were targeted in the right or the left hemisphere, further supporting that our findings do not depend on the location of the optic fibers.

9. While the addition of subpopulation superscripts was a welcome addition, the manuscript remains very densely written and challenging to read.

The nomenclature for the experimental groups used in the behavioral studies has been revised to enhance the readability and clarity of the manuscript.

Minor:

There is a typo on lines 221-222

The typo has been corrected.

Reviewer #2 (Remarks to the Author):

The authors have addressed my concerns.

Reviewer #3 (Remarks to the Author):

Overall, the authors addressed all of my points and seemed to adequately address those raised by other reviewers. A few minor edits or clarification points:

Minor Comment:

For supplementary figures 7 and 8, it would be nice to separate the c-Fos plots for the Glutamate (Figure S7) and GABA (Figure S8). Since these are the populations being stimulated, it seems that they should be depicted separately. Related to this, the authors don't indicate whether they counted eYFP-positive cells from the Chr2 or explicitly avoided these. This is important, because even using a 4-color imaging

system, it seems that one of the RNAscope colors would have overlapped with the eYFP from the viral injections. Some clarification in the results and methods would be

We have revised supplementary figures 7 and 8 to incorporate the changes suggested by the reviewer. Regarding the eYFP signal, RNAscope procedures damage endogenous eYFP, rendering this fluorescent tag undetectable in VTA tissue following RNAscope processing. To confirm eYFP expression, we first performed immunostaining with anti-GFP antibodies on VTA sections. Once eYFP expression was confirmed, we used adjacent tissue sections from the same mouse for RNAscope analysis. We have clarified these procedures in the Materials and Methods section (page 24, lines 1139-1142).

Supplemental Figure 13- Title sentence says “glutamata” instead of “glutamate”

The typo has been corrected.

Reviewer #4 (Remarks to the Author):

The revised version of the manuscript by Barbano et al. makes some improvements on the original submission, including better organization starting with the EM data, and nice corroborating data with the Fos analysis. However, I feel that some of my concerns and those of other reviewers were not adequately addressed, particularly regarding the optogenetic behavioral experiments, and the results are not adequately discussed in the context of the previous literature.

1) 3 out of 4 reviewers commented that using the INTERSECT virus nomenclature (i.e. ConFoff) throughout the paper was confusing, and suggested that they instead refer to groups by the neuron type (i.e. Glu-only). The authors declined to do so for the behavioral studies, and the manuscript continues to suffer from poor readability due to this choice.

The nomenclature for the experimental groups used in the behavioral studies has been revised to enhance the readability and clarity of the manuscript.

2) The authors now report entries into the paired chamber in the RTPP experiment, but fail to put these results in the context of the previous literature as requested. The authors do in fact see greatly elevated entries into the paired side (Sup Fig 10c), similar to what was reported by Yoo et al (2016) with stimulation of all VTA-Glu neurons. These increased entries are an interesting phenotype, as this is not what is typically seen with activation of dopamine neurons (mice will enter the paired side and spend more time there but will not repeatedly enter and exit). How do the authors interpret these results? Why might the all-Glu vs Glu-only results be different in terms of total time spent on the paired side? What might this mean for the role of these neurons in reward signaling?

We have now expanded our discussion regarding the increased number of entries into the photostimulation-paired chamber (page 12, lines 517-527) following the photostimulation of VTA^{glutamate-only} neurons. In this revision, we emphasize that our findings are consistent with previous results from our lab (Wang et al., 2015) but differ from those reported by Yoo et al. (2016). We also provide potential explanations for these differences, including variations in stimulation frequency (20 Hz in our study vs. 40 Hz in Yoo et al.) and differences in the behavioral apparatus used (three-chamber vs. two-chamber setup).

Relatedly, the authors add data looking at ICSS with different frequency and duration stimulation (Sup. Fig 13). (Side note: The diagram in this figure appears to be mislabeled, showing vglut2-Cre/th-Flp instead of vgat Flp.) In contrast to Yoo et al, they do not see a frequency dependence for stimulation of VTA-Glu neurons. This is surprising but is not discussed. Yoo et al also found that mice preferred shorter stimulation of VTA-Glu neurons (1 or 5 s) relative to long duration stimulation (20 or 40 s). Unfortunately, the authors in this manuscript only test 1, 2.5, and 5 seconds, so it is impossible to know whether the Glu-only neurons act similarly to the all-Glu neurons tested by Yoo et al. (They likely are similar, given the high number of entries in the RTPP experiment, which indicates the mice are likely “shuttling” to turn the light on and off for brief durations.) Again, this is one of the most interesting phenotypes of these VTA-Glu neurons in the context of their role in reward learning, but unfortunately it is not discussed at all in the manuscript, and the experimental design did not take into account these previous findings.

We appreciate the reviewer’s careful examination of our work and for pointing out the labeling error in Supplementary Figure 13. This has now been corrected.

The rationale for these dose-response experiments stemmed from a previous request by Reviewer 1 to assess “the natural activity patterns these neural populations would produce.” In line with this request, previous studies from our group (Miranda-Barrientos et al. 2021, Root et al. 2018) have demonstrated that the firing rates of the three VTA subpopulations under investigation can range from 1 Hz to over 30 Hz, with stimulation durations varying from 1 to 20 seconds in response to appetitive or aversive stimuli (Root et al. 2020). We have now explicitly included this rationale in the Results section (page 7, lines 258-262). In addition, we have provided an explanation for the lack of frequency-dependent effects in our dose-response experiment involving VTA^{glutamate-only} neurons. We suggest that this could be due to a ceiling effect, where the chosen stimulation parameters may have already reached maximal efficacy. However, we also acknowledge the possibility of other contributing factors, such as depolarization block, which could limit neuronal excitability at higher stimulation frequencies (page 12, lines 527-529).

Regarding the comparison to Yoo et al., we do reference now their findings that mice preferred shorter trains of photostimulation targeting the entire population of VTA^{glutamate} neurons over longer ones (page 12, lines 519-520). However, their use of stimulation durations that exceed the natural activity patterns of the neurons under study makes direct comparisons between our studies challenging. While we appreciate the reviewer’s interest in this comparison, it is concerning that the reviewer evaluates our findings solely in the context of a single prior study, without considering the broader scientific contributions of our work, somehow overlooking critical insights that our study provides to the field.

3) The added experiments with inhibitory opsins are disappointing. While I appreciate the apparent technical issues with the INTERSECT inhibitory viruses, doing these experiments in the total VTA GABA and Glu populations is of limited benefit, and does not address the original concerns of myself and the other reviewers regarding the interpretation of the optogenetic results. Both Reviewer 1 and my comments suggested that time-locked manipulation of these neurons, particularly during the operant task, would be a much better way to test their function, but this was not done either with stimulation or inhibition. As it stands, the operant task remains uninterpretable and in my opinion does not add anything of benefit to the paper.

We appreciate the reviewer’s feedback and acknowledge the challenges associated with precisely targeting VTA^{glutamate-only} and VTA^{GABA-only} neurons using inhibitory opsins. In response to these concerns, we have now included a new figure (Sup. Fig. 22) that directly examines the role of VTA^{glutamate-only}

neurons in behavior following chemogenetic inhibition of this subpopulation. Importantly, we observed consistent results using both INTRSECT DREADDs and Cre-dependent photoinhibition, suggesting that while the inhibition of the entire population of VTA^{glutamate} and VTA^{GABA} neurons may have some limitations, it remains a valuable approach for studying their roles in behavior, especially in the light of our connectivity findings.

While we respect the reviewer's perspective, we do not share the view that the food self-administration tasks are uninterpretable or that they do not add meaningful insights to the paper. On the contrary, our findings reveal distinct behavioral outcomes following photostimulation of either VTA^{glutamate-only} or VTA^{GABA-only} neurons, providing novel evidence of their differential involvement in food self-administration and reinstatement of food-seeking behavior. This is a key contribution of our study, offering a foundation for future research into the role of VTA microcircuitry in motivated behaviors.

We acknowledge that time-locked manipulations during specific phases of the operant task (e.g., appetitive vs. consummatory aspects) would offer additional granularity in understanding these circuits. However, our approach follows established methodologies in the field, where general optogenetic manipulations have been instrumental in delineating neural circuit function. Our study serves as a pioneering step in characterizing the contributions of these VTA subpopulations, and we anticipate that future research will build upon our findings to further dissect their specific roles in food-motivated behaviors. While a more precise temporal approach is certainly of interest, it extends beyond the scope of the current study.

4) A summary schematic of connectivity of the different cell types would still be helpful.

We have added a new figure (Sup. Fig. 26) that provides a summary schematic illustrating the connectivity among the different cell types identified in this study.

Reviewer #5 (Remarks to the Author):

REVIEWER COMMENTS

Reviewer #1 (Remarks to the Author):

The revised manuscript by Barbano et al. includes an impressive amount of work which is now clearer. Overall, the manuscript will be a substantive addition to the field. There are still some points that require clarification:

Major

1. While the authors clearly demonstrate the existence of local connections among VTA GABAergic and glutamatergic (Glu) subpopulations, the current framing of the results unintentionally implies that the effects observed with optogenetic and chemogenetic manipulations are mediated solely via these local connections. However, the authors have not provided evidence that these subpopulations lack distal projections. The discussion section nicely addresses the distal vs local distinction and prior evidence about the topic. Importantly, the innovative intersectional approach of this study means that prior manipulations in distal regions targeted mixed populations, making it difficult to directly infer the impact manipulating distal projections of pure populations. As a result, the manuscript's results section language should be revised to accurately reflect the findings (local and possibly distal as well) without giving the impression that they demonstrate local manipulations only.

We have revised the language of the Results section to explicitly acknowledge the possibility that long-range projections could contribute to the observed behavioral effects. Specifically, we have added two clarifying sentences. First, following the description of the CPP and operant experiments, we now note that “While findings obtained after stimulating dual VTA^{glutamate-GABA} neurons may reflect activation of long-range projections, evidence suggests that effects observed after stimulating VTA^{glutamate-only}, or VTA^{GABA-only} neurons are primarily due to engagement of local VTA microcircuitry rather than long-range pathways (see Discussion section).” (page 7, lines 269-272), which we had previously developed in pages 12 (lines 492-496) and 13 (lines 559-568). Second, at the end of the Results section (page 10, lines 431-432), we included a statement acknowledging that long-range projections may have also contributed to the observed effects. We hope that these revisions address the reviewer's concern and improve the precision and transparency of the manuscript's interpretation.

2. The authors state that inhibition of VTA Glu neurons increases food intake, but it is not clear from how that the data presented in Fig. 19E demonstrates that conclusion. Could the authors please clarify?

We have clarified the description of the results to more accurately reflect the data shown in Sup. Fig. 19E. Specifically, we now state that Glu-Halo-eYFP mice exhibited greater food intake in the presence of photoinhibition compared to its absence, while Glu-eYFP control mice showed no difference in food intake between conditions (page 9, lines 376-378).

3. Due to the manuscript's richness in experimental results, some interesting findings are unfortunately under-discussed. For instance, none of the anxiety-related experiments are mentioned in the discussion—despite an entire section of results (including Supplementary Figures 15–17, 19H–K, 22H–I, and 24H–K). Notably, it is quite striking that stimulation of both Glu-only and GABA-only neurons produces similar changes in the open field test.

We have added a dedicated paragraph in the Discussion section (page 13, lines 544-557) addressing these findings. Specifically, we now discuss the behavioral effects observed in the open field test following stimulation of VTA^{glutamate-only} and VTA^{GABA-only} neurons and reference the relevant findings from supplementary figures 15–17, 19H–K, 22H–I, and 24H–K to ensure these results are properly integrated into the broader interpretation of our data.

4. I agree with Reviewer 4 that the experiments presented in fig 7 remain difficult to interpret, because the stimulation is not timed. Perhaps this caveat could be added to the discussion.

We have now added a statement to the Discussion section acknowledging the limitation that stimulation was not time-locked to specific phases of behavior, which could lead to alternative interpretations of the findings (page 11, lines 473-477).

minor

1. Abstract: should mention clearly that the study was done on mice (+strain/background, sex)

We have now revised the Abstract to specify that the studies were conducted in dual recombinase *vglut2-Cre/vgat-Flp* transgenic mice (page 2, line 4). Given the journal's word limit for the Abstract, additional details regarding the mouse background are provided in the Methods section (page 19, lines 842-843 and 844). Sex as a biological variable was evaluated (Supplementary Fig. 11), and data from male and female mice were pooled, as no sex differences were observed (page 7, lines 229-231). However, the number of male and female mice used in each experiment is clearly detailed in the Methods section.

2. Line 342: vta stim of gaba-only induced an increase in the time spent in the center (not decrease)

The typo has been corrected.

3. Typo: page 8, line 286 "not"

The typo has been corrected.

Reviewer #2 (Remarks to the Author):

No further comments

We thank the reviewer for the time and effort invested in reviewing our manuscript and for their positive assessment.

Reviewer #3 (Remarks to the Author):

I appreciate that the authors have changed the nomenclature to help with readability. I was willing to give the Authors creative autonomy after receiving the last version. However, this was a consistent response among reviewers and therefore a necessary change.

Two minor comments:

1) In the introduction, the authors might consider defining the rheobase, as many don't know what this term means. "Baseline excitability level(i.e., rheobase)" or "activation threshold (i.e., rheobase)" could both be used.

We have followed the reviewer's recommendation, and we now define rheobase as "the activation threshold" in the Introduction section (page 3, line 29).

Second, on line 341, the authors indicate "Moreover, we observed that VTA photostimulation in GABA-only-ChR2-eYFP mice induced a decrease in the time spent in the center of the open field (Sup. Fig. 17E-G), as an indication of

decrease anxiety." Unless I am missing something, less time in the center would be an indication of increased conflict avoidance and increased anxiety-like behavior. Based on the graph, I think the authors mean that time in the periphery was decreased. However, they should verify and make the correction.

We appreciate the reviewer's comment. This was a typo that has been corrected.

Reviewer #4 (Remarks to the Author):

The second revision of this manuscript by Barbano et al. makes textual changes to improve clarity and adds additional discussion as well as additional experiments with inhibitory DREADD receptors in VTA-Glu only neurons. The strength of this paper remains the elegant EM work, accompanied by corroborating physiology and Fos studies, that provide an important characterization of local connectivity between different types of VTA neurons, which is of great benefit to the field. The behavioral experiments attempting to define the function of these subgroups, and the accompanying interpretation, are less clear. The authors argue in their rebuttal that "Our study serves as a pioneering step in characterizing the contributions of these VTA subpopulations, and we anticipate that future research will build upon our findings to further dissect their specific roles in food-motivated behaviors." Viewed in this light, these experiments do indeed provide an initial characterization, but many unanswered questions remain. I additionally have some concerns about discrepancies between the text and the figures they describe.

Specific comments:

1) Authors should update all figures to include the clarifying "VTAglutamate only" etc. language. Many figures still just use the ConFoff language.

After carefully reviewing all figure legends in both the main and supplementary materials, we found that the "Con/Foff" and similar terms are used exclusively when referring to the names of the viral vectors employed in the experiments. As such, we have retained this nomenclature to ensure transparency, reproducibility, and accuracy, as these labels directly reflect the specific constructs used and how they can be ordered commercially. However, to enhance clarity for readers, we have ensured that the corresponding cell-type specificity (e.g., "VTA^{glutamate-only} neurons") is clearly described in the main text and figure legends wherever appropriate.

2) The authors state at line 341: "Moreover, we observed that VTA photostimulation in GABA-only-ChR2-eYFP mice induced a decrease in the time spent in the center of the open field (Sup. Fig. 17E-G), as an indication of decrease anxiety."

The figures show an increase, not a decrease, in center time.

The typo has been corrected.

3) Line 489: "These findings reinforce the idea that the behaviors observed are governed by local interactions within the VTA, rather than long-range projections. This challenges the widely accepted notion, upheld for over 50 years, that VTA neurons regulate behavior through their distant connections."

This is overstating the novelty of the results, particularly as this statement is made in the context of discussing GABAergic connectivity. Yes, VTA dopamine neurons regulate behavior through their distant connections, but the local inhibition of dopamine neurons by VTA GABA neurons is extremely well described (Tan et al. 2012 and others)

and would certainly qualify as a “widely accepted notion.”

Following the suggestion of the reviewer, we have removed the final sentence of the paragraph (page 12, line 496) to ensure our discussion more accurately reflects the current state of the field.

4) Some of the descriptions of results in the text don't appear to match the data in the supplemental figures, which is quite confusing.

Line 369: “Next, we tested food restricted mice and found that photoinhibition of VTAglutamate neurons in Glu-Halo-eYFP mice resulted in a decrease in the feeding initiation latency (Sup. Fig. 19D), together with an increase in the amount of food eaten (Sup. Fig. 19E), effects that were not observed in Glu-eYFP control mice.”

Line 506: “...local photoinhibition of VTAglutamate neurons is aversive and increased feeding behavior.”

Supplementary Figure 19D shows an increase (not decrease) in feeding latency, and a decrease (not increase) in food eaten. (These effects are also only during the post stimulation period, which is not mentioned in the text.) These effects are in the same direction (though much much smaller) as the effects with stimulation of VTA-Glu only neurons (Fig 6C-D), which seems to contradict the conclusions drawn.

In response to these comments, as well as those from Reviewer 1 (point 2), we have revised the text to clarify the comparisons made, along with the direction and timing of the observed effects (page 9, lines 373-378). Specifically, regarding the comparison between Figure 6 and Supplementary Figure 19, particularly Figures 6D and 19E, which both relate to food consumption, the data show opposing patterns. In Supplementary Figure 19E, photoinhibition of VTA^{glutamate} neurons leads to a modest increase in food intake during the photoinhibition period, while in Figure 6D, photostimulation of VTA^{glutamate-only} neurons results in reduced food intake. This contrast supports our interpretation that activity of VTA^{glutamate-only} neurons suppresses feeding, and that inhibiting this activity can partially lift that suppression, leading to increased consumption.

5) Another discrepancy:

Line 389: “We then evaluated feeding behavior in food restricted mice and found that Glu-only-Gi-mCherry mice showed decrease in their feeding initiation latency (Sup. Fig. 22D), while they increased the amount of food they consumed after chemogenetic inhibition of VTAglutamate-only neurons (Sup. Fig. 22E).”

Supplementary Figure 22D shows no significant change in initiation latency.

Although the figure legend for Supplementary Figure 22D states that J60 administration “tended to decrease feeding initiation latency” in Glu-only-Gi-mCherry mice, we have now clarified in the main text (page 10, line 396) that this trend did not reach statistical significance.

6) A third discrepancy: “When feeding behavior was evaluated by the free-feeding test in food restricted mice, we found that photoinhibition of VTAGABA neurons slightly increased the feeding initiation latency (Sup. Fig. 24D) without modifying the amount of food eaten (Sup. Fig. 24E) in GABA-Halo-eYFP when compared with GABA-eYFP control mice.”

Sup. Fig. 24D shows a decrease, not an increase in latency (and again, this is only during the post-stimulation period, which is not mentioned in the text).

We have clarified the description of the results to more accurately reflect the data shown in Sup. Fig. 24D. Specifically, we now state that GABA-Halo-eYFP mice exhibited a slight increase in feeding initiation latency in the

presence of photoinhibition compared to its absence, while GABA-eYFP control mice showed no difference in feeding initiation latency between conditions (page 10, lines 417-420).

7) The authors seemed to take offense to my suggestion to more thoroughly discuss their RTPP results and how they contrast with the study by Yoo et al, stating in their rebuttal that “it is concerning that the reviewer evaluates our findings solely in the context of a single prior study, without considering the broader scientific contributions of our work, somehow overlooking critical insights that our study provides to the field.”

My intention was to prompt the authors to take one step further in the interpretation of their results, to speculate on why VTA-Glu stimulation generates this unique “shuttling” behavior, and compare where their results differ from other findings in order to benefit the field as a whole. It does not seem out of line to ask for a comparison to a specific study when that study has so many similarities to the current one. My focus on this specific study was because it describes interesting behavioral responses to stimulation of VTA-Glu neurons that likely have important functional implications, and I believed that the current paper could be strengthened by discussing this.

We thank the reviewer for the clarification and appreciate the intent behind the suggestion. We would like to note that several key findings from the Yoo et al. study were already incorporated and discussed in the previous version of the manuscript and, while we appreciate the suggestion to further explore possible mechanisms underlying the distinct “shuttling” behavior observed in our RTPP experiments, we think the discussion is best kept focused on the empirical findings. Our approach reflects a preference for grounding interpretations in observed data rather than speculation, particularly given the format and editorial standards of *Nature Communications*, which tends to prioritize fact-based discussion over broader theorizing. That said, we acknowledge the importance of situating our work within the larger context of the field and have made efforts throughout the manuscript to do so thoughtfully.

I am confused by their comments that Yoo et al use “stimulation durations that exceed the natural activity patterns of neurons” (20 or 40 seconds), when in the previous paragraph they state that they themselves have seen activation durations in these neurons “varying from 1 to 20 seconds” in response to stimuli. So the reported preference in Yoo et al. for 1 or 5 s stimulation of VTA Glu neurons over 20 s stimulation seems perfectly valid.

We want to emphasize that we did not intend to question the validity of the preference findings reported in the Yoo et al. study. Our reference to stimulation durations was meant to highlight that, based on our own prior electrophysiological recordings, VTA^{glutamate-only} and VTA^{GABA-only} neurons typically fire in the range of 1–30 Hz, with activation durations in response to naturalistic stimuli generally falling between 1 and 20 seconds. This context was offered to frame how our stimulation parameters relate to the physiological activity patterns we have observed.

The argument in the discussion that the discrepancy in the place preference studies could be due to “the high frequency used” for stimulation is inaccurate, as Yoo et al. tested frequencies from 1 to 40 Hz and did not see a preference at any frequency. The authors also state in the discussion that they “did not observe differences in the operant responses to obtain photostimulation of VTAglutamate-only neurons at varying frequencies or durations” without acknowledging that the range of durations tested did not include the longer durations tested by Yoo et al, making the comparison impossible.

In response to the reviewer’s comments, we have removed the sentence speculating on potential reasons for the discrepancy in the place preference studies. Instead, we have added a statement acknowledging that “differences in experimental conditions between our study and previous work preclude a direct comparison” (page 12, lines 529-530).

I did not think it was an unusual or unreasonable request that the authors acknowledge where results differ with other reports and speculate as to why they differ, or to acknowledge where conclusions cannot be drawn because the same conditions were not tested. This type of honest discussion does not detract at all from the current study but moves the field forward. No offense was intended.

We agree that acknowledging differences with prior studies and considering possible reasons for those differences is an important part of scientific discourse and contributes to the advancement of the field. We appreciate the feedback and the opportunity to improve the clarity and rigor of our discussion.

8) Finally, my previous concerns about the interpretability of the operant behaviors remain. The authors are arguing that VTA-Glu-only neurons are rewarding, but VTA-GABA-only neurons are aversive. And yet in both cases constant stimulation of these neurons in an operant chamber impairs learning (in slightly different ways). Why? Does Glu-only stimulation feel rewarding to the animals so they have no need to pursue further reward? Is it their motivation that is impacted? Or their ability to link actions and outcomes? Or is the entire dopamine system short-circuited by this stimulation? Of course the authors don't need to answer all questions related to these neurons in this one experiment, but it is difficult to see how our understanding of their function is increased here.

We have now added a statement to the Discussion section (page 10, lines 473-477) acknowledging the limitation that stimulation was not time-locked to specific phases of behavior, which could lead to alternative interpretations of the findings, as the ones suggested by the reviewer. While we do not claim to resolve these mechanisms fully, we think our findings provide a meaningful first step toward characterizing the distinct contributions of VTA^{glutamate-only} and VTA^{GABA-only} neurons to food-motivated behavior and highlight the need for future experiments using more temporally precise manipulations.

Reviewer #5 (Remarks to the Author):

We thank the reviewer for participating in this initiative.

ANSWERS TO REVIEWER COMMENTS

Reviewer #1 (Remarks to the Author):

The authors have addressed my concerns. I congratulate them on their interesting and comprehensive manuscript which will be of great use to the field.

We thank the reviewer for their positive feedback and thoughtful assessment.

Reviewer #3 (Remarks to the Author):

No further comments.

We thank the reviewer for their positive feedback and thoughtful assessment.

Reviewer #4 (Remarks to the Author):

I have one remaining concern regarding the over-interpretation of non-significant findings in this manuscript.

In the previous round of comments I pointed out Supplemental Figs. 19D-E, 22D, and 24D-E. In all cases the authors are making claims about results that are not statistically significant. In fact, I was quite confused during the last round of revision because where there are statistical differences (in Fig. 19 and 24) they are only during the post-stimulation period, and are going in the opposite direction of what the authors claim in the text. I now realize that my confusion was because the authors are referring in the text to “differences” during the stimulation period, but none of these are statistically significant, and frankly in most cases they don’t look like anything close to a “trend”.

The revised text continues to claim differences in these figures that do not exist, saying things like “tended to” or “slight increase” without acknowledging the lack of statistical significance (except regarding Fig 22). For example, the legend for 24D simply says that photoinhibition “increased feeding initiation latency” but there is no significance during the stim period in the figure, and no post-hoc P value provided.

It is occasionally appropriate to identify trends in data that do not reach significance, but in these cases it must be made abundantly clear in the text that the difference is not significant, and the exact post-hoc P value should be provided. If that P value is larger than 0.1, I would argue that it is inappropriate to claim even a non-significant trend. The authors need to edit their text and figure legends to more accurately reflect the data as they are.

In response to the reviewer’s comment, we have revised the text (page 9, lines 373-379; page 10, lines 397-398; page 10, lines 418-421) and supplementary figure legends (Supplementary figures 19, 22 and 24) to more precisely reflect the data presented.

A final note. The revised text regarding Sup. Fig. 19D states: “This effect was more pronounced when compared to the increased latency observed after photostimulation of VTA^{glutamate-only} neurons (Sup. Fig. 6C).”

First, Sup. Fig. 6C is a slice electrophysiology figure. I believe the authors are referring to Main Fig. 6C. Second, this comparison is not appropriate. The question is whether inhibiting these neurons has an effect compared to control animals, which it quite clearly does not.

In response to the reviewer’s comment, we have removed the sentence “This effect was more pronounced when compared to the increased latency observed after photostimulation of VTA^{glutamate-only} neurons (Sup. Fig. 6C)” to

ensure an accurate interpretation of the data.

Reviewer #5 (Remarks to the Author):

We thank the reviewer for participating in this initiative.

** See Nature Portfolio's author and referees' website at www.nature.com/authors for information about policies, services and author benefits.

This email has been sent through the Springer Nature Tracking System NY-610A-NPG&MTS

Confidentiality Statement:

This e-mail is confidential and subject to copyright. Any unauthorised use or disclosure of its contents is prohibited. If you have received this email in error please notify our Manuscript Tracking System Helpdesk team at <http://platformsupport.nature.com>.

Details of the confidentiality and pre-publicity policy may be found here <http://www.nature.com/authors/policies/confidentiality.html>